

# MuSA: The Multiscale Snow Data Assimilation System (v1.0)

Esteban Alonso-González[1*], Kristoffer Aalstad[2*], Mohamed Wassim Baba[3], Jesús Revuelto[4], Juan Ignacio López-Moreno[4], Joel Fiddes[5], Richard Essery[6], and Simon Gascoin[1]

[1]Centre d'Etudes Spatiales de la Biosphère, Université de Toulouse,CNRS/CNES/IRD/INRA/UPS, Toulouse, France
[2]Department of Geosciences, University of Oslo, Oslo, Norway
[3]Center for Remote Sensing Application (CRSA), Mohammed VI Polytechnic University (UM6P), Ben Guerir, Morocco
[4]Instituto Pirenaico de Ecología, CSIC, Zaragoza, Spain
[5]WSL Institute for Snow and Avalanche Research SLF, Davos, Switzerland
[6]School of GeoSciences, University of Edinburgh, Edinburgh, UK
[*]These authors contributed equally to this work.

**Correspondence:** Esteban Alonso-González (esteban.alonso-gonzalez@univ-tlse3.fr)

**Abstract.** Accurate knowledge of the seasonal snow distribution is vital in several domains including ecology, water resources management, and tourism. Current spaceborne sensors provide a useful but incomplete description of the snowpack. Many studies suggest that the assimilation of remotely sensed products in physically based snowpack models is a promising path forward to estimate the spatial distribution of snow water equivalent (SWE). However, to date there is no standalone, open
source software dedicated to snow data assimilation. Here we introduce a new data assimilation toolbox, the Multiscale Snow Data Assimilation System (MuSA). MuSA was developed to fuse remotely sensed information with the energy and mass balance Flexible Snow Model (FSM2). MuSA was designed to be user-friendly and scalable. It enables assimilation of different state variables such as the snow depth, SWE, snow surface temperature, binary or fractional snow-covered area, and snow albedo and could be easily upgraded to assimilate other variables such as liquid water content or snow density in the future.
MuSA allows the joint assimilation of an arbitrary number of these variables, through the generation of an ensemble of FSM2 simulations. The characteristics of the ensemble (i.e. the number of particles and their covariance) may be controlled by the user, and it is generated by perturbing the meteorological forcing of FSM2. The observational variables may be assimilated using different algorithms including the particle filters and smoothers as well as ensemble Kalman filters and smoothers along with their iterative variants. We demonstrate the wide capabilities of MuSA through two snow data assimilation experiments.
First, 5 m resolution snow depth maps derived from drone surveys are assimilated in a distributed fashion in the Izas catchment (Central Pyrenees). Furthermore, we conducted a joint assimilation experiment, fusing MODIS land surface temperature and fractional snow-covered area with FSM2 in a single cell experiment. In light of these experiments, we discuss the pros and cons of assimilation algorithms, including their computational cost.

## 1  Introduction

The snow cover has a profound effect on the water cycle (García-Ruiz et al., 2011) and ecosystems (Lin and West, 2022) of high latitude and mountain regions. It represents a natural reservoir of freshwater resources, sustaining crop irrigation, hydropower





generation and drinking water supply to a fifth of humanity (Barnett et al., 2005). In addition, the ski industry is an important economic driver in many mountain areas. As a consequence, a good knowledge of the snow cover properties has a strong scientific, societal, and economic value (Sturm et al., 2017).

Due to the harsh environmental conditions that often prevail in snow dominated areas, in situ monitoring of the snowpack based on automatic devices and weather stations is both costly and logistically challenging. In addition, due to the variable nature of the snowpack (López-Moreno et al., 2011), even dense monitoring networks may suffer from a lack of representativeness (Molotch and Bales, 2006). Yet, estimating SWE spatial distribution is important to make accurate predictions of snowmelt runoff in alpine catchments. Satellite remote sensing provides spatial information about snow-related variables in-

cluding (i) snow cover spatial extent at various spatio-temporal scales (Aalstad et al., 2020; Gascoin et al., 2019; Hall et al., 2002; Hüsler et al., 2014), (ii) snow depth (Lievens et al., 2019; Marti et al., 2016), (iii) albedo (Kokhanovsky et al., 2020) or (iv) snow surface temperature (Bhardwaj et al., 2017) at the global scale. However, the direct estimation of key variables such as the snow water equivalent (SWE) or density by means of remote sensing techniques, remains challenging (Dozier et al., 2016). The only remote sensing tools that have shown some potential to retrieve SWE are passive microwave sensors.

Unfortunately, their coarse resolution and the fact that they tend to saturate above a certain SWE threshold, prevents their usage over mountainous regions or areas with a thick snowpack (Luojus et al., 2021).

Numerical modeling allows the estimation of SWE at different spatio-temporal scales using meteorological information derived from automatic weather stations (Essery, 2015; Liston and Elder, 2006a) or atmospheric models (Alonso-González et al., 2018; Wrzesien et al., 2018). Nonetheless, snowpack models exhibit a number of limitations that may cause strong

biases in the SWE simulations (Wrzesien et al., 2017). Snowpack model uncertainties originate partly from their simplified representation of physical processes (Günther et al., 2019; Fayad and Gascoin, 2020), but most importantly from errors in the meteorological forcing (Raleigh et al., 2015). In this context, the fusion of remote sensing products with snowpack models using data assimilation is key to improve snowpack simulations (Girotto et al., 2020; Largeron et al., 2020).

Several remotely sensed products may be used to update snowpack models. Snow covered area from optical sensors was

the first product to be assimilated (Clark et al., 2006; Durand et al., 2008; Kolberg and Gottschalk, 2006). Snow cover area (SCA) assimilation remains extensively used in both distributed (Margulis et al., 2016) and semi-distributed models (Thirel et al., 2013) due to the long time series of SCA observations and the development of new higher resolution products (Baba et al., 2018). It is possible to further improve snowpack simulations by assimilating snow depths (Deschamps-Berger et al., 2022; Smyth et al., 2020). The assimilation of remotely sensed surface reflectances may also be beneficial (Charrois et al.,

2016; Cluzet et al., 2020; Revuelto et al., 2021b), but further research is needed on this topic to demonstrate advantages over assimilating derived higher level products such as fractional snow cover area (FSCA) and albedo.

Whereas snowpack models are increasingly available as open source software and remote sensing products as open data, to our knowledge there is no standalone open source application to develop snow data assimilation experiments. This specific issue was highlighted by Fayad et al. (2017) as a strong limitation to advance knowledge on the snow cover in regions which

receive less attention from the mainstream research community. In addition, some data assimilation frameworks are based on highly specific implementations tied to operational constraints (Cluzet et al., 2021). This situation prevents the develop-





ment of reproducible snow data assimilation studies and challenges the comparison of the performance of different snow data assimilation algorithms.

This is why we have developed a new open source data assimilation toolbox, the Multiscale Snow Data Assimilation system
(MuSA). MuSA is an ensemble-based snow data assimilation tool. It enables the fusion of multiple observations with a physically based snowpack model while taking into account various sources of model, forcing, and measurement uncertainty. It is an open-source collaborative project entirely written in the Python programming language. It should facilitate the development of snow data assimilation experiments, as well as the generation of snowpack reanalyses and near real time snowpack monitoring. It was designed with a modular structure to foster collaborative development, allowing advanced users to seamlessly
implement new features with minimal effort. In the following sections, we describe the features of MuSA and show different examples of its usage, assimilating remote sensing products of different spatio-temporal resolutions.

## 2   Overview of the data assimilation system

The core of MuSA is the energy and mass balance of the snowpack model, the Flexible Snow Model (FSM2 Essery, 2015). FSM2 has from two to three levels of representation of different key processes related to the energy and mass balance of the
snowpack. The most complex configuration is chosen by default in MuSA, leading to a more detailed simulation of the internal snowpack processes. Albedo is computed from the age of the snow, decreasing its value as snow ages and increasing it with fresh snowfalls, instead of diagnosing it as a function of the surface temperature. Thermal conductivity of the snowpack is computed as a function of the snow density, instead of using a fixed value. Snow density is computed considering overburden and thermal metamorphism, instead of using an empirical estimation increasing the density as function of age or using a fixed
value. Turbulent energy fluxes are computed as a function of atmospheric stability. Melt water percolation in the snowpack is computed using the gravitational drainage, instead of a bucket model. Although this is the default configuration, it is possible to choose any other FSM2 setup, which may result in slight performance differences both in terms of the computational cost and accuracy of the model (Günther et al., 2019).

MuSA was designed as a Python program encapsulating the FSM2 Fortran code (Figure 1). It handles the forcing and initial
files as well as the FSM2 runs and outputs, by internally generating the needed ensemble of simulations from simple configuration (config.py) and constants (constants.py) files that should be filled by the user. Then, it solves most of the challenges of ensemble-based snow data assimilation frameworks for the user. The data assimilation algorithms are independently implemented on a grid cell basis, allowing both point scale and parallel spatially distributed simulations. Also, in its current version MuSA provides support for the direct insertion of snow depth. However this feature may not be maintained in the future as it
involves strong assumptions especially with multilayer geometries. The outputs of MuSA consist of the posterior mean snow simulation from FSM2 (updated_idx_idy.csv), the posterior standard deviations of the FSM2 ensemble (sd_idx_idy.csv), and information related to the posterior perturbation parameters and the observations (DA_idx_idy.csv) and the original simulation without perturbation (OL_idx_idy.csv). Additionally, it is possible to store the posterior ensembles of every cell.

The following data assimilation algorithms are currently implemented in MuSA:



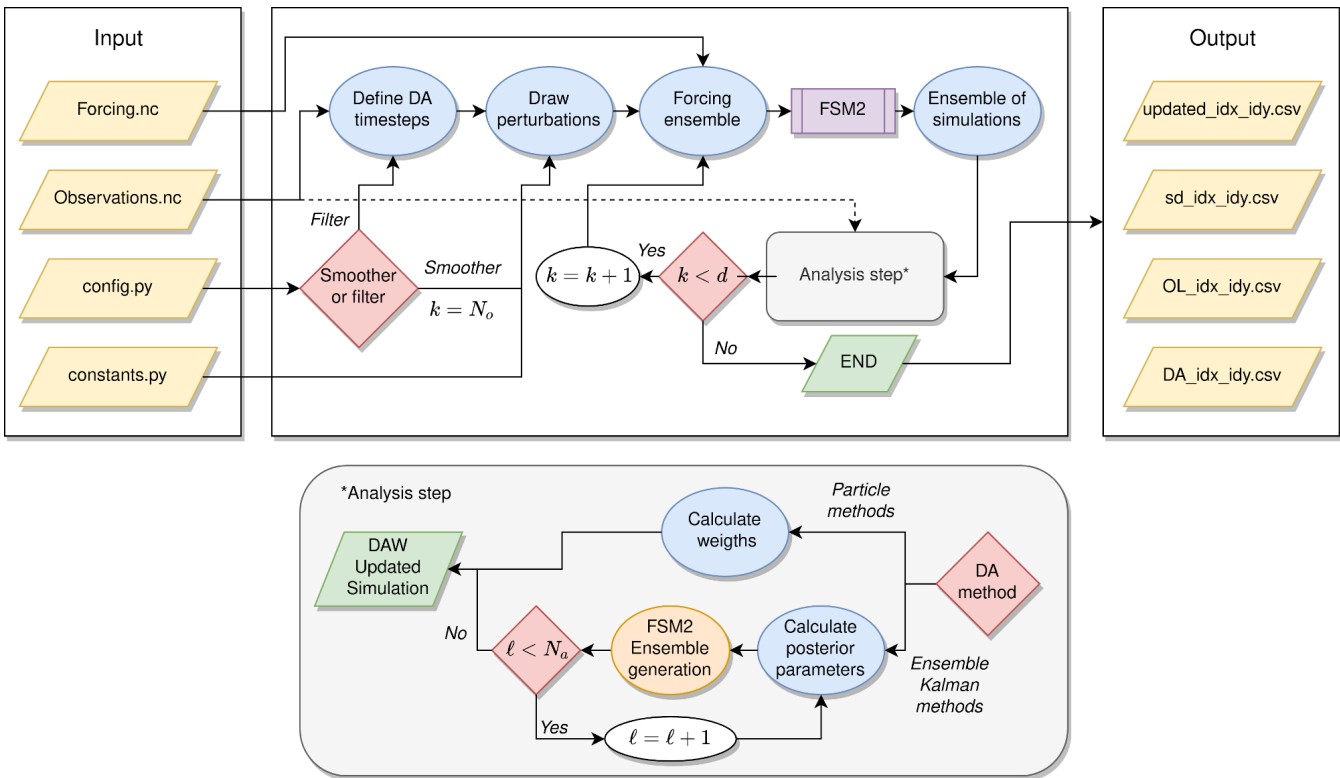

**Figure 1.** MuSA internal workflow. $k$, $d$, $\ell$, $N_a$, idx, and idy refer to the timing of the analysis step, number of observations, Kalman iteration number, total number of assimilation cycles, longitude cell index and latitude cell index respectively.

- Particle Filter (PF; Gordon et al., 1993; van Leeuwen et al., 2019)

- Ensemble Kalman Filter (EnKF; Evensen, 1994; van Leeuwen, 2020)

- Ensemble Kalman filter with Multiple Data Assimilation (EnKF-MDA; Emerick and Reynolds, 2012)

- Particle Batch Smoother (PBS; Margulis et al., 2015), a batch smoother variant of the PF.

- Ensemble Smoother (ES; van Leeuwen and Evensen, 1996), a batch smoother variant of the EnKF.

- Ensemble Smoother with Multiple Data Assimilation (ES-MDA; Emerick and Reynolds, 2013)

Note that the Ensemble Kalman schemes involving multiple data assimilation are iterative schemes based on tempering the likelihood (Stordal and Elsheikh, 2015; van Leeuwen et al., 2019). For the PF, several standard resampling algorithms (see Li et al., 2015, for a review) are available in MuSA, namely: bootstraping, residual resampling, stratified resampling, and systematic resampling. In addition to these standard resampling techniques, we have also implemented a heuristic approach based on redrawing from a normal approximation to the posterior which is loosely inspired by the idea of using a proposal





distributions (Särkkä, 2013; van Leeuwen et al., 2019). This redrawing from the approximate posterior generates new samples of perturbation parameters at each assimilation step. In the case of a total collapse of the PF with this resampling strategy the posterior standard deviation of the perturbation parameters is 0 (all weight is assigned to one particle) which corresponds to a degenerate distribution. In this case, drawing new perturbation parameters from the degenerate posterior distribution would not generate new particles. To avoid this, in case of a complete collapse of the PF, MuSA uses the prescribed prior standard deviations for generating perturbation parameters, corrected by a scaling factor (to be selected by the user) to avoid over-inflating the ensemble.

The inputs of MuSA are composed of (i) gridded meteorological forcing to generate the ensemble of FSM2 simulations and (ii) gridded observations which are assimilated. Optionally, it is possible to provide a mask, covering areas where simulations should not be implemented in order to save computational resources or restrict the runs to a specific area inside the provided domain. These inputs must be provided as files encoded in the Network Common Data Form (NetCDF; Rew and Davis, 1990) format. MuSA is able to handle different observational products as long as the meteorological forcing and the observations share the same geometry (i.e. same number of cells in the longitudinal and latitudinal dimensions). As in many other distributed snowpack models, there is currently no communication between grid cells in MuSA. This makes MuSA easily parallelizable as each cell is simulated and updated completely independently of the others (i.e., embarrassingly parallel). As a result, the computational cost of the runs scales almost linearly with the number of computational units or the number of cells to solve respectively. Different parallelization schemes are already implemented, including multiprocessing for single node runs and message passing interface (MPI) and Portable Batch System job arrays, providing support for computing clusters. Due to the embarrassingly parallel nature (no communication between tasks) of the MuSA computational problem, other parallelization schemes can easily be implemented. All the data processing is done by default in the temporary file system, but the user can choose any other location such as a specific temporary folder in a multi-node cluster or a random access memory drive in a local workstation to speed up the I/O processes.

## 2.1 Ensemble generation

Other than for direct insertion, the DA algorithms implemented in MuSA all require a prior ensemble (i.e. a collection) of simulations to represent uncertainty. The number of ensemble members (also known as particles; van Leeuwen et al., 2019) in the ensemble should be specified by the user, as it drastically affects both the computational cost of the experiments and the performance of the data assimilation algorithms. To generate an ensemble of snowpack state trajectories, MuSA perturbs the forcing and performs an ensemble of FSM2 runs with the perturbed forcing. An arbitrary number of forcing variables can be perturbed. The perturbation of the forcing is performed by drawing random perturbation parameters from a normal distribution defined by its standard deviation and mean, or the mean and standard deviation of the underlying normal function in case of choosing a log-normal probability distribution. The prior standard deviation and mean of the distribution should be provided by the user. MuSA supports normal or log-normal probability functions to generate constant in time additive or multiplicative perturbations respectively, depending on the physical bounds of the variable to be perturbed. For example, additive perturbations





can be used to generate air temperature time series, whereas multiplicative perturbations are recommended for precipitation to
avoid negative values, or shortwave radiation to avoid negative values and positive night-time values.

Once the forcing is perturbed, MuSA partitions the precipitation between the liquid and solid phase. To estimate the phase
of the precipitation, two different approaches are implemented. The simpler approach is based on a logistic function of the 2
m air temperature (Kavetski and Kuczera, 2007). A more complex psychrometric energy balance method is also implemented
in MuSA. This approach uses the relative humidity and 2 m air temperature to infer the surface temperature of the falling
hydrometeors, from which the phase of the precipitation can be estimated (Harder and Pomeroy, 2013).

## 2.2   Meteorological forcing

The variables required as forcing to run MuSA include downwelling (i.e., incoming) shortwave and longwave radiation
$[\mathrm{W\,m^{-2}}]$, total (sum of liquid and solid) precipitation $[\mathrm{kg\,m^{-2}s^{-1}}]$, surface atmospheric pressure [Pa], 2 m air temperature
[K], relative humidity [%], and 10 m wind speed $[\mathrm{m\,s^{-1}}]$. In the current version of MuSA, the forcing must be provided in
the netCDF format at an hourly time step in a grid based geometry, without any specific requirement on the native spatial
resolution. The most likely candidates to be used as meteorological forcing for MuSA will be the outputs of atmospheric
simulations or automatic weather station data. These include atmospheric reanalyses, such as ERA5 (Hersbach et al., 2020)
or MERRA-2 (Gelaro et al., 2017) and regional climate model outputs (Alonso-González et al., 2021) as well the outputs of
different downscaling approaches driven by the aforementioned datasets (e.g. Fiddes and Gruber, 2014; Gutmann et al., 2016;
Havens et al., 2017; Fiddes et al., 2022; Liston and Elder, 2006b). Other sources of information include automatic weather
stations or gridded products derived from the interpolation of point scale meteorological information. The forcing variables
should be provided in the International System of Units, but it is possible to perform simple unit transformations internally to
facilitate the preprocessing of different forcing data sources, using the following affine transformation.

$$\psi_k = a\phi_k + b, \tag{1}$$

where $\phi_k$ (for $k = 0, \ldots, n$) is a forcing time series provided by the user with units $[\Phi]$ and $\psi_k$ with converted units $[\Psi]$ is the
transformed forcing time series after scaling by $a$ with units $[\Psi\,\Phi^{-1}]$ and translating by $b$ with units $[\Psi]$.

It should be noted that the forcing is read along the time dimension. As mentioned above, each grid cell is solved inde-
pendently including the reading of the forcing. However, this memory cost reduction comes with an associated computational
cost due to the potential need of reading a long time dimension. To alleviate this, the computational cost of MuSA can be
reduced by setting the chunk of the netCDF forcing files along the time dimension, specially with long or large (i.e. many grid
cells) simulations. To speed up the relaunching of the simulations, MuSA generates an intermediate binary file with the forcing
information of each cell needed to run a complete simulation. This accelerates the initialization of experiments that involve
performing different MuSA runs for the same grid cells.





## 2.3 Observations and masked cells

MuSA is able to assimilate an arbitrary number of different observations that can be irregularly distributed in time and space. It is also possible to develop joint assimilation experiments that assimilate more than one variable type at the same time. To date, MuSA offers support to assimilate snow depth [m], snow water equivalent (SWE) [mm], surface temperature [K], snow-covered area (either binary or fractional) [-] and albedo [-]. A temporally and spatially constant scalar corresponding to the assumed observation error variance must be provided for each of the observed variables that are assimilated. This implies a diagonal

observation error covariance matrix, $\mathbf{R}$, which is tantamount to assuming that observation errors are uncorrelated in both time and space. The geometry of the observations must match the geometry of the forcing, i.e. both datasets must share the same grid extent and resolution. The time step, however, does not have to be the same so that the observations can be irregularly spaced in time. Also, it is not necessary that the observations of the different variables occur at the same time step, allowing the assimilation of products from different satellite platforms with different orbits. Optionally, a binary mask can be provided in a

separate file to delineate the basin of interest. If provided, MuSA will skip the cells that are not covered by the mask. Again, the mask must be provided on the same grid as the forcing and the observations. Many remotely sensed products exhibit gaps due to the presence of clouds or failures in the close sensing devices. As a general rule we do not recommend filling the gaps of the observed series. This filled data would exhibit a different error variance than the direct estimations from the remote sensors, introducing uncertainties which are not compatible with the data assimilation process. MuSA will internally handle the gaps

in the observations, provided that the netcdf files are generated using the regular conventions, assimilating just the timesteps where the information is available.

## 3 Data assimilation

Data assimilation (DA) is the exercise of fusing uncertain information from observations and (typically) geoscientific models (Evensen et al., 2022; Reich and Cotter, 2015). The DA schemes used in MuSA vary in their approach to state estimation as

well as in the underlying assumptions and algorithms employed. In this section we provide an overview of these schemes as well as the underlying theory.

### 3.1 Bayesian inference

In line with most modern approaches to DA (Wikle and Berliner, 2007; Carrassi et al., 2018; van Leeuwen et al., 2019) the assimilation schemes used in MuSA are built on the foundation of Bayesian inference (e.g. MacKay, 2003; Särkkä, 2013).

In this approach, the target quantity is the posterior distribution $p(\mathbf{x}|\mathbf{y})$, the probability of the model state $\mathbf{x} \in \mathbb{R}^m$ given the observations $\mathbf{y} \in \mathbb{R}^d$, which can be inferred using Bayes' rule

$$p(\mathbf{x}|\mathbf{y}) = \frac{p(\mathbf{y}|\mathbf{x})p(\mathbf{x})}{p(\mathbf{y})},$$

(2)

in which $p(\mathbf{y}|\mathbf{x})$, the probability of the observed data given a model state, is called the likelihood while $p(\mathbf{x})$, the probability of the state before considering the observations, is known as the prior. According to the likelihood principle, the likelihood



should be seen as a function of the model state, not the fixed observations. So it is implicitly understood that the state is as a vector of random variables, while the observation vector is deterministic. In this Bayesian interpretation, rather than being narrowly synonymous with frequency, probability is a broad measure of uncertainty. The evidence (or marginal likelihood) in the denominator of (2) is given by

$$p(\mathbf{y}) = \int p(\mathbf{x}, \mathbf{y}) \, \mathrm{d}\mathbf{x} = \int p(\mathbf{y}|\mathbf{x}) p(\mathbf{x}) \, \mathrm{d}\mathbf{x} \,, \qquad (3)$$

where the range of integration is understood to be over all possible values of $\mathbf{x}$, namely the support of the prior. This evidence is just a function of the observations, which are fixed, obtained by integrating the numerator (product of prior and the likelihood), so it simply serves as normalizing constant in this incarnation of Bayes' theorem which we can thus rewrite as

$$p(\mathbf{x}|\mathbf{y}) \propto p(\mathbf{y}|\mathbf{x}) p(\mathbf{x}) \,, \qquad (4)$$

to emphasize that the posterior is proportional to the product of the likelihood (what the data tells us) and the prior (what we
knew before considering the data). As we tend to deal with continuous variables in DA, the are usually probability density rather than mass functions. Since the posterior must integrate to one, we can safely ignore the evidence term as long as we are just dealing with the usual first level of inference where we fit a specific model. The evidence is nonetheless vital in the second level of inference where we compare models, but we do not pursue that here (see MacKay, 2003). Thus, we will henceforth use the notation $Z = p(\mathbf{y})$ to emphasize that the evidence simply serves as a normalizing constant in this overview.

To get the Bayesian engine to run and infer the posterior, we need to specify the two input distributions on the right hand side of (2), namely the prior and the likelihood. To construct the likelihood, we specify an observation (or equivalently: measurement, data generating, forward) model of the form

$$\mathbf{y} = \mathcal{H}(\mathbf{x}^\star) + \boldsymbol{\epsilon} \,, \qquad (5)$$

where $\mathbf{x}^\star$ contains the true values of the state variables, $\mathcal{H}(\cdot)$ is the observation operator which maps from state to observation
space, and $\boldsymbol{\epsilon}$ is the observation error. In practice the observation operator essentially picks out predicted observations, i.e. the state variables that correspond to observations, from the full state vector. As is commonly done in DA (Carrassi et al., 2018), we assume that the observation errors are additive, unbiased, and follow a Gaussian distribution, i.e. $\boldsymbol{\epsilon} \sim \mathrm{N}(\mathbf{0}, \mathbf{R})$ where $\mathbf{R}$ is the observation error covariance matrix. Thereby the likelihood becomes

$$p(\mathbf{y}|\mathbf{x}) = A \exp\left(-\frac{1}{2}\boldsymbol{\epsilon}^\mathrm{T} \mathbf{R}^{-1} \boldsymbol{\epsilon}\right) = A \exp\left(-\frac{1}{2}[\mathbf{y} - \widehat{\mathbf{y}}]^\mathrm{T} \mathbf{R}^{-1} [\mathbf{y} - \widehat{\mathbf{y}}]\right) \,, \qquad (6)$$

where $A = \det\left(2\pi\mathbf{R}\right)^{-1/2}$ is a normalizing constant, $(\cdot)^\mathrm{T}$ denotes the transpose, $\widehat{\mathbf{y}} = \mathcal{H}(\mathbf{x})$ are the predicted observations, and we have used that $\boldsymbol{\epsilon} = \mathbf{y} - \widehat{\mathbf{y}}$ conditional on $\mathbf{x}$ being true. Roughly speaking, the likelihood can be seen as a model misfit term which quantifies how likely the actual observations are given a particular model state. The states with a higher likelihood will correspond to those with predicted observations closer to the actual observations. The next ingredient is the prior distribution over states, $p(\mathbf{x})$, which can be specified based on initial beliefs which may include physical bounds, expert opinion, 'objective'
defaults (using e.g. maximum entropy), or knowledge from earlier analyses. For the latter, consider the case of assimilating





two conditionally independent observations $\mathbf{y}_1$ and $\mathbf{y}_2$. Then the posterior can be obtained using either batch or recursive estimation (Särkkä, 2013). The batch approach involves solving directly for

$$p(\mathbf{x}|\mathbf{y}_1,\mathbf{y}_2) \propto p(\mathbf{y}_1,\mathbf{y}_2|\mathbf{x})p(\mathbf{x}) = p(\mathbf{y}_2|\mathbf{x})p(\mathbf{y}_1|\mathbf{x})p(\mathbf{x}), \tag{7}$$

while for the recursive approach we note that

$$p(\mathbf{x}|\mathbf{y}_1) \propto p(\mathbf{y}_1|\mathbf{x})p(\mathbf{x}), \tag{8}$$

so we can split up the batch solution by first conditioning on $\mathbf{y}_1$ to get an intermediate posterior, then use this as a prior when updating with $\mathbf{y}_2$ to get the full posterior as follows

$$p(\mathbf{x}|\mathbf{y}_1,\mathbf{y}_2) \propto p(\mathbf{y}_2|\mathbf{x})p(\mathbf{x}|\mathbf{y}_1). \tag{9}$$

This recursive approach will give the same answer as the batch approach, but it can often be helpful to split the inference into smaller chunks especially for dynamic problems. We can extend recursive estimation to an arbitrary number of observations as long as they are conditionally independent.

In theory, evaluating the posterior simply involves taking the product of two terms. Naïvely, this suggests that we can estimate the posterior through a simple grid approximation. Unfortunately, inferring the posterior is usually akin to looking for a needle in a haystack due to the curse of dimensionality (Snyder et al., 2008) which manifests when we have a high dimensional state space and/or highly informative data. As such, we need to adopt efficient algorithms when doing DA in practice. Roughly speaking, the most widely used Bayesian DA schemes can be split into two major computational approaches: variational techniques and Monte Carlo methods (Carrassi et al., 2018). Despite success in operational meteorology (Bannister, 2017), the variational approach has received little attention in cryospheric applications (Dumont et al., 2012) since it involves gradient terms that are difficult to implement and has not yet shown any clear gains in performance over Monte Carlo. As such, in MuSA we restrict ourselves to Monte Carlo methods where the naming reflects that these approaches rely on using (pseudo)random numbers to sample from probability distributions (MacKay, 2003).

## 3.2 Prediction, filtering, and smoothing

The subset of Monte Carlo methods used in MuSA can be split into two classes: those derived from the EnKF (Evensen, 1994) and the PF (Gordon et al., 1993). These can be further subdivided into filters and smoothers depending on how time dynamics are handled (Carrassi et al., 2018). As such, it is instructive to appreciate the differences between stochastic prediction, filtering, and smoothing (see Jazwinski, 1970; Särkkä, 2013) as well as how these are implemented in MuSA. To do so, we introduce the following notation: we consider discrete points in time $t_k = t_0 + k\Delta t$ for $k = 0,\ldots,n$ where $t_0$ is the initial time and $\Delta t$ is the model timestep and we let $\mathbf{x}_k = \mathbf{x}(t_k)$ denote the state at a given time. We employ analogous notation for the observations $\mathbf{y}_k$, but (without loss of generality) we assume that the initial time is unobserved. Note that in practice we usually do not have observations at every model time step, we have just assumed this here to simplify notation. The extension of filters and





smoothers to the more realistic case with sparse observations in time is straightforward and merely requires a slight change in indexing so that filtering and smoothing only takes place using predicted observations at observation times.

In DA, prediction involves estimating the future state given past observations: $p(\mathbf{x}_k|\mathbf{y}_{1:\ell})$ where $\mathbf{y} = [\mathbf{y}_1, \ldots, \mathbf{y}_\ell]$ and $\ell < k$. The archetypal example of this is the familiar task of weather forecasting through ensemble-based numerical weather

prediction (Bauer et al., 2015). In the context of prediction, the concept of future and past are just relative to one another and not necessarily real time. So one can also do predictions of the past, as long as any observations considered are from the even more distant past. Prediction also enters as a step in both filtering and smoothing, so it is helpful to understand how to implement it probabilistically. The prediction step from timestep $k-1$ to $k$ can be formulated as follows:

$$\mathbf{x}_k = \mathcal{M}(\mathbf{x}_{k-1}) + \boldsymbol{\eta}_{k-1}, \tag{10}$$

where $\mathcal{M}(\cdot)$ is the dynamical model (FSM2 in this case) while $\boldsymbol{\eta} \sim \mathrm{N}(\mathbf{0}, \mathbf{Q})$ is the additive model error (process noise) term which we assume to be independent in time and follow a zero-mean Gaussian distribution with covariance matrix $\mathbf{Q}$. These assumptions can be relaxed without loss of generality, but their convenience and broad justifiability mean that they are often employed in practice (Carrassi et al., 2018). Crucially, the above prediction step produces Markovian (memoryless) dynamics where the current state depends only on the previous state and white noise. This Markov property is crucial in making

the filtering and smoothing problems tractable. It implies $p(\mathbf{x}_k|\mathbf{x}_{0:(k-1)}) = p(\mathbf{x}_k|\mathbf{x}_{k-1})$ which lets us factorize and simplify distributions such as the full prior as follows

$$p(\mathbf{x}_{0:k}) = p(\mathbf{x}_k|\mathbf{x}_{0:(k-1)})p(\mathbf{x}_{0:(k-1)}) = p(\mathbf{x}_k|\mathbf{x}_{k-1})p(\mathbf{x}_{0:(k-1)}), \tag{11}$$

where the transition density is Gaussian of the form $p(\mathbf{x}_k|\mathbf{x}_{k-1}) = \mathrm{N}(\mathbf{x}_k|\mathbf{x}_{k-1}, \mathbf{Q})$. Applying this recursively we obtain

$$p(\mathbf{x}_{0:k}) = p(\mathbf{x}_k|\mathbf{x}_{k-1})p(\mathbf{x}_{k-1}|\mathbf{x}_{k-2})\ldots p(\mathbf{x}_1|\mathbf{x}_0)p(\mathbf{x}_0) = p(\mathbf{x}_0)\prod_{j=1}^{k} p(\mathbf{x}_j|\mathbf{x}_{j-1}), \tag{12}$$

Using this kind of factorization together with marginalization also helps us construct the (marginal) predictive distribution $p(\mathbf{x}_k|\mathbf{y}_{1:\ell})$ where $\ell < k$ as follows

$$p(\mathbf{x}_k|\mathbf{y}_{1:\ell}) = \int p(\mathbf{x}_k, \mathbf{x}_{k-1}|\mathbf{y}_{1:\ell})\, \mathrm{d}\mathbf{x}_{k-1} = \int p(\mathbf{x}_k|\mathbf{x}_{k-1})p(\mathbf{x}_{k-1}|\mathbf{y}_{1:\ell})\, \mathrm{d}\mathbf{x}_{k-1}. \tag{13}$$

This is the Chapman-Kolmogorov equation (Särkkä, 2013) which can be applied recursively to obtain the predictive distribution at the current time step using the transition density together with previous predictive distributions.

From prediction we move to filtering which involves estimating the current state given current and past observations: $p(\mathbf{x}_k|\mathbf{y}_{1:k})$. This is the problem solved by sequential DA where an archetypal example is the initialization of numerical weather predictions as new observations become available to delay the effects of chaos (Bauer et al., 2015). To construct the filtering distribution we first re-introduce our Gaussian observation model in the dynamical context where we make the usual assumption that the current observations are conditionally independent of both the observation and state histories (Särkkä,

2013) resulting in the dynamic likelihood

$$p(\mathbf{y}_k|\mathbf{x}_{0:k}, \mathbf{y}_{1:(k-1)}) = p(\mathbf{y}_k|\mathbf{x}_k) = A_k \exp\left(-\frac{1}{2}\left[\mathbf{y}_k - \widehat{\mathbf{y}}_k\right]^{\mathrm{T}} \mathbf{R}_k^{-1} \left[\mathbf{y}_k - \widehat{\mathbf{y}}_k\right]\right), \tag{14}$$





where $\widehat{\mathbf{y}}_k = \mathcal{H}(\mathbf{x}_k)$ are the predicted observations at the current timestep and we have added a time index to the normalizing constant ($A_k$) and the observation error covariance matrix ($\mathbf{R}$), respectively, to emphasize that both the number and types of observations at a given point in time may vary. Combining Markovian state dynamics with a conditionally independent

observation model means that we end up with a state-space or hidden Markov model (Cappé et al., 2005) where the states at each timestep are hidden (or latent) because they are not observable due to measurement error. The filtering distribution can now be obtained by combining the predictive distribution for $\ell = k-1$ which serves as the prior and the dynamic likelihood through Bayes' theorem as follows

$$p(\mathbf{x}_k|\mathbf{y}_{1:k}) \propto p(\mathbf{y}_k|\mathbf{x}_k)p(\mathbf{x}_k|\mathbf{y}_{1:(k-1)}). \tag{15}$$

As such, prediction and filtering can be applied one after the other in time to sequentially obtain the filtering distributions of interest for all integration time-steps $k = 1, \ldots, n$ . That is, starting from the initial prior $p(\mathbf{x}_0)$ we run the dynamical model to $k = 1$ to get the transition density in the product $p(\mathbf{x}_1|\mathbf{x}_0)p(\mathbf{x}_0)$ and marginalize to obtain the predictive distribution $p(\mathbf{x}_1)$ which we use as the prior when assimilating the observations $\mathbf{y}_1$ to estimate the filtering distribution $p(\mathbf{x}_1|\mathbf{y}_1)$ . We continue this cycle, running the dynamical model to $k = 2$ to obtain $p(\mathbf{x}_2|\mathbf{x}_1)p(\mathbf{x}_1|\mathbf{y}_1)$ and marginalize to obtain $p(\mathbf{x}_2|\mathbf{y}_1)$

which we use as the prior when assimilating $\mathbf{y}_2$ to arrive at $p(\mathbf{x}_2|\mathbf{y}_{1:2})$ . This filtering process of "online" assimilation as observations become available sequentially in time is appealing for operational forecasting since it can continue indefinitely with low memory requirements while outputting the filtering and prediction distributions of interest. In practice when using Monte Carlo methods, we do not operate with the distributions themselves but rather with an ensemble (i.e. a collection) of samples from these. Typically, due to the often prohibitively large size of a full spatio-temporal ensemble, one would only store

summary statistics such as the posterior mean and standard deviation of the state for each point in space and time as the outputs of the analysis.

In addition to filtering, we are also able to solve some smoothing problems in MuSA. The key difference between filtering and smoothing is that the later considers future observations, i.e. the (marginal) smoothing distribution is $p(\mathbf{x}_k|\mathbf{y}_{1:\ell}$ where $\ell > k$ . Several types of smoothing problems exist (Cosme et al., 2012; Särkkä, 2013), namely: fixed-lag smoothing ($\ell = k + l$

with $l$ constant) which is equivalent to filtering but where the state is lagged relative to the observations, fixed-point smoothing ($k$ is fixed) where the posterior at a fixed point in time (such as an initial condition) is conditioned on both past and future observations, and fixed-interval smoothing ($\ell$ is fixed) where we estimate the posterior for points in a time interval given all observations in that interval. In MuSA we focus on a form of the fixed-interval smoothing known as batch smoothing in which the states within the time interval, henceforth called the data assimilation window (DAW), are updated in a single batch using

all observations such that $\ell = n$.

The batch smoothing approach has been shown to be especially well suited for reanalysis type problems in land and snow data assimilation since it allows for backward propagation of information (Dunne and Entekhabi, 2005; Durand et al., 2008). We will restrict our attention to strong-constraint batch smoothing with indirect updates (Evensen, 2019; Evensen et al., 2022) where it is assumed that the dynamical model is perfect (i.e., no model error $\boldsymbol{\eta}$) and that all uncertainty stems from parameters

that are constant within the DAW. This approach is adopted both for simplicity as well as the fact that it ensures that the





updated states are dynamically consistent with the physics in the model. It also has the advantage of implicitly ensuring that state variables such as snow depth or SWE remain within their physically non-negative bounds. In this approach, an ensemble of realizations of the model are run forward for an entire DAW during which all the observations and the corresponding predicted observations from the model are stored. At the end of the DAW, these observations are all assimilated simultaneously
in a batch update to provide samples from the posterior parameter distribution. For seasonal snow the water year becomes a natural choice for the DAW. With this choice, potentially highly informative observations made during the ablation season are able to inform the preceding accumulation season (Margulis et al., 2015). This backwards propagation of information is crucial to help reconstruct peak SWE which is usually of particular interest to snow hydrologists (Dozier et al., 2016).

### 3.3   Consistency

A potential problem in DA is that the stochastic perturbations and updates that are imposed on the model and the resulting dynamics may lead to inconsistent model states. We can define (at least) two different types of inconsistencies: (i) dynamical inconsistency and (ii) physical inconsistency where model states and/or parameters violate their physical bounds.

Dynamical inconsistency with respect to the underlying deterministic model can enter into the DA exercise either through stochastic model error terms in prediction steps or through the assimilation step itself. In the case of weak constraint DA,
where imperfections in the model are represented by stochastic model error terms , this kind of weak inconsistency exists by design and is a feature rather than a bug. In particular, the dynamical inconsistency created by the stochastic terms is meant to explicitly account for model uncertainty arising for example due to unresolved processes (Palmer, 2019). As such, the open-loop (no DA) stochastic model dynamics will be inconsistent with a deterministic version of the model. Arguably calling such a stochastic model dynamically inconsistent is somewhat of a misnomer, since it aspires to be more consistent with reality than
a purely deterministic version of the same model.

The assimilation step itself can also introduce dynamical inconsistencies, particularly when applying filtering algorithms (Dunne and Entekhabi, 2005). For the EnKF these manifest as sawtooth-like patterns (jumps) in the dynamics at each assimilation step due to the fact that the Kalman-based analysis actually moves the ensemble from being samples of the prior to being (approximate) samples from the posterior. For the PF dynamical inconsistency also enters at the assimilation step when-
ever resampling is performed. The resampling step tends to kill off unpromising low weight particles while reproducing more promising higher weight particles. Since this evolutionary step is implemented sequentially in time when filtering, it will also introduce discontinuities in the dynamics of the model ensemble. In the case of both the EnKF and the PF, the discontinuity introduced at the assimilation step is an expected result of filtering. If one wishes to avoid this behavior, one should instead turn to smoothing algorithms.

Physical inconsistency can also enter both through stochastic perturbations and the assimilation step itself. Many snowpack state variables (and potential observables) have a relatively limited dynamic range and are either low bounded, e.g. snow depth and SWE are non-negative, or double bounded, e.g. snow albedo and FSCA are physically confined to the range $[0, 1]$. This means that directly applying assimilation routines which assume unbounded (e.g. Gaussian) distributions on the prior, likelihood, or the model error can lead to physical inconsistencies in snow DA.





A simple but naive solution is to just explicitly enforce state variables to lie within their physical bounds using minimum and/or maximum functions. Unfortunately, this is the same as truncating the underlying probability distributions and may degrade the performance of the assimilation scheme due to a strong violation of assumptions. Although this is a major concern for the EnKF-based schemes, which make explicit assumptions about Gaussianity, in practice it may also impact the PF where Gaussian distributions remain a convenient choice. A better way to deal with issues around physical inconsistency is to apply transformations to bounded random variables. Typically, these transformations will map such random variables from the bounded physical space to an unbounded space that can accommodate Gaussian distributions through Gaussian anamorphosis techniques (Bertino et al., 2003)). This can be carried out using transforms that have been empirically constructed or using analytic functions. In MuSA, following (Aalstad et al., 2018), we employ analytical Gaussian anamorphosis where we use a logarithm transform for variables that are physically lower bounded and a logit transform for variables that are physically double bounded. The corresponding inverse functions, i.e. the exponential and logistic transforms, can be used to map these unbounded variables back to the bounded physical space. For bounded variables we will often construct the prior distributions using these transforms. For example, assigning a lognormal prior can ensure that a variable never exceeds its lower bounds. In such a case the assimilation step takes place in the unbounded log transformed space where this variable is normally (i.e. Gaussian) distributed while the model integration takes place in the bounded physical space, which can be recovered using the exponential transform, where the variable is lognormally (i.e. log-Gaussian) distributed.

A key step taken in MuSA to reduce inconsistencies in snow data assimilation is to split the state vector in two: $\mathbf{x} = [\mathbf{uv}]$ where $\mathbf{u} \in \mathbb{R}^{m_p}$ are parameters while $\mathbf{v} \in \mathbb{R}^{m_s}$ are internal model states. Only the parameters, which can include both internal model parameters and forcing perturbation parameters, are assumed to be random variables in MuSA. This approach corresponds to the so-called forcing formulation (as opposed to the model-state formulation) of the DA problem (Evensen et al., 2022). The internal model states are taken to be deterministic given these parameters. Such a setup is tantamount to assuming that all the uncertainty in the model dynamics stems from the forcing data and internal parameters. This assumption is largely in line with previous findings (Raleigh et al., 2015) particularly for intermediate complexity snow models such as FSM2 (Günther et al., 2019). This is gradually becoming a relatively standard setup in snow data assimilation experiments (Margulis et al., 2015; Magnusson et al., 2017; Aalstad et al., 2018; Alonso-González et al., 2021) although it has arguably not been as properly formalized as elsewhere in the DA literature (Evensen, 2019; Evensen et al., 2022). Through such a split we ensure that internal model states remain both dynamically and physically consistent given the parameters. The parameters, in turn, are made physically consistent by applying analytical Gaussian anamorphosis transforms in the DA scheme. As such, we will let denote anamorphosed parameters that have undergone a forward transform to the unbounded space. It is implicitly assumed that these are inverse transformed to the physical space before they are passed to the model to help evolve the internal states forward in time.

In the filtering algorithms in MuSA, both the stochastic parameters and the conditionally deterministic internal states are dynamic. The parameter dynamics evolve according to simple jitter (Farchi and Bocquet, 2018) of the form:

$$\mathbf{u}_k = \mathbf{u}_{k-1} + \boldsymbol{\eta}_{k-1}, \tag{16}$$





where we recall that $\boldsymbol{\eta}_{k-1} \sim \mathrm{N}(\mathbf{0}, \mathbf{Q})$ and that the parameters are defined in the unbounded transformed space. Probabilisti-
cally, this corresponds to the Markovian transition density $p(\mathbf{u}_k|\mathbf{u}_{k-1}) = \mathrm{N}(\mathbf{u}_k|\mathbf{u}_{k-1}, \mathbf{Q})$. The internal state variables, on the other hand, evolve according to the full dynamical model ( SM2) which also depends on the dynamic parameters $\mathbf{u}_k$ through $\mathbf{v}_k = \mathcal{M}(\mathbf{u}_k, \mathbf{v}_{k-1})$. We reiterate that it is implicitly understood that these parameters are transformed back to physical space when they are used in the dynamical model. These dynamics corresponds to the transition density (e.g. Evensen, 2018)

$$p(\mathbf{v}_k|\mathbf{u}_k, \mathbf{v}_{k-1}) = \delta\left(\mathbf{v}_k - \mathcal{M}(\mathbf{u}_k, \mathbf{v}_{k-1})\right), \tag{17}$$

where $\delta(\cdot)$ is the Dirac delta function which emphasizes that the internal state is not only Markovian but also conditionally deterministic given the parameters. This just formalizes that the only uncertainty in the internal state dynamics stems from the parameters to which we assign an initial prior $p(\mathbf{u}_0)$. When dealing with seasonal snow, the initial prior for the internal states, $p(\mathbf{v}_0)$, is known $\mathbf{v}_0 = \boldsymbol{\nu}_0$ and thus deterministic so $p(\mathbf{v}_0) = \delta(\mathbf{v}_0 - \boldsymbol{\nu}_0)$ since the annual integration period starts at the beginning of the water year where we assume that a snowpack has not yet formed so internal state variables are either $0$ (snow
depth, SWE) or undefined (e.g. snow surface temperature). Together with the dynamic likelihood $p(\mathbf{y}_k|\mathbf{u}_k)$, these distributions can be used to construct the target marginal filtering distribution $p(\mathbf{u}_k|\mathbf{y}_{1:k})$ which we can estimate with ensemble Kalman or particle filtering algorithms. By combining this marginal with appropriate densities we can also estimate the joint filtering distribution $p(\mathbf{u}_k, \mathbf{v}_k|\mathbf{y}_{1:k})$ from which we can compute posterior expectations for the internal states of interest. The derivation of this distribution is relatively analogous to that for the smoother below, but involves more steps and is thus not included
herein. In practice, (approximate) samples from this joint filtering distribution are obtained from the posterior ensemble after each observation time and the subsequent assimilation step while running the filtering schemes in MuSA.

For the smoothing algorithms in MuSA, we instead assume that the dynamical model is perfect so the components of $\mathbf{u}$ are constant (i.e. time-invariant) but uncertain parameters, which can be either internal parameters or related to the forcing, while $\mathbf{v}_{0:n} = [\mathbf{v}_0, \dots, \mathbf{v}_k, \dots, \mathbf{v}_n]$ remain the dynamic internal state variables with initial state $\mathbf{v}_0$ at $t_0$ and final state $\mathbf{v}_n$ at $t_n$
which is the end of what is now an annual DAW. The prediction step with the assumed perfect dynamical model then becomes $\mathbf{v}_k = \mathcal{M}(\mathbf{u}, \mathbf{v}_{k-1})$ which implies the following transition density:

$$p(\mathbf{v}_k|\mathbf{v}_{k-1}, \mathbf{u}) = \delta\left(\mathbf{v}_k - \mathcal{M}(\mathbf{u}, \mathbf{v}_{k-1})\right). \tag{18}$$

This is just a formal way of denoting that the only uncertainty in the dynamics stems from time invariant but uncertain parameters to which we assign a prior. Thereby, the full prior for the entire DAW can be factorized as follows

$$p(\mathbf{x}) = p(\mathbf{u}, \mathbf{v}_{0:n}) = p(\mathbf{v}_{0:n}|\mathbf{u})p(\mathbf{u}) = p(\mathbf{u})p(\mathbf{v}_0)\prod_{k=1}^{n} p(\mathbf{v}_k|\mathbf{v}_{k-1}, \mathbf{u}), \tag{19}$$

where as before $p(\mathbf{v}_0) = \delta(\mathbf{v}_0 - \boldsymbol{\nu}_0)$ is the prior for the initial condition of the state. The full posterior then becomes

$$p(\mathbf{x}|\mathbf{y}_{1:n}) = p(\mathbf{u}, \mathbf{v}_{0:n}|\mathbf{y}_{1:n}) \propto p(\mathbf{y}_{1:n}|\mathbf{u})p(\mathbf{u}, \mathbf{v}_{0:n}), \tag{20}$$

which can be marginalized to obtain the marginal posterior for the parameters

$$p(\mathbf{u}|\mathbf{y}_{1:n}) \propto \int p(\mathbf{y}_{1:n}|\mathbf{u})p(\mathbf{u}, \mathbf{v}_{0:n})\,\mathrm{d}\mathbf{v}_{0:n} = p(\mathbf{y}_{1:n}|\mathbf{u})p(\mathbf{u}), \tag{21}$$





where the batch likelihood is

$$p(\mathbf{y}_{1:n}|\mathbf{u}) = A_{1:n}\exp\left(-\frac{1}{2}[\mathbf{y}_{1:n} - \widehat{\mathbf{y}}_{1:n}]^{\mathrm{T}}\mathbf{R}_{1:n}^{-1}[\mathbf{y}_{1:n} - \widehat{\mathbf{y}}_{1:n}]\right),\tag{22}$$

in which $\widehat{\mathbf{y}}_{1:n} = [\widehat{\mathbf{y}}_1,\ldots,\widehat{\mathbf{y}}_k,\ldots,\widehat{\mathbf{y}}_n]$ contains the predicted observations $\widehat{\mathbf{y}}_k = \mathcal{H}(\mathbf{u},\mathbf{v}_k)$ for all observation time steps, while $A_{1:n} = \det(2\pi\mathbf{R}_{1:n})^{-1/2}$ where $\mathbf{R}_{1:n} = \mathrm{diag}(\mathbf{R}_1,\ldots,\mathbf{R}_k,\ldots,\mathbf{R}_n)$ is the batch observation error covariance matrix which is a block diagonal matrix containing the observation error covariance matrices for all observation time steps. Note that the

state $\mathbf{v}_{0:n}$ is conditionally deterministic given $\mathbf{u}$, it is not explicitly included in the batch likelihood. By combining the batch likelihood with the prior for the parameters, we have all that we need to estimate the posterior for the parameters $p(\mathbf{u}|\mathbf{y}_{1:n}) \propto p(\mathbf{y}_{1:n}|\mathbf{u})p(\mathbf{u})$. The main difference to the filtering distribution is that all the data are assimilated in a single batch to update static parameters. Once we have obtained the posterior for the parameters, due to the strong constraint, we recall that we can easily recover the joint posterior for the states and parameters $p(\mathbf{v}_{0:n},\mathbf{u}|\mathbf{y}_{1:n}) \propto p(\mathbf{v}_{0:n}|\mathbf{u})p(\mathbf{u}|\mathbf{y}_{1:n})$ from which we can

compute posterior expectations for the internal states of interest. In practice, the steps involved in estimating the posterior and subsequent expectations depend on which batch smoothing algorithm is used.

### 3.4 Particle methods

Particle methods (Särkkä, 2013), also known as Sequential Monte Carlo (Cappé et al., 2005; Chopin and Papaspiliopoulos, 2020), are appealing since they impose few assumptions, can be relatively easy to derive, and are simple to implement in their

basic form. In principle, they can be used to solve any Bayesian inference problem, including the aforementioned filtering and smoothing problems. The so-called bootstrap filter described by Gordon et al. (1993) arguably marks the introduction of particle filters to the wider scientific community. Nonetheless, the underlying idea of sequential importance resampling (SIR) was already well known in statistics literature from which it emerged (see Smith and Gelfand, 1992, and references therein). For a more up to date perspective, van Leeuwen et al. (2019) provide a comprehensive review of particle methods for geoscientific

data assimilation, including a discussion of the state-of-the-art and promising avenues for further developments.

Particle methods have rapid become the most widely adopted approach to data assimilation within the snow science community. The PF, in particular, has become increasingly popular in snow data assimilation studies. This began with the seminal work of Leisenring and Moradkhani (2011) and Dechant and Moradkhani (2011) showing how the PF can outperform the EnKF in snow data assimilation applications, assimilating SNOTEL SWE and passive microwave data, at least if relatively

simple models and a large ensemble is used. Charrois et al. (2016) subsequently applied the PF in synthetic (twin) experiments that assimilated MODIS-like reflectance data into the Crocus snow model for a site in the French Alps. Magnusson et al. (2017) showed how snow depth assimilation using the PF improved the estimation of several variables, including SWE and snowmelt runoff, in an intermediate complexity snow model for many sites across the Alps. The study of Baba et al. (2018) applied the PF to assimilate novel SCA satellite retrievals from Sentinel-2, demonstrating marked improvements in snowpack simulations

across the sparsely instrumented High-Atlas mountains of Morocco. Piazzi et al. (2018) investigated the potential of performing a joint (i.e., multivariate) assimilation of several snowpack variables obtained from ground-based observations at 3 sites in the Alps using the PF, noting that degeneracy was more likely to occur for this setup than in more typical uni-variate snow data





assimilation experiments. Smyth et al. (2019) assimilated monthly snow depth observations into an intermediate complexity
snow model with the PF at a well instrumented site in the California Sierra Nevada, improving both snow density and SWE

estimates with promising implications for the assimilation of snapshots of snow depth retrieved from airborne and space-borne
platforms. These results were confirmed by Deschamps-Berger et al. (2022) who showed that the assimilation of a single snow
depth map per season is sufficient to improve the simulated spatial variability of the snowpack. Recent efforts, such as the
synthetic study of Cluzet et al. (2021), have focused on the challenge of spatially propagating snowpack information from
observed to unobserved locations using the PF.

Another line of studies has adopted particle smoothing schemes, particularly the particle batch smoother (PBS Margulis et al.,
2015), for snow reanalysis. Here a batch of remotely sensed, typically FSCA, observations are assimilated simultaneously to
weight (assign probability mass to) an ensemble of model trajectories for the entire water year. Margulis et al. (2016) used
the PBS to perform a 90 meter resolution snow reanalysis for the California Sierra Nevada covering the entire Landsat-era
from 1985 to 2015. Cortés and Margulis (2017) applied a similar setup to conduct a snow reanalysis for the extratropical

Andes. Aalstad et al. (2018) compared the performance of the PBS with ensemble Kalman-based smoothers when assimilating
FSCA data from Sentinel-2 and MODIS for sites in the high-Arctic and found that the PBS markedly outperformed non-iteratve
ensemble Kalman-based smoothers in line with Margulis et al. (2015). Baldo and Margulis (2018) developed a multi-resolution
snow reanalysis framework using the PBS and demonstrated, through tests in a basin in Colorado, that this could match the
performance of the original single-resolution approach at a fraction of the computational cost. In DA experiments in the Swiss

Alps, Fiddes et al. (2019) showed how clustering offers an alternative promising avenue for speeding up snow DA with the
PBS, allowing for hyper-resolution reanalyses. Alonso-González et al. (2018) carried out a snow reanalysis over the sparsely
instrumented Lebanese mountains by forcing FSM with quasi-dynamically downscaled meteorological reanalysis data and
subsequently assimilating MODIS-based FSCA retrievals with the PBS. Liu et al. (2021) recently performed an 18-year 500
m resolution snow reanalysis for High Mountain Asia by using the PBS to jointly assimilate MODIS and Landsat FSCA data.

Although all of these studies have focused on using FSCA data for snow reanalysis, other remotely sensed retrievals could also
be considered. For example, the work of Margulis et al. (2019) has shown that the PBS can also be used to assimilate infrequent
lidar-based snow depth retrievals with marked improvements in the estimation of snowpack-related variables. The next logical
step would thus be to jointly assimilate FSCA and emerging satellite-based snow depth retrievals Marti et al. (2016); Treichler
and Kääb (2017); Lievens et al. (2019).

Importance sampling is key to understanding these particle methods. Despite what the name may suggest, this is actually
not an approach to directly generate samples from a target distribution of interest. Instead, it allows us to estimate expectations
of functions with respect to a target distribution by drawing from another distribution that it is easier to sample from (MacKay,
2003). In DA, the posterior is the target distribution of interest and the expectation of some function $g(\mathbf{x})$ with respect to the
posterior is defined as follows (Särkkä, 2013)

$$\mathrm{E}\left[g(\mathbf{x})|\mathbf{y}\right] = \int g(\mathbf{x})p(\mathbf{x}|\mathbf{y})\,\mathrm{d}\mathbf{x}, \tag{23}$$





where the expectation of $g(\mathbf{x}) = \mathbf{x}$ yields the posterior mean $\widehat{\boldsymbol{\mu}}$ while the expectation of $g(\mathbf{x}) = (\mathbf{x} - \widehat{\boldsymbol{\mu}})^2$ yields the posterior variance $\boldsymbol{\sigma}^2$. If we could generate $N$ independent samples from the posterior, $\mathbf{x}^{(i)} \sim p(\mathbf{x}|\mathbf{y})$, we could estimate the expectation in (23) numerically using direct Monte Carlo integration as follows

$$\mathrm{E}[g(\mathbf{x})|\mathbf{y}] \simeq \frac{1}{N} \sum_{i=1}^{N} g\left(\mathbf{x}^{(i)}\right), \tag{24}$$

where the $\mathbf{x}^{(i)}$ with $i = 1, \ldots, N$ denote $N$ independent samples from the posterior. Thanks to the law of large numbers and the central limit theorem, we know that this approximation will converege almost surely to the true expectation as $N \to \infty$ with a standard error inversely proportional to $\sqrt{N}$ (Chopin and Papaspiliopoulos, 2020). Unfortunately, we are rarely able to generate independent samples directly from the posterior.

In importance sampling, we instead use a proposal (or importance) distribution $q(\mathbf{x})$ that we know how to sample from
which must have at least the same support as the posterior (i.e. $q(\mathbf{x}) > 0$ wherever $p(\mathbf{x}|\mathbf{y}) > 0$). Then, by multiplying the integrand with $1 = q(\mathbf{x})/q(\mathbf{x})$, we can solve (23) using importance sampling-based Monte Carlo integration

$$\mathrm{E}[g(\mathbf{x})|\mathbf{y}] = \int g(\mathbf{x}) \frac{p(\mathbf{x}|\mathbf{y})}{q(\mathbf{x})} q(\mathbf{x}) \, \mathrm{d}\mathbf{x} \simeq \frac{1}{N} \sum_{i=1}^{N} g(\mathbf{x}^{(i)}) \frac{p(\mathbf{x}^{(i)})}{q(\mathbf{x}^{(i)})} = \frac{1}{N} \sum_{i=1}^{N} g(\mathbf{x}^{(i)}) \widehat{w}(\mathbf{x}^{(i)}), \tag{25}$$

where the $\mathbf{x}^{(i)}$ are now samples from the proposal and we have defined the normalized weights $\widehat{w}(\mathbf{x}) = p(\mathbf{x}|\mathbf{y})/q(\mathbf{x})$. Following the nomenclature of Chopin and Papaspiliopoulos (2020), these weights are normalized in the sense that their expectation with
respect to $q(\mathbf{x})$ equals 1. hurdle remains in that we only know the posterior up to an unknown normalizing constant (i.e., the evidence $Z$) so in practice we can only evaluate the un-normalized posterior. Recalling the definition of the evidence in (3), we can nonetheless use importance sampling to approximate it as follows

$$Z = \int f(\mathbf{x}) \, \mathrm{d}\mathbf{x} = \int \frac{f(\mathbf{x})}{q(\mathbf{x})} q(\mathbf{x}) \, \mathrm{d}\mathbf{x} \simeq \frac{1}{N} \sum_{i=1}^{N} \widetilde{w}(\mathbf{x}^{(i)}), \tag{26}$$

where $f(\mathbf{x}) = p(\mathbf{y}|\mathbf{x})p(\mathbf{x})$ is the un-normalized posterior and we have defined the un-normalized weights $\widetilde{w}(\mathbf{x}) = f(\mathbf{x})/q(\mathbf{x})$
which no longer have an expectation of 1 with respect to $q(\mathbf{x})$. Using the samples $\mathbf{x}^{(i)} \sim q(\mathbf{x})$, this evidence approximation, and that $p(\mathbf{x}|\mathbf{y}) = f(\mathbf{x})/Z$, we can now solve (23) as follows

$$\mathrm{E}[g(\mathbf{x})|\mathbf{y}] = \frac{1}{Z} \int \frac{f(\mathbf{x})}{q(\mathbf{x})} q(\mathbf{x}) \, \mathrm{d}\mathbf{x} \simeq \sum_{i=1}^{N} g(\mathbf{x}^{(i)}) w(\mathbf{x}^{(i)}), \tag{27}$$

where, letting $w^{(i)} = w(\mathbf{x}^{(i)})$ for economy, the *auto*-normalized weights are given by $w^{(i)} = \widetilde{w}^{(i)} \left( \sum_{j=1}^{N} \widetilde{w}^{(j)} \right)^{-1}$ with the property that $\sum_{i=1}^{N} w^{(i)} = 1$. Note that with these auto-normalized weights $q(\mathbf{x})$ also only needs to be known up to a nor-
malizing constant $Z_q$ since any such constants cancel out in the auto-normalization step. Furthermore, direct Monte Carlo integration can be seen as a special case of importance sampling where the proposal is the target distribution itself which leads to uniformly equal weights with a value of $1/N$.

Mathematically, importance sampling is tantamount to a particle representation of the posterior through a sum of weighted Dirac delta functions centered on the sampled states $\mathbf{x}^{(i)} \sim q(\mathbf{x})$ of the form $p(\mathbf{x}|\mathbf{y}) \simeq \sum_{i=1}^{N} w^{(i)} \delta(\mathbf{x} - \mathbf{x}^{(i)})$ (Särkkä, 2013).





We can see this by recalling that the Dirac delta has the properties that $\int \delta(\mathbf{x}-\mathbf{x}^{(i)})\,\mathrm{d}\mathbf{x} = 1$ and $\int g(\mathbf{x})\delta(\mathbf{x}-\mathbf{x}^{(i)})\,\mathrm{d}\mathbf{x} = g\left(\mathbf{x}^{(i)}\right)$, such that by inserting the particle representation in (23) we have

$$\mathrm{E}\left[g(\mathbf{x})|\mathbf{y}\right] \simeq \int \sum_{i=1}^{N} g(\mathbf{x})w^{(i)}\delta\left(\mathbf{x}-\mathbf{x}^{(i)}\right)\,\mathrm{d}\mathbf{x} = \sum_{i=1}^{N} g(\mathbf{x}^{(i)})w^{(i)}, \tag{28}$$

which is equal to the result in (27). This particle representation is helpful since we can conceptualize distributions as consisting of a set of particles (or points) in state space whose probability mass are given by their weights.

The catch with importance sampling is that, unless the proposal is nearly identical to the target distribution, all the probability mass tends to collapse onto just a few particles as the dimensions of a problem increase (MacKay, 2003). This is the so-called degeneracy problem of particle methods which is closely tied to the aforementioned curse of dimensionality (Farchi and Bocquet, 2018; Snyder et al., 2008). One way to partly circumvent degeneracy in a sequential setting is to employ resampling techniques (see Li et al., 2015) where a new set of equally weighted particles is drawn based on the weights (i.e., probability

masses) of the existing particles. Effectively, resampling tends to reproduce particles with a higher weight while removing particles with lower weight. Resampling thus provides obvious links between particle methods and more heuristic genetic algorithms since the weights can be interpreted as a kind of "fitness" (Chopin and Papaspiliopoulos, 2020). The standard metric (e.g. Särkkä, 2013) for monitoring symptoms of degeneracy is the effective sample size

$$N_{\mathrm{eff}} = \frac{1}{\sum_{i=1}^{N}\left(w^{(i)}\right)^2}, \tag{29}$$

where a healthy ensemble of particles would have $N_{\mathrm{eff}} = N$ while a completely degenerated ensemble has $N_{\mathrm{eff}} = 1$. This metric can be used to adaptively determine the need for resampling based on a requirement that the ratio $N_{\mathrm{eff}}/N$ stays above some "healthy" threshold. Although resampling ensures that weight is more evenly spread among the particles, the occurrence of several identical particles results in sample impoverishment which can in the worst case also lead to a degenerate representation of the posterior. Particle diversity can nonetheless often be rejuvenated implicitly through stochastic terms (jitter) in the

dynamical model or more explicitly with a few iterations of Markov Chain Monte Carlo (MCMC; Gilks and Berzuini, 2001).

Having explained importance resampling, the final step is to tie this together with the sequential aspect of particle methods. In the context of both particle filtering and smoothing, importance resampling can be applied sequentially in time as observations become available to the state space model. For the particle filter, this occurs by first drawing particles from the initial prior $\mathbf{x}_0^{(i)} \sim p(\mathbf{x}_0)$, then for $k = 1, \ldots, n$ perform SIR:

1. Propagate the particles forward in time through the dynamical model from time $k-1$ to $k$ where new observations are available.

   2. Calculate the auto-normalized weights $w_k^{(i)}$ given the current observations.

   3. Resample the particles, possibly only if the ratio $N_{\mathrm{eff}}/N$ is below some threshold.

The recipes for implementing most flavors of particle smoothers tend to be a little bit more involved (Särkkä, 2013). Fortu-

nately, the particle batch smoother (PBS) in MuSA is relatively straightforward since in practice it is equivalent to standard





sequential importance sampling (SIS) which is just SIR without the resampling step. As such, SIS is quite prone to degeneracy. Nonetheless, an advantage with SIS is that, due to the absence of resampling, the dynamical state history of each particle will be completely consistent with the dynamical model. For the PBS we are only interested in the final (i.e. at time $t_n$) weights $w_n^{(i)}$ which, when attached to the full state histories, give us an approximation to the full posterior $p(\mathbf{x}_{0:n}|\mathbf{y}_{1:n})$ rather than the

filtering distribution $p(\mathbf{x}_k|\mathbf{y}_{1:k})$. This is advantageous in snow data assimilation since it allows information from observations during the ablation season to propagate backwards in time and influence states in the preceding accumulation season. The final weights can either be computed sequentially using SIS or in a batch update, since these are equivalent when the dynamics are Markovian and the observations are conditionally independent in time. The batch approach used in the PBS is appealing since it can be wrapped around the model allowing the dynamics to evolve freely for the whole data assimilation window from $t_0$ to

$t_n$ without the need for interruptions in the time integration, typically resulting in marked run time acceleration compared to sequential approaches.

Having discussed the particle methods in some detail, what remains is to outline their implementation in MuSA and in particular the choice of proposal distribution. For simplicity, we adopt the standard and simplest approach for particle methods which is to use the prior as the proposal (van Leeuwen and Evensen, 1996). Note that this analogous to what was done in the

seminal bootstrap filter of Gordon et al. (1993), and is to our knowledge the only approach considered so far for snow data assimilation. Thereby, recalling that $f(\mathbf{x}) = p(\mathbf{y}|\mathbf{x})p(\mathbf{x})$ and inserting for $q(\mathbf{x}) = p(\mathbf{x})$ then $f(\mathbf{x})/q(\mathbf{x}) = p(\mathbf{y}|\mathbf{x})$ which gives us the following simple expression for the auto-normalized weights

$$w^{(i)} = \frac{f(\mathbf{x}^{(i)})/q(\mathbf{x}^{(i)})}{\sum_{j=1}^{N} f(\mathbf{x}^{(j)})/q(\mathbf{x}^{(j)})} = \frac{p(\mathbf{y}|\mathbf{x}^{(i)})}{\sum_{j=1}^{N} p(\mathbf{y}|\mathbf{x}^{(j)})}, \tag{30}$$

which is simply the normalized likelihood of the prior particles $\mathbf{x}^{(i)} \sim p(\mathbf{x})$. With the usual Gaussian likelihood employed in

MuSA these weights are given by

$$w^{(i)} = \frac{\exp\left(-\frac{1}{2}\left[\mathbf{y} - \widehat{\mathbf{y}}^{(i)}\right]^{\mathrm{T}} \mathbf{R}^{-1}\left[\mathbf{y} - \widehat{\mathbf{y}}^{(i)}\right]\right)}{\sum_{j=1}^{N} \exp\left(-\frac{1}{2}\left[\mathbf{y} - \widehat{\mathbf{y}}^{(j)}\right]^{\mathrm{T}} \mathbf{R}^{-1}\left[\mathbf{y} - \widehat{\mathbf{y}}^{(j)}\right]\right)}, \tag{31}$$

where $\widehat{\mathbf{y}}^{(i)} = \mathcal{H}(\mathbf{x}^{(i)})$ are the predicted observations for the $i$-th particle. In practice, to ensure numerical stability, we first compute the natural logarithm of the weights by using the log-sum-exp trick to avoid potential overflow (Murphy, 2022). Subsequently, the weights can be diagnosed by taking the exponential of these stable logarithms.

In the current version of MuSA we apply resampling at each observation timestep for the PF independently of what the effective sample size is. Furthermore the state $\mathbf{x}$ is split into parameters $\mathbf{u}$ and internal states $\mathbf{v}$. The PF is then implemented by first drawing the initial parameters from the prior $\mathbf{u}^{(i)} \sim p(\mathbf{u})$ then for $k = 1, \ldots, n$ timesteps:

- For all particles ($i = 1, \ldots, n$) jitter the parameters $\mathbf{u}_k^{(i)} = \mathbf{u}_{k-1}^{(i)} + \boldsymbol{\eta}_{k-1}^{(i)}$ and run the dynamical model forward in time $\mathbf{v}_k = \mathcal{M}(\mathbf{u}_k, \mathbf{v}_{k-1})$.

- If this is a timestep with observations, update the ensemble of particles by calculating the auto-normalized weights $w_k^{(i)} \propto \exp\left(-\frac{1}{2}[\mathbf{y}_k - \widehat{\mathbf{y}}_k^{(i)}]^{\mathrm{T}} \mathbf{R}_k^{-1}[\mathbf{y}_k - \widehat{\mathbf{y}}_k^{(i)}]\right)$ using the current observations $\mathbf{y}_k$ and predicted observations $\widehat{\mathbf{y}}_k^{(i)} = \mathcal{H}\left(\mathbf{u}_k^{(i)}, \mathbf{v}_k^{(i)}\right)$.





– If this is a timestep with observations, resample the particles based on their weights $w_k^{(i)}$ and then reset all the weights to be equal, i.e. $w_k^{(i)} = 1/N$.

Note that the particles will all have a posterior weight equal to due to the resampling which happens at each observation
timestep. Combined with the transition density, these weights form the prior for the next observation time. This explains why no weights from the prior enter into the posterior weight calculation, since these are all equal and so cancel in the auto-normalization. For the PF, the DAW can be understood to be the time interval between two neighboring observation times. The prior mean and variance of the internal states can be obtained by taking the (unweighted) ensemble means and variances of the particle trajectories $\mathbf{v}_k^{(i)}$ in the DAW before the resampling step. The posterior means and variances of the internal states
can be obtained by taking the (unweighted) ensemble mean and variance of the particle trajectories $\mathbf{v}_k^{(i)}$ in the DAW after the resampling step.

In MuSA, the PBS is even more straightforwards to implement than the PF. The PBS algorithm proceeds by first drawing the prior parameters $\mathbf{u} \sim p(\mathbf{u})$, then for time steps $k = 1, \ldots, n$:

– For all particles run the dynamical model forward in time, $\mathbf{v}_k = \mathcal{M}(\mathbf{u}, \mathbf{v}_{k-1})$.

– If this is a timestep with observations, append the current observations $\mathbf{y}_k$ and predicted observations $\widehat{\mathbf{y}}_k^{(i)} = \mathcal{H}(\mathbf{u}^{(i)}, \mathbf{v}_k^{(i)})$ to the batch of observations $\mathbf{y}$ and predicted observations $\widehat{\mathbf{y}}^{(i)}$, respectively.

– If $k = n$, calculate the auto-normalized weights for the entire DAW using the batch of observations and predicted observations $w^{(i)} \propto \exp\left(-\frac{1}{2}[\mathbf{y} - \widehat{\mathbf{y}}^{(i)}]^{\mathrm{T}} \mathbf{R}^{-1} [\mathbf{y} - \widehat{\mathbf{y}}^{(i)}]\right)$.

For the PBS, the DAW is the entire water year. The prior mean and variance of the internal states can be obtained by taking the
(unweighted) ensemble means and variances of the particle trajectories $\mathbf{v}_k^{(i)}$ in the DAW. The posterior means and variances of the internal states are obtained by taking the weighted ensemble mean and variance of the particle trajectories $\mathbf{v}_k^{(i)}$ in the DAW.

### 3.5  Ensemble Kalman methods

The ensemble Kalman filter (EnKF) was originally proposed by Evensen (1994) as a Monte Carlo version of the original Kalman filter (KF; see Jazwinski, 1970; Särkkä, 2013). The original KF assumes that all distributions involved in filtering are
Gaussian and that both the dynamical and observation models are linear. These assumptions come on top of the usual filtering assumptions of Markovian state dynamics and conditionally independent observations. If all these assumptions are met, the exact filtering distribution turns out to be a Gaussian of the form $p(\mathbf{x}_k|\mathbf{y}_{1:k}) = \mathrm{N}(\mathbf{x}_k|\mathbf{m}_k, \mathbf{P}_k)$ and there is a set of closed form (i.e. analytical) equations for the mean $\mathbf{m}_k$ and covariance $\mathbf{P}_k$, which completely defines this Gaussian filtering distribution, known as the KF equations(Särkkä, 2013). Geoscientific models are almost never linear so these equations can usually not be
applied directly in practice. Even when they can, the need to explicitly store and update the state covariance matrix $\mathbf{P}_k$ with dimensions $m \times m$ can be computationally prohibitive in high dimensional problems typical in geoscience (Carrassi et al., 2018).



There exist several modified versions of the KF such as the extended KF (EKF) based on Taylor series approximations and the unscented KF (UKF) based on the unscented transform that can both partly circumvent the linearity assumption (Särkkä, 2013). Since both the EKF and UKF also require an explicit covariance matrix update and are considerably more challenging to implement than the KF, they have only rarely been used for geoscientific DA. Instead, since its introduction by Evensen (1994), it is the EnKF that has become one of the methods of choice for DA. The reasons for this are that the EnKF is relatively straightforward to implement and understand (Katzfuss et al., 2016), it can handle non-linear models, the state covariance evolves implicitly with the ensemble, and it has proven to work well in very high dimensional operational applications both in meteorology and oceanography (see Carrassi et al., 2018, for a review). Although it makes the same Gaussian linear assumptions as the KF, in practice it can often still function well when these assumptions are strongly violated thanks to techniques such as Gaussian anamorphosis (Bertino et al., 2003) and iterations (Emerick and Reynolds, 2013). The ensemble Kalman approach can also be applied to solve smoothing problems as first shown by van Leeuwen and Evensen (1996). These smoothers, particularly the so-called ensemble smoother (ES), play an important role in solving reanalysis problems that arise in both land surface and snow data assimilation (Dunne and Entekhabi, 2005; Durand and Margulis, 2006).

The EnKF was one of the first schemes to be used for snow data assimilation. Slater and Clark (2006) used the EnKF to assimilate SWE data into a conceptual snow model at several sites in Colorado, leading to marked improvements in SWE estimates compared to both the control model runs and interpolated observations. Clark et al. (2006) used a snow depletion curve to assimilate synthetic satellite retrievals of FSCA into a conceptual snow hydrology model of a catchment in Colorado, leading to minor improvements in simulated streamflow. Durand and Margulis (2006) jointly assimilated synthetic satellite retrievals of brightness temperature and albedo into the SSiB3 model using the EnKF at site in the California Sierra Nevada to yield marked improvements compared to the open loop. Andreadis and Lettenmaier (2006) used the EnKF to assimilate MODIS FSCA into the VIC model for the Snake River basin in the Pacific Northwest region of the USA, improving model estimates of both FSCA and SWE.

Following on from these initial studies there have been several more recent snow DA applications using the EnKF. In a synthetic experiment in northern Colorado, De Lannoy et al. (2010) showed how the EnKF can be used to assimilate coarse scale passive microwave SWE observations into high resolution runs of the Noah land surface model, leading to large reductions of error in SWE estimation compared to the open loop. In a follow up study at the same site, De Lannoy et al. (2012) jointly assimilated real MODIS FSCA and AMSR-E SWE retrievals into the Noah model to demonstrate the complementary nature of these types of satellite observations for SWE estimation. Magnusson et al. (2014) used the EnKF to assimilate data from ground-based stations in the Swiss Alps and showed that assimilating fluxes (snowmelt and snowfall), rather than directly assimilating SWE data, improved the SWE estimation. The study of Huang et al. (2017) demonstrated that the assimilation of in-situ SWE data into a hydrological model with the EnKF could improve streamflow simulations for several basins in the western USA. Stigter et al. (2017) performed a joint assimilation of in-situ snow depth data and FSCA satellite retrievals with the EnKF to optimize parameters and subsequently estimate the climate sensitivity of SWE and snowmelt runoff for a Himalayan catchment. More recently, Hou et al. (2021) showed how machine learning techniques can be used to construct empirical observation operators to assimilate satellite-based FSCA with the EnKF.





The ensemble smoother (ES), the batch smoother version of the EnKF, has also been used extensively for snow DA, mainly for reanalysis. Durand et al. (2008) were the first to demonstrate the value of the ES for snow reanalysis by noting that it

could be used in a more robust Bayesian approach to solve the traditional problem of snow reconstruction. They showed substantial improvements in SWE estimation for the posterior compared to the prior after having assimilated synthetic FSCA observations into SSiB3. A key advantage of the ES over the EnKF is that it allows information to propagate backwards in time, so that observations made in the ablation season can update the preceding accumulation season. Following up this study with a real experiment assimilating Landsat FSCA retrievals into SSiB3 using the ES at a basin in the California Sierra Nevada,

Girotto et al. (2014a) demonstrated explicitly how snow reanalysis using the ES is a more robust probabilistic generalization of the traditional deterministic approach to snow reconstruction. Subsequently, Girotto et al. (2014b) applied the same ES snow reanalysis framework to produce a multi-decadal high resolution snow reanalysis for the Kern River watershed in the California Sierra Nevada. Oaida et al. (2019) used the ES to assimilate MODIS FSCA observations into the VIC model to produce a kilometer-scale snow reanalysis over the entire western USA. Aalstad et al. (2018) introduced an iterative version

of the ES, the ensemble smoother with multiple data assimilation scheme (ES-MDA; Emerick and Reynolds, 2013), for snow reanalysis and showed how this could outperform both the ES and the PBS when assimilating FSCA retrievals at sites on the high-Arctic Svalbard archipelago.

A derivation of the KF equations, which the EnKF and ES are largely based on, is beyond the scope of this work. Full multivariate Bayesian derivations of these equations can be found on page 197 of Jazwinski (1970) and Section 4.3 of Särkkä

(2013). The EnKF itself is thoroughly derived in Evensen (2009) and Evensen et al. (2022). There are actually several variants of the ensemble Kalman analysis step (see Carrassi et al., 2018). In MuSA we use the so-called stochastic rather than deterministic (or square-root) implementation. This stochastic formulation was proposed by Burgers et al. (1998) who suggested adding stochastic perturbations to the observations to ensure adequate ensemble spread, but it was recently corrected by van Leeuwen (2020) who showed that perturbations should actually be applied to the predicted observations. In practice, due to

the symmetry of the perturbations, this recent correction should have minimal impact but it provides much needed clarification and is consistent with the underlying Bayesian theory.

Here we present the equations for both the stochastic EnKF (van Leeuwen, 2020) and the ES (van Leeuwen and Evensen, 1996), as well as their iterative versions based on the multiple data assimilation (MDA) scheme of Emerick and Reynolds (2012, 2013), in the form that they are used in MuSA. Let $N_a$ denote the number of assimilation cycles (iterations) per-

formed in a pseudo (rather than model) time. For the standard EnKF and ES we set $N_a = 1$ while for their iterative variants $N_a > 1$, typically with $N_a = 4$ (Emerick and Reynolds, 2013; Aalstad et al., 2018). The superscript $\ell$ is used to index these iterations. Let $\mathbf{U}^{(\ell)} = \left[\mathbf{u}^{(1)(\ell)}, \ldots, \mathbf{u}^{(i)(\ell)}, \ldots, \mathbf{u}^{(N)(\ell)}\right]$ denote the $m_p \times N$ parameter matrix containing the ensemble ($i = 1, \ldots, n$) of parameter vectors $\mathbf{u}^{(i)(\ell)}$ for iteration $\ell$. Recall that the subset of these parameters that are physically bounded have undergone the relevant analytic transformations for Gaussian anamorphosis, and the corresponding inverse transforms

are applied back to physical space when these are passed through the dynamical model (see Aalstad et al., 2018). Similarly, let $\widehat{\mathbf{Y}}^{(\ell)} = \left[\widehat{\mathbf{y}}^{(1)(\ell)}, \ldots, \widehat{\mathbf{y}}^{(i)(\ell)}, \ldots, \widehat{\mathbf{y}}^{(N)(\ell)}\right]$ denote the predicted observation matrix containing the ensemble of predicted ob-



servations $\widehat{\mathbf{y}}^{(i)(\ell)} = \mathcal{H}\left(\mathbf{u}^{(i)(\ell)}, \mathbf{v}^{(i)(\ell)}\right)$. Then the stochastic ensemble Kalman methods proceed by first drawing the initial parameters from the prior $\mathbf{u}^{(\ell=0)} \sim p(\mathbf{u})$, then for $\ell = 0 : N_a$ iterations:

- Run the dynamical model forward in time for all ensemble members $i$ to obtain the internal states $\mathbf{v}^{(i)(\ell)}$ and predicted observations $\widehat{\mathbf{y}}^{(i)(l)}$ corresponding to the ensemble of parameter vectors $\mathbf{u}^{(i)(\ell)}$.

- If $\ell < N_a$ perform the ensemble Kalman analysis step

$$\mathbf{U}^{(\ell+1)} = \mathbf{U}^{(\ell)} + \mathbf{K}^{(\ell)}\left[\mathbf{Y} - \left(\widehat{\mathbf{Y}}^{(\ell)} + \boldsymbol{\mathcal{E}}_\alpha^{(\ell)}\right)\right], \tag{32}$$

where $\mathbf{Y}$ is an $d \times N$ matrix containing $N$ copies of the observation vector $\mathbf{y}$ while the observation error term is given by

$$\boldsymbol{\mathcal{E}}_\alpha^{(\ell)} = \sqrt{\alpha^{(\ell)}}\mathbf{R}^{1/2}\boldsymbol{\epsilon}^{(\ell)}, \tag{33}$$

in which $\boldsymbol{\epsilon}^{(\ell)}$ is an $d \times N$ matrix containing draws from a standard Gaussian $\mathrm{N}(0,1)$ and $\alpha^{(\ell)} = N_a$ is the observation error inflation coefficient. The (ensemble) Kalman gain $\mathbf{K}^{(\ell)}$ is an $m_p \times d$ matrix given by

$$\mathbf{K}^{(\ell)} = \mathbf{C}_{\mathbf{U}\widehat{\mathbf{Y}}}^{(\ell)}\left(\mathbf{C}_{\widehat{\mathbf{Y}}\widehat{\mathbf{Y}}}^{(\ell)} + \alpha^{(\ell)}\mathbf{R}\right)^{-1}, \tag{34}$$

where, in these ensemble Kalman methods, the model-based covariance matrices are estimated from the ensemble such that the $m_p \times d$ parameter-predicted observation covariance matrix is given by

$$\mathbf{C}_{\mathbf{U}\widehat{\mathbf{Y}}}^{(\ell)} = \frac{1}{N}\mathbf{U}^{(\ell)'}\widehat{\mathbf{Y}}^{(\ell)'\mathrm{T}}, \tag{35}$$

while the $d \times d$ predicted observation covariance matrix is given by

$$\mathbf{C}_{\widehat{\mathbf{Y}}\widehat{\mathbf{Y}}}^{(\ell)} = \frac{1}{N}\widehat{\mathbf{Y}}^{(\ell)'}\widehat{\mathbf{Y}}^{(\ell)'\mathrm{T}}, \tag{36}$$

where primes $(\cdot)'$ denote deviations from the ensemble mean.

Recall that for $N_a = 1$ we recover the non-iterative stochastic EnKF and ES, while for $N_a > 1$ we are using iterative versions of these schemes that involve multiple data assimilation with inflated observation errors. This is tantamount to tempering the likelihood as discussed by Stordal and Elsheikh (2015); van Leeuwen et al. (2019) which explains why these iterative schemes perform better than their non-iterative counterparts for non-linear models in that they involve a more gradual transition from the prior to the posterior. Despite what the name might suggest, this multiple data assimilation approach does not actually violate the consistency of Bayesian inference by using the data more than once due to the way the observation error inflation is constructed, particularly due to the constraint that $\sum_{\ell=1}^{N_a]} = 1$. It is possible to satisfy this constraint both with uniform and non-uniform inflation coefficients (c.f. Evensen, 2019). For simplicity, following Emerick and Reynolds (2013), we currently opt for former as a default in MuSA by setting $\alpha^{(\ell)} = N_a$ set $\alpha^{(\ell)} = N_a$ while allowing for the latter as an option.

To keep the notation as simple as possible, we have not explicitly introduced time indices for these ensemble Kalman methods. Instead we can simply note that for the iterative and non-iterative EnKF the for loop above runs inside another outer





loop that runs across all observation times. That is to say, the $\ell$ loop is run several times sequentially for multiple DAWs where each window is from one observation time to the next, such that the posterior parameters and internal states obtained at $\ell = N_a$ for the current DAW that ran up to the current observation time becomes the initial prior at $\ell = 0$ for the next DAW that runs from the current observation time up to the next observation time. Recall that the parameters also undergo jitter dynamics when

710   filtering.

For the non-iterative and iterative ensemble batch Kalman smoothers used in MuSA, namely the ES and the ES-MDA, there is no longer an outer time loop encapsulating the iterations. Instead, the DAW is the entire water year. In these smoother schemes, the parameters do not undergo any jitter. As such, the entire prior and posterior ensemble of state trajectories for a given water year will be consistent with the prior and posterior ensemble of parameters, respectively, that are both static for

715   the water year in question. Recall that each water year is treated independently in MuSA which currently focuses on seasonal snow.

## 4   Data and experimental setup

In the following section, we illustrate the capabilities of MuSA using a case study and perform a benchmarking of the implemented data assimilation algorithms. We developed two data assimilation experiments in the Izas experimental catchment

720   (Revuelto et al., 2017) in the Spanish Pyrenees (see Figure 2). The first experiment shows the capabilities of MuSA to assimilate hyper-resolution products in both a single cell and distributed fashion, whereas the second demonstrates the capabilities to develop joint assimilation experiments. First, we generated hourly meteorological forcing at 5 m resolution over the Izas catchment by means of the MicroMet meteorological distribution system (Liston and Elder, 2006b), forced by the ERA5 atmospheric reanalysis (Hersbach et al., 2020).

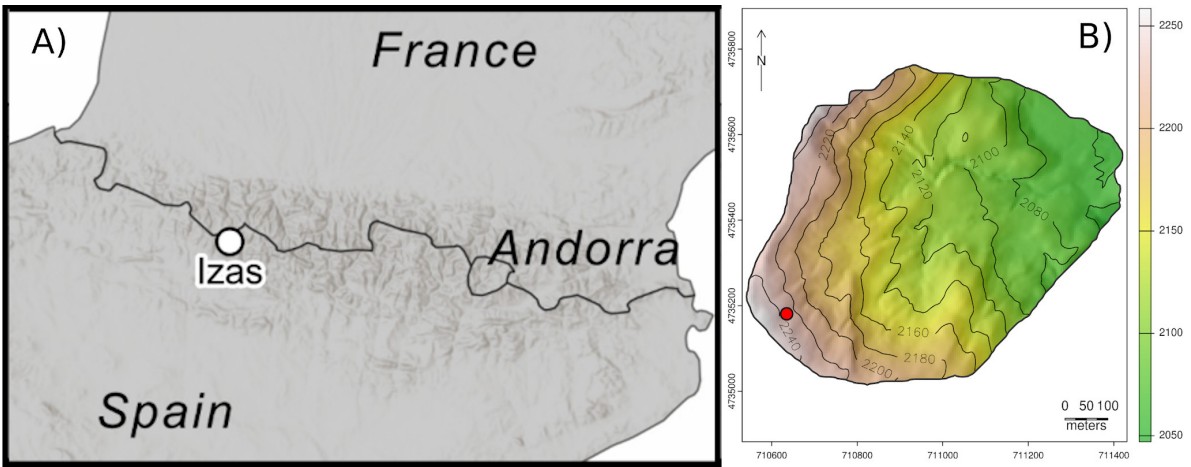

**Figure 2.** Location of the Izas experimental basin in the Pyrenees (A) and topography (B). The red circle indicates the location of the cell used for the intercomparison of algorithms described below.



## 4.1 Drone data assimilation experiment

For the first experiment, we ran MuSA to assimilate snow depth retrievals obtained from a fixed wing drone (**?**Revuelto et al., 2021a), over two different snow seasons (2018/2019 and 2019/2020) in the Izas catchment using different data assimilation algorithms. The drone surveys were not equally distributed in time along the seasons, and the surveys present spatial gaps in some of the cells. The whole drone dataset is composed of 18 temporally distributed snow depth maps retrieved at 5m spatial resolution. Due to the high-resolution of the observations, we expect MuSA to be able to reproduce the wind redistribution patterns by inflating the uncertainty of the forcing to compensate for the deposition and removal of wind-blown snow.

The simulation ensemble was composed of 200 particles. The generation of the ensemble for both the single cell and distributed simulation was developed by perturbing only the temperature and precipitation fields. The temperature was perturbed with a constant additive bias parameter, randomly drawn from a normal distribution defined by its mean ($\mu = 0$) and standard deviation ($\sigma = 2$). Similarly, the precipitation was perturbed by a multiplicative fixed in time bias parameter, drawn from a log-normal distribution defined by the mean ($\mu = 0$) and standard deviation ($\sigma = 0.63$) of the underlying normal distribution. These prior distributions were defined based on prior predictive testing (not shown here). The variances of the observation errors were set to $0.04$ [m$^2$] and was estimated from comparison with terrestrial laser scanning snow depth maps (**?**Revuelto et al., 2021a). In the case of the single cell comparison, we followed two different resampling strategies, the bootstrap and redraw from a normal approximation of the posterior. In the iterative versions of the ensemble Kalman based approaches we fixed the number of assimilation cycles to $N_a = 4$.

The single cell simulations were performed in a cell located in a topographic concavity within the catchment, and therefore with exceptional accumulations and snow depth up to 6 m due to wind redistribution and preferential deposition (Comola et al., 2019). This is a challenge for the data assimilation process, as the observations fell very far from the prior range of the ensemble of snow depths predicted by FSM2. The distributed simulation was developed in a supercomputing cluster using 20 nodes with 10 cores each. For this case study, we choose the most computationally efficient data assimilation algorithm, namely the PBS, to speed up the calculations. We subsequently compared the spatial distribution of the snowpack with one of the snow depth maps derived from the drone surveys that were not included in the assimilation.

## 4.2 Joint satellite data assimilation experiment

For the second experiment, we performed a coarser 1 km resolution experiment over the same area. We developed a single cell joint assimilation experiment, updating the FSM2 simulations with FSCA and land surface temperature (LST) retrievals from the MODIS sensor on board the Terra satellite, by using version 6 of the products MOD11A1 (Hall and Riggs, 2016) and MOD10A1 (Wan et al., 2015), respectively. The selected pixel for the MODIS retrievals was the one whose centroid fell closest to the Izas catchment centroid. While the FSCA retrievals from MODIS have a 500 m spatial resolution, the LST products have a resolution of 1000 m. To assimilate this information into FSM2 using the MuSA platform, we aggregated the FSCA products to match the 1 km LST grid. The forcing was generated by aggregating the 5 m forcing from the previous distributed data assimilation experiment. The variances of the observation errors were set to $0.15$ and $10$ [K$^2$] for the FSCA and





the LST, respectively. The data assimilation experiment was done using the ES-MDA scheme. This data assimilation scheme was chosen because it showed a better accuracy compared with the other algorithms (not shown here), that in some cases were
not able to meaningfully update the simulations with the observations at all.

The number of iterations was again set to $N_a = 4$. The ensemble was composed of $N = 300$ particles by perturbing all the forcing variables (temperature, precipitation, short and longwave radiation, wind speed, relative humidity and surface air pressure) with normal additive or lognormal multiplicative random noise depending on whether or not the forcing variable has negative support so as to not generate physically impossible values.

### 4.3  Computational benchmarks

Finally, we developed a single cell benchmark, to measure the computational cost in terms of wall clock time for each algorithm. The benchmark was developed in a local machine with an Intel(R) Core(TM) i5-1145G7 processor and 32GB of memory running an Ubuntu 20.04 system. The comparison was performed using 100, 200 and 300 particles and four iterations for the iterative ensemble Kalman approaches in a single cell. The reported values of the benchmarks are the average of 10 MuSA
runs, and includes the FSM2 compilation time ($\simeq$2 seconds using the GNU Fortran compiler 10.3.0 in the aforementioned local machine) which is negligible compared with the whole run.

## 5  Results

### 5.1  Single cell and distributed hyper-resolution drone data assimilation

The results show how the performance of the different data assimilation algorithms differs, even with the same initial conditions
and experimental setup (Figure 3 and Figure 4, single cell test comparison). The deterministic open loop simulation exhibited a root mean squared error (RMSE) compared with the observations of 3.1 m, higher than the posterior average of any of the algorithms implemented in MuSA as shown in Table 1:

| | Scheme | | | | | | |
|---|---|---|---|---|---|---|---|
| Metric | PF-c | PF-r | EnKF | EnKF-MDA | PBS | ES | ES-MDA |
| RMSE [m] | 1.51 | 0.04 | 0.98 | 0.34 | 0.17 | 0.16 | 0.09 |

**Table 1.** Evaluation metrics for the the posterior ensemble mean based on the assimilated observations.

The particle filter collapsed early in the singe cell simulation where all the weight is shared by just a few and eventually a single particle (see Figure 3), due to the large difference between the observations and the ensemble of predictions at the first
assimilation step. The particle filter with redraw allowed the assimilation process to recover from the initial collapse through particle rejuvenation, leading to a more realistic non-degenerate ensemble simulation. Conversely, the ensemble Kalman-based





approaches were less prone to ensemble degeneracy. Here, the ensemble Kalman filter produced unsatisfactory results when the observations fall very far from the ensemble. However, we did not observe this issue with the iterative version of the ensemble Kalman filter.

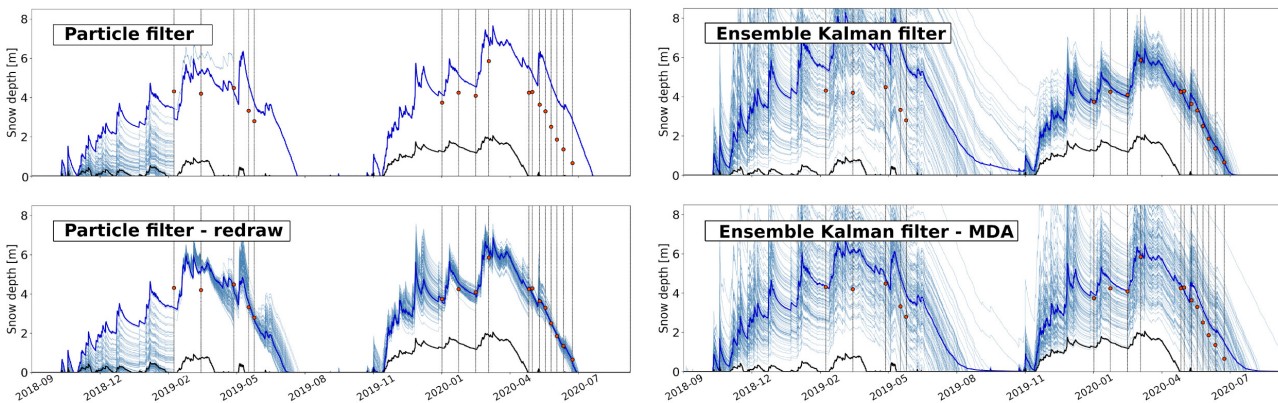

**Figure 3.** Intercomparison of filter based data assimilation experiments. The black line is a deterministic (no perturbations) open-loop simulation and the red dots are the observations, the light blue lines are the ensemble of posterior state while the dark blue line is the ensemble mean.

Figure 4 shows the results of the experiment using the smoother based approaches. In this particular case, the posterior snow depth time series generated by using the PBS are much closer to the observations than when using particle filters with the bootstrap resampling (Figure 3). In addition, due to the absence of resampling, the PBS did not generate discontinuities in the series that can be observed in the particle filters posteriors (Figure 3). The ensemble (Kalman) smoothers also outperformed their filter counterpart in this particular case. Moreover, the iterative ensemble (Kalman) smoother produced simulations closer

to the observations compared to the other smothers as confirmed in Table 1).

    There is an obvious improvement in the snow depth spatial distribution patterns after running MuSA compared with the open-loop simulations for the hyper-resolution distributed simulation as shown in Figure 5. The updated SWE products exhibited very consistent spatial patterns compared with the independent (i.e. non-assimilated) drone-based snow depth map, while the open loop simulations exhibited a much lower and less realistic spatial variability. Similarly, the updated FSM2 snow depth

simulations were very consistent with the non-assimilated snow depth maps with a coefficient of determination of $R^2$=0.96 while the open loop simulation exhibited an $R^2$=0.03

    Figure 6 shows the spatial distribution of the weighted mean temperature bias (based on a normal prior distribution, i.e. an additive factor) and precipitation bias (based on log-normal prior distribution, i.e. a multiplicative factor) over the Izas catchment after the assimilation of the drone-based snow depth retrievals. The distribution of the perturbation parameters

showed spatial patterns consistent with the topography of Izas in both of the variables, ranging from -4 to 2 [K] in the case of the additive temperature bias and from 0.5 to 3 in the case of the multiplicative precipitation bias..





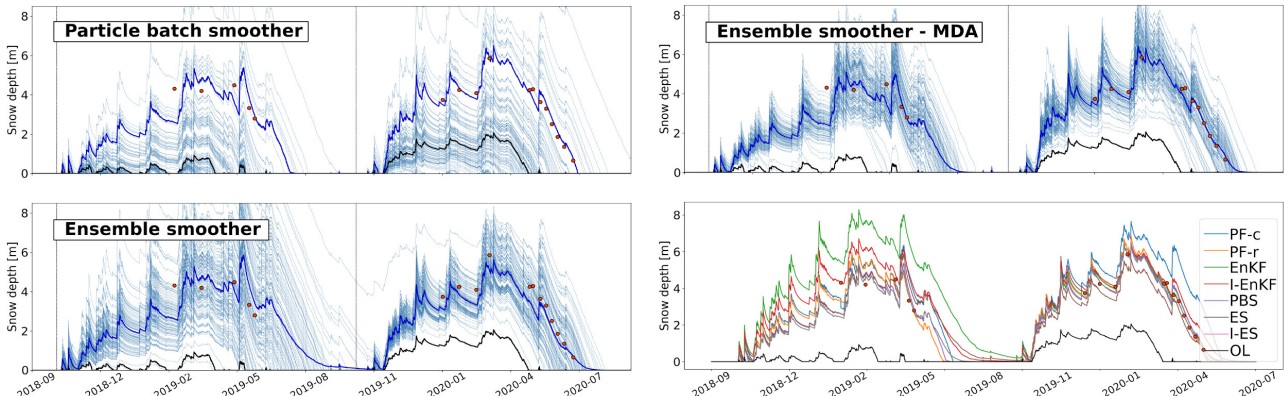

**Figure 4.** Intercomparison of smoother based data assimilation experiments and intercomparison of all the data assimilation algorithms. The black line is a deterministic (no perturbations) open-loop simulation and the red dots are the observations, the light blue lines are the ensemble of posterior state while the dark blue line is the ensemble mean. PF-c, PF-r, I and OL refers to particle filter (classic resampling), particle filter (redraw from posterior resampling), iterative versions of the algorithms and the open loop, respectively.

## 5.2 Coarser-scale joint satellite data assimilation

The results of the LST and FSCA joint assimilation is shown in Figure 7. As expected, the difference between the observations and the updated simulations was reduced, compared with the open loop simulations. The root mean squared error (RMSE) between the observations and the open loop LST was 7.2 [K] compared to 5.1 [K] for the updated simulations. Similarly, the RMSE between the observations and the open loop FSCA was 0.38, compared to 0.14 for the updated simulations. Such modifications of the FSCA and LST state variables resulted in a completely different SWE simulation, that reached values up to 900 mm of peak SWE compared to the 300 mm reached by the open loop simulation.

## 5.3 Computational benchmarks

In a single cell benchmark experiment, the different data assimilation algorithms showed large variations in computational cost (Figure 8). In this comparison the values ranged from 39 seconds per cell per year $[\mathrm{s\,c^{-1}\,y^{-1}}]$ in the case of the PBS to 270 $[\mathrm{s\,c^{-1}\,y^{-1}}]$ for 100 particles and 4 iterations in the case of the EnKF-MDA. These benchmarks showed that for all the cases the computational cost increased almost linearly with the number of particles. As expected, the iterative versions of the ensemble Kalman approaches are more demanding in terms of computational cost, due to the larger number of FSM2 simulations required by the data assimilation algorithm. Filtering is also generally more expensive than (batch) smoothing since it invokes more frequent calls to I/O operations. This is especially clear for the costly iterative ensemble Kalman methods, but note that the relative increase in wall clock time for filtering relative to smoothing is more or less the same across all methods and corresponds to a factor of roughly 1.5.





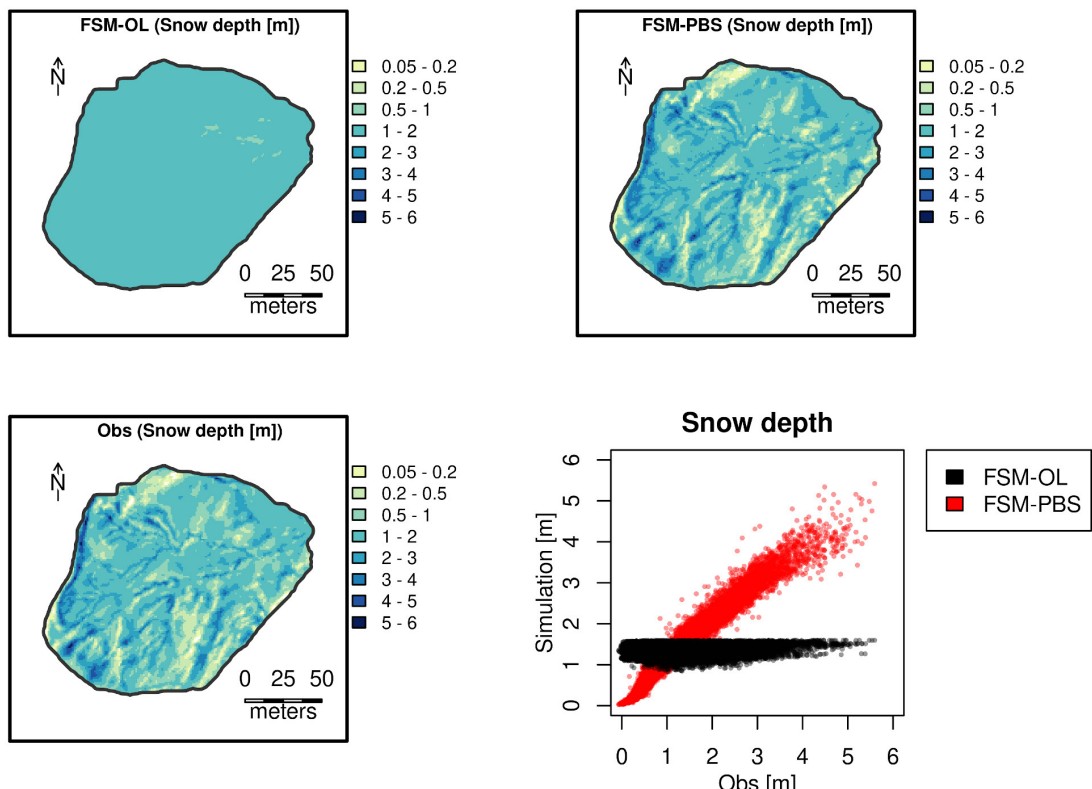

**Figure 5.** Open loop FSM simulation and updated FSM simulation (upper panels). Independent drone snow depths compared to the open loop (black) and the posterior (red), respectively (lower panels).

## 6    Discussion

The results from the intercomparison of different data assimilation experiments exhibited a large variability in performance and computational cost across the schemes. This highlights the need for thorough testing, as the suitability of the assimilation algorithms will vary depending on the problem at hand. Despite the fact that all of the data assimilation algorithms improved the simulations compared with the deterministic open loop simulations, their performance differed markedly. In fact, this is sometimes not noticed in the literature, where often the choice of one algorithm over others is not sufficiently, or at all,

justified. The lack of tools to compare the performance of different data assimilation algorithms has probably contributed to this problem, since it requires substantial coding effort to implement all available options in each data assimilation experiment. As an example, ensemble Kalman-based data assimilation approaches are often perceived as suboptimal for snow science applications due to the Gaussian linear assumption (Helmert et al., 2018; Largeron et al., 2020). In spite of this view, our first experiment showed that the iterative version of ensemble-based Kalman smoothers outperformed the other smoother

algorithms, as also found by Aalstad et al. (2018). hese findings are consistent with the broader DA literature, where basic

# Perturbation parameters

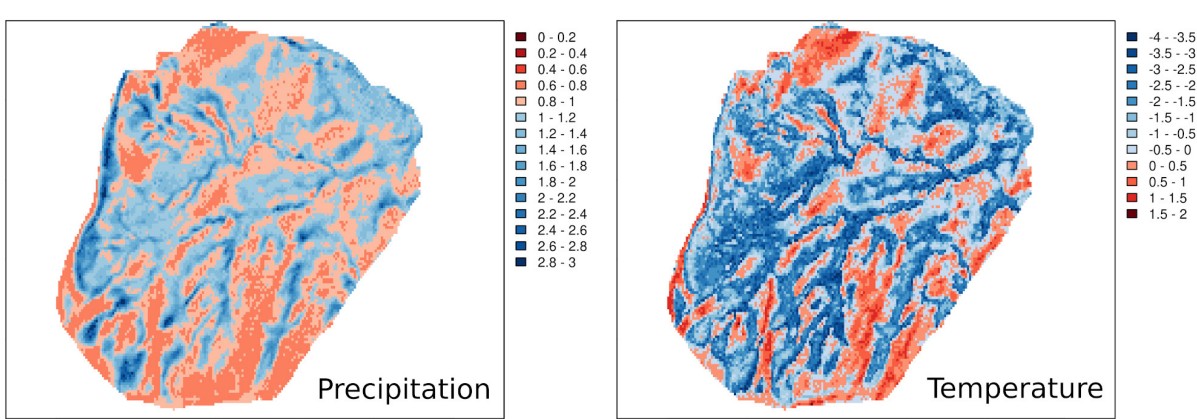

**Figure 6.** Posterior perturbation parameters for the precipitation (multiplicative) and temperature (additive).

particle methods tend to suffer more from the curse of dimensionality (Snyder et al., 2008). In their review, van Leeuwen et al. (2019) suggest promising remedies to this problem with more sophisticated particle methods that invoke remedies such as using iterations or proposal distributions. The particle filter using redraws from a normal approximation to the posterior that we implemented in MuSA was loosely inspired by the use of proposal distributions and can overcome the degeneracy

problems in the more classic particle filter (see Table 1). The power of MuSA is that it simplifies the task of comparing different experimental set-ups by allowing the intercomparison of data assimilation algorithms as well as the implementation of new ones. In the future, new data assimilation algorithms will be implemented, including iterative versions of the PBS and particle filters as well MCMC methods which are the gold standard for Bayesian inference (Neal, 1993; Apte et al., 2007), but have received relatively little attention from the snow community (Kolberg and Gottschalk, 2006) due to their often prohibitive

computational cost.

The selection of the data assimilation algorithm is not the only possible source of variability in the updated simulations. As an example, MuSA has two different precipitation partitioning methods implemented. In addition to the fact that the selection of the precipitation phase partitioning method would have in the deterministic simulations, the perturbations in the forcing can also impact precipitation phase in different ways depending on the selected partitioning method. For example, perturbing the

relative humidity forcing would only impact precipitation phase for one of the partitioning methods in FSM2. In addition, different strategies to generate the prior ensemble of simulations may impact the performance of the data assimilation algorithms especially in the case of PBS and particle filters which are prone to degeneracy.

MuSA was also able to assimilate hyper-resolution snow depth maps in a distributed fashion. The assimilation of snow depth products has been shown to be a very robust approach to SWE estimation. The posterior maps of the perturbation parameters

showed intricately detailed and physically-consistent spatial patterns, especially considering the fact that there is not any cell intercommunication in MuSA. The appearance of this consistent spatial pattern in the perturbation parameters indicates that







**Figure 7.** Joint LSTA (top panels) and FSCA (middle panels) data assimilation, and its influence on the SWE simulations (bottom panel).).

they are compensating for a missing processes in the model, which in this case is most likely to be the wind-driven ablation and accumulation processes since most of the spatial variability in melt energy is provided by MicroMet shortwave radiation routine (e.g. Baba et al., 2019). Thus, MuSA can be used to study the importance of missing snow processes in the FSM2 model. In 855 the particular case of the assimilation of hyper resolution snow depth maps, it may help to address the unresolved problem of wind redistribution in numerical snowpack modeling (Vionnet et al., 2021) by providing spatio-temporally continuous (i.e.





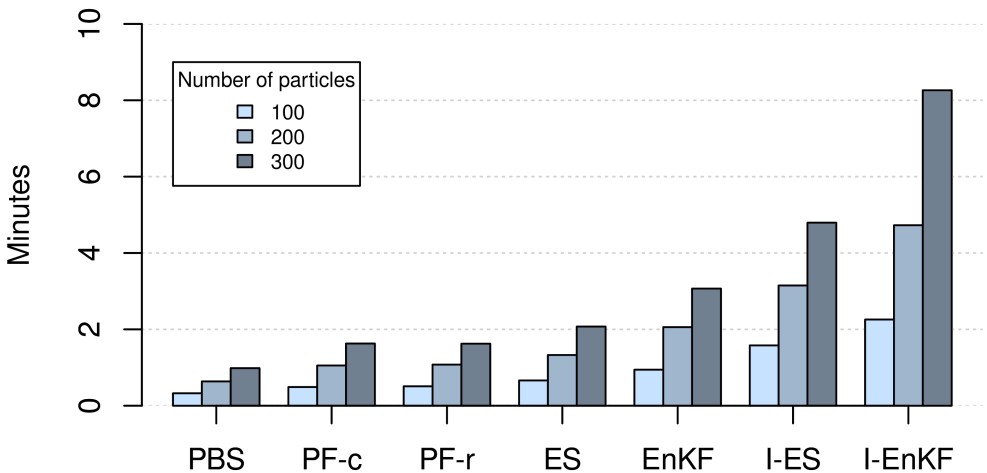

**Figure 8.** Benchmarks for the computational cost, expressed in terms of wall clock time, of the different assimilation schemes implemented in MuSA.

gap free) and physically consistent reconstructions of the snowpack dynamics that can be used as a target when designing and calibrating new process parameterizations in snow model development.

Despite the fact that there are some other examples of assimilating snow depth products, most of these have been carried out 860 at the point scale, or were developed at coarser spatiotemporal resolutions (Deschamps-Berger et al., 2022). To our knowledge the assimilation experiment performed in Section 5.1 is the first test of assimilating drone retrievals in a numerical snow model. This approach should be useful for the monitoring of hydrological experimental catchments in snow dominated areas, providing a cost effective method to generate spatially and temporally continuous reconstruction of the SWE distribution. Using MuSA in combination with the ever increasing capabilities of drones, it is possible to develop joint assimilation experiments using 865 snow depth and other variables such as snow surface temperature or albedo at unprecedented resolutions. Furthermore, the assimilation of hyper-resolution snow depth maps may become a common practice in the future, even at wider scales, thanks to the emerging remote sensing snow depth retrievals.

The joint assimilation of FSCA and LST retrievals from MODIS, had a large impact in the updated SWE simulations. The assimilation of the FSCA provides information when FSCA saturates at 1, i.e. during most of the snow season in snow domi-870 nated areas, or during the polar night in the absence of light. This preliminary experiment suggests that the joint assimilation of LST and FSCA may be beneficial to improve distributed snow reanalyses at larger scales. There are several examples of





assimilating SCA in both its binary and fractional form (e.g. Margulis et al., 2016). However, the assimilation of LST in numerical snow models remains largely underexplored (Navari et al., 2016). This is probably due to the relative lack of accurate LST information at the desired scale for snow applications especially over complex terrain, as the currently available satellite

sensors that are able to provide LST information either exhibit a spatial resolution that can be too coarse or revisit times that are too infrequent (Largeron et al., 2020). Nonetheless, given the agenda of several space agencies, LST information will be readily available in the future at high spatiotemporal resolutions. This includes the Thermal infraRed Imaging Satellite for High-resolution Natural resource Assessment (TRISHNA; Lagouarde et al., 2018) satellite from the Centre National d'Études Spatiales (CNES) or the Land Surface Temperature Monitoring (LSTM) mission from the European Space Agency (ESA).

The computational cost of each data assimilation algorithm and implementation depends on different factors. First, the computational cost of the Particle Filter and PBS approaches is different to the Kalman and the iterative Kalman approaches. This is due to the number of FSM2 runs required to run each algorithm. In the Particle Filter and PBS approaches, the number of FSM2 runs are $N_r = N_e$ where $N_r$ denotes the number of runs (per grid cell) and $N_e$ is the number of particles. On the other hand, the computational cost in terms of the number of FSM2 runs in the EnKF and ES will be $N_r = 2N_e$. For the iterative

versions of these ensemble Kalman-based approaches the number of runs will be $N_r = (N_a + 1)N_e$ where $N_a$ is the number of assimilation cycles, i.e. the number of iterations in the ensemble Kalman update equations, selected by the user as explained in Section 3.5. In general, the ensemble Kalman techniques require more runs because they actually move (rather than just reweight) the parameter ensemble after assimilating the observations. To get the corresponding updated states (and predicted observations) FSM2 must be rerun with the updated parameters. In the non-iterative case this effectively requires a single rerun

of the FSM2 ensemble, while for the iterative case it requires $N_a$ reruns of the FSM2 ensemble.

However, the number of FSM2 runs is not the only source of computational cost. The current version of MuSA is a wrapper around the FSM2 model. This simplifies the implementation of other snow models if required by the user. It also means that MuSA can easily incorporate FSM2 upgrades. Nonetheless, this implementation comes with a computational cost due to the I/O operations and the system calls that are done in the background when the MuSA system is running. The time expended on

I/O operations and system calls will be higher in the parallel runs due to the use of disk space from different processes at the same time. This suggests that a better encapsulation of FSM2 running the snowpack simulations in memory may improve the overall MuSA performance.

It is worth noting that MuSA may be also used to implement simpler (and therefore) faster snowpack models (e.g. Aalstad et al., 2018), if numerical efficiency is required. Conversely, more sophisticated multilayer models such as Crocus (Vionnet

et al., 2012) and SNOWPACK (Bartelt and Lehning, 2002) could also be encapsulated in MuSA in the future. The implementation of more sophisticated models that include detailed radiative transfer schemes may provide MuSA the capability of ingesting new remotely sensed information such as surface reflectances or radar backscatter. In addition, as FSM2 has support for different temporal resolutions, the future versions of MuSA will support different timesteps, allowing to reduce the computational cost both in terms of run time and data storage requirements. As such, there is room for reducing the computational

cost of MuSA if necessary, which would open up the possibility of implementing it even at continental and hemispheric scales.



# 7 Conclusions

MuSA is a new snow data assimilation system that encapsulates the FSM2 snowpack model. There are 6 different ensemble-based data assimilation algorithms implemented in MuSA, as outlined in detail in Section 3, with five different resampling strategies in the case of particle filters. MuSA also supports direct insertion in the case of snow depth. The data assimilation
algorithms and the characteristics of the ensemble generation are provided by the user through simple configuration files. MuSA is able to assimilate different observational variables either independently or jointly (i.e. in the same assimilation step), even if these variables do not share the same time-step or if they have gaps. The system is highly scalable, such that it is possible to run it both on local machines and supercomputing infrastructures.

We used the MuSA system to assimilate snow depth maps derived from drone retrievals over the Izas experimental catchment
for two different seasons at 5 meters spatial resolution. The behavior of each data assimilation algorithm differed considerably, demonstrating that case-specific testing of algorithms rather than just relying on the literature can be very helpful for designing successful snow data assimilation experiments. In addition, we developed a single cell data assimilation experiment, fusing LST and FSCA retrievals from MODIS with the FSM2 model. The results indicated a strong potential in the joint exploitation of these remotely sensed variables, suggesting that more research is needed on this regard. Finally, we presented a benchmark
of the computing cost of the data assimilation algorithms. The choice of data assimilation algorithms had a considerable impact on the computational expense of the system, and should be considered for high resolution or large scale runs.

*Code and data availability.* The MuSA code is available at github.com/ealonsogzl/MuSA, the version of MuSA used for this paper (with a subset of input data) can be found at https://doi.org/10.5281/zenodo.7014570 while the complete input data for Izas used in the present study can be found at https://doi.org/10.5281/zenodo.7015271. The original FSM2 code is found at github.com/RichardEssery/FSM2, and in the
MuSA repository with slight modifications from the original version. The MODIS data used herein, namely MOD10A1 (Hall et al., 2002) and MOD11A1 (Wan et al., 2015), are available for download from NSIDC and the NASA EOSDIS Land Processes DAAC, respectively. ERA5 data is available for download from the Copernicus Climate Data Store.

*Author contributions.* Conceptualization: EAG, KA, Data curation: EAG, MWB, JR, Formal analysis: EAG, KA, Funding acquisition: EAG, JLP, SG, Investigation: EAG, KA, Methodology: EAG, KA, Project administration: EAG, JLP, SG, Resources:EAG, JLP, Software: EAG,
KA, Supervision: JLP, SG, Validation: EAG, KA, Visualization: EAG, KA, SG, Writing - original draft preparation: EAG, KA, Writing - review & editing: EAG, KA, MWB, JR, JLP, JF, RE, SG

*Competing interests.* The authors declare that they have no conflict of interest.





*Acknowledgements.* Esteban Alonso-González has been funded by the CNES postdoctoral fellowship. Data from Izas experimental catchment was collected in the frame of HIDROIBERNIEVE- CGL2017-82216-R funded by the Spanish Ministry of Economy and Competitiveness project. Kristoffer Aalstad was funded by the Research Council of Norway through the Spot-On project (#301552), and acknowledges support from the LATICE strategic research area at the University of Oslo. J. Revuelto is supported by the Grant IJC2018-036260-I.





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
