# Peer review of "MuSA: The Multiple Snow Data Assimilation System (v1.0)"

_Geoscientific Model Development, 2022_

## Referee Comment (RC2)

[referee-annotated manuscript omitted]

---

## Author Comment (AC1)

**REVIEWER # 1**

**COMMENT # 1.1**

*General Comments:*

*Overall, this is an excellent paper worthy of publication in GMD. The topic of snow data assimilation is of high scientific importance and providing a unifying framework for implementing such methods is to be commended and should be of high value to the community. The structure of the manuscript and presentation are generally clear. There are many specific comments listed below which aim to improve the communication of the proposed framework and aid users in its implementation and use. In particular, more details on the sample problems would aid in the reproducibility and extension of the work to other problems. In testing the code, it appears that the github repository code works, but the sample provided on zenodo has a bug. More details are provided below.*

**Reply:**

We are grateful to the reviewer for the thoughtful comments and suggestions to our manuscript. We have compiled a revised version and in the following provide a point-by-point reply to all issues raised. References are included at the end of this document.

**COMMENT # 1.2**

*Specific Comments:*

*The comments provide herein represent a list of relatively minor additions and/or corrections that would improve the paper.*

*1. Title: The usage of "Multiscale" in the title does not seem particularly warranted. My expectation based on the title was that the implementation would be flexible enough to model snow at multiple scales (resolutions) and/or assimilate data at multiple scales (resolutions). It is not clear from the presentation whether either is the case. Or is the meaning meant to convey multiple temporal scales? The authors should consider whether the title should be changed for clarity. If there is an aspect of what you are proposing that is indeed "multiscale" you should emphasize that more for the reader's benefit.*

**Reply:**

We would like to clarify that MuSA can be run at a range of spatial resolutions, as shown in the manuscript from km to m scale. Assimilating multiscale observations is also possible, just that these are usually pre-processed (to the spatial geometry of the model grid) before being fed into MuSA. In terms of temporal scales of assimilation it is also quite flexible since it builds on the concept of data assimilation windows. We nonetheless agree with the reviewer that the term Multiscale can be misleading or at least too narrow and have thus opted for the term Multiple. This change more clearly highlights that the toolbox can assimilate multiple observations at multiple spatiotemporal scales using multiple data assimilation schemes. The name of our toolbox (and thus the title) has thus been changed to *The Multiple Snow Data Assimilation System* while the acronym MuSA remains unchanged.

COMMENT # 1.3

*2. Line 85: In mentioning the "posterior mean snow simulation from FSM2" it would be useful to know what variables that contains. Maybe the FSM2 variables could be shown in a Table?*

**Reply:**

The output of FSM2 contains the entire physical state of the snowpack as represented by this intermediate complexity model. Moreover, in MuSA it is simple to output more variables (such as fluxes) than the ones that are defined by default. This is relevant, as if other variables are selected for being output it will be possible to assimilate them directly. Some modifications are expected in the near future, this is the reason why we did not originally include it in the manuscript (it is in the Github repository). After considering the suggestion of the referee, we have included the following.

**Changes:**

**2. Overview of the data assimilation system**

...

In its current version, MuSA is able to assimilate the following variables:

- SWE (mm)

- Snow depth (m)

- Land surface temperature (K)

- Fractional snow-covered area (-)

- Albedo (-)

- Sensible heat flux to the atmosphere (W m$^{-2}$)

- Latent heat flux to the atmosphere (W m$^{-2}$)"

We expect to provide support for even more variables in the future.

*3. Line 132: In mentioning the additive/multiplicative perturbations there is no description as to whether they are perfectly independent or perfectly correlated or something in between. In other words, do all pixels get the same perturbation (i.e. from the same random number) or do they get fully independent perturbations (i.e. each sampled independently). Mentioning here or elsewhere that this neglects spatially-correlated errors/uncertainties would be appropriate. It is mentioned earlier that the model structure is fully independent, but saying whether the perturbations are as well would clarify the setup.*

**Reply:**

Thanks for the suggestion, we have now clarified this in the text.

**Changes:**

**2.1 Ensemble generation**

...

The perturbation of the forcing is performed by drawing spatially independent random perturbation parameters from a...

*4. Line 413: In describing the time-invariant perturbations it may be worth mentioning what the implications of that are vs. other options (independent in time or correlated in time).*

**Reply:**

The implications of using time-invariant perturbations has been added to the text following the reviewer's suggestion.

**Changes:**

**3.3 Consistency**

...

 As noted by (1), when these perturbation parameters are interpreted as errors this can be viewed as a limiting case of perfect time-correlation where the errors become constant biases. The lack of dynamics in the perturbation parameters is thus

akin to assuming that errors in the forcing are constant in time within a particular water year. This simplifying assumption imposes a longer memory in the system than with jitter and facilitates the propagation of information backwards in time using smoothers. Given this transition density, the full prior for the entire DAW can be factorized as follows...

COMMENT # 1.6

**5.** *Line 654: I think a couple of sentences describing the mechanics of the iterative nature of the method and why it outperforms other methods would be warranted.*

**Reply:**

The mechanics of the iterative nature of the MDA method are now described in Section 3.5.

**Changes:**

**3.5 Ensemble Kalman methods**

...

In this iterative scheme the prior ensemble moves gradually to the posterior ensemble through a tempering procedure (see 2, and references therein). The iterations thus mitigate the impact of the linearity assumption inherent in ensemble Kalman methods (3), typically leading to marked improvements compared to the ES for nonlinear models without the risk of degeneracy associated with particle methods as the curse of dimensionality rears its head (4; 5).

COMMENT # 1.7

**6.** *Lines 696-703: More explanation of what it meant by "inflated observation errors" and how it fits into the method would be helpful to the reader. An elaboration on the note about the "multiple data assimilation approach does not actually violate" would also be helpful. Since this particular method is less standard than others, I assume most readers would benefit from more detail here.*

**Reply:**

We have now thoroughly explained what is meant by inflated observation errors. More generally, we have also elaborated on the idea behind the MDA schemes.

**Changes:**

**3.5 Ensemble Kalman methods**

...

Recall that for $N_a = 1$ we recover the non-iterative stochastic EnKF and ES, while for $N_a > 1$ we are using iterative versions of these schemes that involve multiple data assimilation with inflated observation errors.  The term "multiple data assimilation" refers to the assimilation of the same data multiple times rather than an assimilation of different types of data (joint assimilation). We speak of inflated observation errors since the role of the coefficients $\alpha^{(\ell)}$ is to inflate the observation error covariance $\mathbf{R}$ in the Kalman gain $\mathbf{K}^\ell$ as well as the observation error term $\varepsilon_\alpha^{(\ell)}$. This inflation is tantamount to tempering the likelihood as discussed by  (2) and (6) which explains why these iterative schemes perform better than their non-iterative counterparts for non-linear models in that they involve a more gradual transition from the prior to the posterior. Despite what the name might suggest, this multiple data assimilation approach does not actually violate the consistency of Bayesian inference by using the data more than once due to the way the observation error inflation is constructed, particularly due to the constraint that  $\sum_{\ell=1}^{N_a} 1/\alpha^{(\ell)} = 1$. Simply stated, this constraint ensures consistent results with a linear model since by construction we get the same result by assimilating the data once with the original uninflated ($\alpha = 1$) observation errors as assimilating the same data multiple times with an inflated ($\alpha > 1$) observation errors. With a nonlinear model, practice has shown that these iterations of the ensemble Kalman analysis ensure that the approximate posterior is closer to the true posterior than if a more conventional single uninflated iteration is used (3). It is possible to satisfy  the constraint on the $\alpha^{(\ell)}$ both with uniform and non-uniform inflation coefficients (c.f. 1). For simplicity, following (7), we currently opt for former as a default in MuSA by setting $\alpha^{(\ell)} = N_a$  for all $\ell$ by default while allowing for  non-uniform coefficients as an option.

COMMENT # 1.8

**7.** *Line 741: Can you provide some justification of the choice of four (4) for the number of assimilation cycles?*

**Reply:**

In general, the number of iterations of MDA $N_a$ is a compromise between computational cost, since each iteration adds $N_e$ simulations, and accuracy, since the performance (in terms of approximating the true posterior) improves as $N_a$ increases. For the latter, however, the improvement is usually asymptotic. As such there is usually

an optimal choice of $N_a$ that balances the quality of the posterior ensemble approximation against the cost of the iterative ensemble of model runs. Experience (e.g. 8; 3), has shown that $N_a = 4$ tends to be a satisfactory choice. However, with any particular model this is usually worth exploring in a sensitivity analysis. Since this was not the main topic of this paper, such an exploration was not pursued herein. In principle, $N_a$ can is set by the user and should be viewed as a tuneable hyperparameter by taking the aforementioned compromise into account.

**Changes:**

**4.1 Single cell and distributed assimilation of drone-based snow depth retrievals**
. . .
In the iterative versions of the ensemble Kalman based approaches we fixed the number of assimilation cycles to $N_a = 4$ as a compromise between computational cost and performance. The former is directly proportional to $N_a$ while the latter converges to an optimum as $N_a$ increases. This choice of $N_a$ is also in line with sensitivity analyses performed elsewhere (7; 8; 3). . .

COMMENT # 1.9

*8. Section 4: I would urge that more consistent (and maybe simpler) language be used throughout to refer to the three experiments being done so that readers can follow more easily. It doesn't seem that "drone data" or "satellite data" are as relevant to the first two experiments compared to the first being a spatially-distributed (snow depth) data assimilation experiment and the second being a point-scale joint (FSCA+LST) data assimilation experiment. For the benchmark case, it is not clearly defined what experiment is actually being done. Is it snow depth or LST+FSCA assimilation? There is a mention of "single cell" which may imply it is the same setup as the second experiment, but this is not clear as currently presented.*

**Reply:**

In an effort to homogenize the section titles and make them clearer as suggested by the reviewer, we have changed the section titles. In particular, Section 4.1&5.1 are now named **Single cell and distributed assimilation of drone-based snow depth retrievals** and Section 4.2&5.2 are now named **Joint assimilation of satellite-based LST and FSCA retrievals** In the text, we have also emphasized that the benchmark was conducted using drone data in the Izas catchment.

**Changes:**

**4.3 Computational benchmarks**

. . .

The comparison was performed using 100, 200 and 300 particles and four iterations for the iterative ensemble Kalman approaches in a single cell, assimilating drone snow depth retrievals at a random location in the Izas catchment. The reported values of the benchmarks are the average of 10 MuSA runs, and includes the FSM2 compilation time ($\simeq$2 seconds using the GNU Fortran compiler 10.3.0 in the aforementioned local machine) which is negligible compared with the whole run.

COMMENT # 1.10

*9. Section 4: In an effort to make the sample experiments more reproducible for the readers, I would suggest tabulating any key parameter differences (beyond default values) in the config.py and/or constants.py input files that are specific to each experiment being done. It would also be helpful to connect the individual experiments to the theory provided earlier in the paper, i.e. description of the states, measurement, etc. In particular, if transforms are used with respect to the measurements (as referred to on Lines 359-370), it would be useful to see the form of the those transforms in the experimental setup in Section 4.*

**Reply:**

Thanks for the suggestions. We have added a config.py file to the Zenodo repository with different configuration suggestions. It should be noted that the constants.py file is the same for all experiments and is stored on github. In its current versions the transformations are rather simple, and therefore although we considered adding a new figure to visualize these we thought that might be a bit excessive. Instead, as an example, we have generated this simple example in Python using Google Colab that you can experiment with online:

Simple example of Gaussian anamorphosis (hyperlink)

COMMENT # 1.11

*10. Section 4: Perhaps in each case you can explain what the measurement model is for that experiment, i.e., is it just an internal model state (snowdepth, LST?) or a prescribed diagnostic relationship (FSCA?). In cases where it is a prescribed diagnostic relationship, how is that handled within the framework? I imagine that the current FSCA is built-in to FSM2, but what if an alternative representation was desired. Would that be handled via modification of the FSM2 snow model, or via another method.*

**Reply:**

At the moment, all the variables that can potentially be assimilated are calculated as state variables in FSM2. In the specific case of FSCA, if a different representation is desired, we would recommend implementing it directly in FSM2 as this variable interacts with several components of the surface energy balance in the model. In any case, it is very easy to include new variables derived from FSM2 outputs (e.g. statistics such as snow cover frequency, season length, etc.), and this could be done in MuSA without having to modify the FSM2 code, or with minimal modifications to it in case other internal states of the model are needed.

**Changes:**

**4. Data and experimental setup**

. . .

 Note that all the variables that we assimilate in these experiments are state variables in FSM2. We . . .

COMMENT # 1.12

*11. Section 5: I found the organization of Section 4 flowing into Section 5.1 hard to follow. As mentioned above, I would suggest using the same language to refer to the three experiments throughout to help in this regard. It wasn't clear to my why the benchmark (single cell) and distributed snow depth results were presented together in Section 5.1. They are described as two different experiments in Section 4 and so I think it would be easier to follow if they were treated as such in Section 5. Lumping them together in 5.1 seems a bit disjointed. Or maybe "single cell" here does not refer to the benchmark case (although "single cell" is used in that context too)?*

**Reply:**

As previously mentioned, and following the suggestion of the reviewer, we have changed the names of these sections. "Single cells" simply refers to the fact that MuSA has been launched at a specific single grid cell in the Izas domain. For the drone-based snow depth retrievals the single cell is at a the resolution of 5 m and used to compare algorithms (Sections 4.1&5.1) as well as computational cost (Sections 4.3&5.3). For the joint assimilation of satellite-based FSCA and LST retrievals the single cell is at a much coarser resolution of 1 km as described in (Sections 4.2&5.2). This should be clearer now that we have renamed the sections.

COMMENT # 1.13

*12. With respect to Table 1, it is not clear what the reference data being used to compute RMSE is. It implies snow depth, but the description of what data was assimilated in the benchmark experiments is unclear (see comment above). The notation used for each scheme is also not defined. Perhaps define PF-c, PF-r, in caption?*

**Reply:**

This has been clarified as outlined below. In addition, we have recalculated the metrics in Table 1, since upon reviewing the code we found a bug in the RMSE calculation. In any case, the relative performance of each algorithm has not changed, so the conclusions remain the same.

**Changes:**

**Table 1 caption**
 RMSE for the reference run (Ref), particle filter with bootstrap resampling (PF-b), particle filter with redraw resampling (PF-r), ensemble Kalman filter (EnKF), ensemble Kalman filter with multiple data assimilation (EnKF-MDA), particle batch smoother (PBS), ensemble smoother (ES), and ensemble smoother with multiple data assimilation (ES-MDA). These errors were computed using the assimilated drone-based snow depth observations as the truth and using the posterior ensemble mean  as the  estimate from the respective DA schemes. All the schemes were run with $N_e = 200$ particles and the MDA schemes used $N_a = 4$ iterations…

…

**5.1 Single cell and distributed assimilation of drone-based snow depth retrievals**
The results show how the performance of the different data assimilation algorithms differs, even with the same initial conditions and experimental setup, when the posterior ensembles are compared against the assimilated snow depth observations (Table 1, Figures 3&4)….

COMMENT # 1.14

*13. In the context of Figure 3, it would be helpful to explain the meaning of "MDA" when only snow depth is being assimilated. I believe this method actually differs in this case due to its iterative nature rather than multi-data? This comes into play later where different notation is used to refer to iterative versions of method. Perhaps you can harmonize how you refer to iterative methods across the manuscript.*

**Reply:**

As pointed out in an earlier reply, "Multiple data assimilation" (MDA) is synonymous with iterative in this context and refers to the fact that the these ensemble Kalman with MDA methods are composed of a number of gradual transitions rather than a single abrupt movement of the ensemble. As such, they assimilate the same data multiple times but with a constrained inflation of the observation error to avoid a circular analysis (double dipping) as explained in response to Comment #1.7. This MDA method is applicable regardless of whether one or several variables are assimilated at the same time (joint assimilation). The nature of the assimilated observations does not influence the method per se, so the fact that only snow depth is being assimilated does not have any bearing on the use of MDA.

As suggested by the reviewer, we have now homogenised the nomenclature throughout the text and figures by using multiple data assimilation (MDA) instead of Iterative (I).

Comment # 1.15

**14.** *Figure 3. Refer to which experiment this corresponds to. And is this a particular cell? Is it the one shown in Figure 2?*

**Reply:**

Yes, it is in the particular cell highlighted in Figure 2. We have clarified this by adding the following in Figures 3&4:

**Changes:**

**Figure 3&4 captions**
. . . in the single cell of the Izas catchment highlighted in Figure 2. . .

Comment # 1.16

**15.** *Line 797: The reader would benefit from more description of how the prior forcing perturbations are generated in this context and how the posterior emerges from that. Can you clarify whether prior was identical across space and why patterns in the posterior emerge. Is there anything to be learned from the posterior uncertainty of these, i.e. is one more certain than the other (i.e. precip. vs. temperature? And why are the posterior patterns between the two fields seemingly so highly correlated. More discussion either here or in Section 6 would benefit the reader.*

**Reply:**

This is discussed below, where we hypothesise that the spatial patterns emerge from the fact that in the reference simulation there is no implicit representation of wind redistribution. That redistribution is implicitly induced in the simulations by assimilating the drone snow depth maps, allowing MuSA to generate these spatial patterns. It is difficult to estimate, and outside the scope of this paper, whether one parameter is more uncertain than the other. It is likely that there is some equifinality as well. This problem could be addressed by jointly assimilating more variables related to the energy balance such as LST and albedo. In any case, such a study would require specific work to be fully relevant, in the same way that we try not to dwell on the intercomparison between algorithms as these are topics that deserve much more attention than is given here. Here we try to limit ourselves to describing the capabilities of MuSA by means of examples that may be of interest, and hopefully a source of inspiration, to future users.

COMMENT # 1.17

**16.** *Discussion associated with Figure 6. Indicate that the fields in Figure 6 are the posterior mean. Units should be associated with temperature. Could more discussion be provided to hypothesize why the patterns are what they show.*

**Reply:**

We have added the units to the caption, thanks for the suggestion. As for the spatial patterns, in the manuscript (see **6. Discussion**) we hypothesise that it is a consequence of the wind redistribution patterns that are not explicitly represented in FSM2 but can be captured implicitly by perturbing the forcing. For example, the precipitation bias perturbation parameter effectively accounts for both biases in the large scale precipitation field as well as the local effects of wind redistribution of snow that occurs mainly during the accumulation season. But to broaden the discussion in this sense would be somewhat speculative and, we believe, outside the scope of this paper.

COMMENT # 1.18

**17.** *Figure 7: There are inconsistencies (and typos.) between the use of what should be "LST" in the caption and "SST" in the figure. Is SST meant to be "snow surface temperature". If that is preferred, SST should be used throughout instead of LST. The acronym "IKS" should be defined in the caption.*

**Reply:**

We have corrected these acronyms such that they are consistent throughout the manuscript, they are now always LST and ES-MDA. Thank you for pointing this out.

*18. Figure 8: Acronyms need to be defined in the caption and reconciled with earlier ones. How does the Ensemble Smoother – MDA compare to any of these? Is it the same as I-ES?*

**Reply:**

These acronyms have now been homogenised in the text and figures, thank you.

*19. Line 869: It is not clear what is meant by: "The assimilation of the FSCA provides information when FSCA saturates at 1, . . . ". Should this read ". . . does not provide information"?*

**Reply:**

The first use of FSCA in this sentence was a typo, it should have said LST. It has now been corrected.

**Changes:**

**6. Discussion**

. . .

The assimilation of  LST has the potential to provide additional information when FSCA saturates at 1,  for example during most of the  accumulation season and during the polar night in the absence of sunlight.

*20. Code and data availability: It seems that the MuSA code from the original github repository vs. the version provided on zenodo are different. In particular, when run on a mac, the github version worked, while the zenodo version did not. It appears to center on differences in the code, where the latter crashed out when finding the OS to be 'darwin' (macOS) instead of 'linux'. I suggest making sure to reconcile the two so that the one posted on zenodo works. It would also be helpful for reproducing the results to 1) tabulate key parameters specific to each experiment (as suggested above) and 2) providing the actual input files for each experiment with the code distribution. This would make it much easier to reproduce the results from the paper and extend the framework to other cases rather than having to interpret which*

*parameters to change.*

**Reply:**

This error comes as a surprises to us, and we have not been able to find where it comes from. It is true that MuSA checks the operating system, and is not to be used if darwin (macOS) or linux is not identified. Note that for Windows users we have now tried MuSA successfully using the Windows Subsystem for Linux (WSL). The OS checking function is identical in the Zenodo and Github version, which makes sense as the copying is done automatically since both repositories are connected. In any case, small differences between Zenodo and Github will always occur, this is expected behaviour. The goal of using Zenodo is to be able to provide a given release with a unique DOI. But all recent activity on Github that isn't part of a release will be out of sync with Zenodo, this is unavoidable. We have added the LST and FSCA observations from MODIS to the repository as suggested, as well as a tabulated configuration file.

COMMENT # 1.22

*Technical Corrections:*
*This is not an exhaustive list of typos, but ones that jumped out:*
*1. In Figure 1 there is a typo., where "weigths" should instead be "weights".*

**Reply:**

Corrected, thanks.

COMMENT # 1.23

*2. Line 205: Typo. in the phrase "the are usually".*

**Reply:**

We have corrected the typo and restructured the sentence for clarity.

**Changes:**

**3.1 Bayesian inference**
... These are usually probability density rather than mass functions as we tend to deal with continuous variables in DA

*3. Line 383: Typo. in the phrase "we will let denote anamorphosed".*

**Reply:**

This has been corrected, a **u** was missing

**Changes:**

**3.3. Consistency**
. . .
As such, we will let $\underset{\sim}{u}$ denote anamorphosed parameters that have undergone a forward transform to the unbounded space.

*4. Line 391: "SM2" should be "FSM2".*

**Reply:**

Corrected.

*5. Line 491: "converege" should be "converge".*

**Reply:**

Corrected.

*6. Line 790: "smothers" should be "smoothers".*

**Reply:**

Corrected.

**REFERENCES**

[1] G. Evensen, "Accounting for model errors in iterative ensemble smoothers," *Computational Geosciences*, vol. 23, p. 761–775, 2019.

[2] A. S. Stordal and A. H. Elsheikh, "Iterative ensemble smoothers in the annealed importance sampling framework," *Advances in Water Resources*, vol. 86, p. 231–239, 2015.

[3] G. Evensen, "Analysis of iterative ensemble smoothers for solving inverse problems," *Computational Geosciences*, vol. 22, p. 885–908, 2018.

[4] C. Snyder, T. Bengtsson, P. Bickel, and J. Anderson, "Obstacles to High-Dimensional Particle Filtering," *Monthly Weather Review*, vol. 136, p. 4629–4640, 2008.

[5] N. Pirk, K. Aalstad, S. Westermann, A. Vatne, A. van Hove, L. Tallaksen, M. Cassiani, and G. Katul, "Inferring surface energy fluxes using drone data assimilation in large eddy simulations," *Atmospheric Measurement Techniques Discussions*, 2022.

[6] P. J. van Leeuwen, H. R. Künsch, L. Nerger, R. Potthast, and S. Reich, "Particle filters for high-dimensional geoscience applications: A review," *Quarterly Journal of the Royal Meteorological Society*, vol. 145, p. 2335–2365, 2019.

[7] A. A. Emerick and A. C. Reynolds, "Ensemble smoother with multiple data assimilation," *Computers & Geosciences*, vol. 55, p. 3–15, 2013.

[8] K. Aalstad, S. Westermann, T. V. Schuler, J. Boike, and L. Bertino, "Ensemble-based assimilation of fractional snow-covered area satellite retrievals to estimate the snow distribution at Arctic sites," *The Cryosphere*, vol. 12, p. 247–270, 2018.

---

## Author Comment (AC2)

COMMENT # 2.1

*General Comments*

*In this paper, the authors developed MuSA (v1.0), a standalone snow data assimilation system. MuSA provides a comprehensive framework encapsulating FSM2 (a widespread open-source intermediate complexity snowpack model) with six variants of ensemble bayesian data assimilation algorithms revolving around the Ensemble Kalman Filter (EnKF), the Particle Filter (PF) and their smoothing counterparts. As such, this framework is a great contribution to the snow data assimilation community, as it allows to seamlessly compare different assimilation strategies. It also seems accessible as an educational tool and outreach. Furthermore, the literature only offers data assimilation implementations that are strongly bound to operational constraints (such as country-specific numerical weather prediction models or high-performance computing infrastructures), and are therefore not transferable in space, while MuSA could conceptually be used anywhere, anytime, something which is definitely missing at the moment. Indeed, it is sometimes virtually impossible to compare the performance of algorithms produced by different teams worldwide. The authors present the potential of MuSA in two different assimilation experiments: assimilating drone-base high-resolution maps over a small catchment in the Pyrenees (with a focus on a single "pixel"), and a combination of snow cover fraction and Land Surface temperature from MODIS, which allow to exhibit the different behaviour of the respective algorithms over two snow seasons and make a good entry point for new users.*

**Reply:**

The authors would like to thank Bertrand Cluzet for his positive and constructive comments on our work. We hope that the MuSA tool will be useful in enabling a wider audience within the snow science community to conduct their own experiments in the field of data assimilation.

COMMENT # 2.2

*The data assimilation algorithms implemented in MuSA are all already available in the literature. Only a tiny level of novelty resides in the use of the heuristic "redrawing from a normal approximation of the posterior" which is introduced in l. 99-107 to fight PF degeneracy but would deserve more stance and details. The idea is similar to approaches where the perturbation parameters are not resampled, only resampling the model states (Cluzet et al., 2021), or where perturbation are added to the posterior resampled parameters (Piazzi et al., 2018).*
*Then comes a very long Section 3 (17 pages) of theoretical developments and digressions*

*which is rather ambitious as it tries to bring all the algorithms together into the framework of Bayesian inference almost in a review-like style. This comes at the cost of lengthy digressions and theoretical developments that have little interest in explaining the interest of MuSA and the presented results but have the merit of putting everything together in one single piece of literature, which will be really appreciated by the community outside of data assimilation experts and is in the spirit of GMD. I strongly recommend the authors to try and make this section more concise and accessible to such a public and am pleased to provide some suggestions in the attached annotated manuscript.*

**Reply:**

The strength of MuSA lies in its modularity. Its design facilitates the implementation of new algorithms with different capabilities. We believe that for the first version, it is sufficient to implement the algorithms that the community has already experimented with. Note, however, that the iterative versions of ES and EnKF have yet to receive widespread adoption for snow DA (other than 1) despite their demonstrated strong potential in other fields (2; 3; 4). In any case, the implementation of several new algorithms in the near future is already in the pipeline, with several prototypes under development. Following the MuSA philosophy, we consider that Section 3 with the theoretical basis is necessary. We are committed to a demystification and generalisation of the use of data assimilation techniques within the cryospheric sciences. To the best of our knowledge, there are not many similar exercises that attempt to explain the functioning and basic theory of the different DA algorithms that are in common use in snow science. This gap has made the implementation of these techniques difficult for groups that are unfamiliar with them. Moreover, we consider it appropriate to introduce how a new tool works, and not simply refer to a large number of references where explanations of the underlying theory and individual algorithmic implementations are disseminated. We nonetheless agree that the section can be reduced somewhat, and we appreciate the reviewer's recommendations. The reviewer's comments have made this reduction exercise easier, and this part is now more concise and clearer.

COMMENT # 2.3

*The results represent a nice illustration of the capabilities of MuSA. They have a limited level of scientific novelty and lack methodological depth for any significant conclusion to be drawn. At the light of this consideration, I recommend the authors to temper or even remove most statements regarding the conclusions drawn from their results in the discussions (see the notes below and the attached PDF). I don't think, for instance, that this study is substantiated enough to discuss the relative advantages of EnKF vs. PF approaches, but it definitely lays a*

*nice context to do so in further work. In this spirit, I recommend reorganizing their discussion with the aim of highlighting the novelty and versatility of the method, detailing potential use cases including educational purposes. More references would also be needed in that section. I think that this would ultimately sound more GMD-like and would increase the impact of the paper. The code is well documented and seems written in a neat way that will guarantee its accessibility to the community and beyond. Overall, I think that this paper is relevant for publication in GMD, as it provides a significantly new modelling tool to the snow data assimilation community and beyond. I am confident that the authors can address the minor recommendations detailed above. Below are listed some technical points. Please also make sure to address all the comments and suggestions in the attached commented manuscript.*

**Reply:**

Once again, we appreciate the positive comments. We have restructured some parts following the reviewer's recommendations. Below, we respond in detail one by one to all the reviewer's comments and suggestions. This response includes a discussion of whether or not a comparison of ensemble Kalman and particle approaches is warranted given our results.

COMMENT # 2.4

***Technical notes***
*Several code performance/implementation features are disseminated throughout the manuscript. This is definitively an important added value from MuSA, that would deserve more stance. Consider grouping them?*

**Reply:**

We have modified some statements concerning the performance and implementation in line with the comments annotated by the reviewer in the attached PDF. These features are now clearer. However, we consider that they are in their right place to link theory and experiments with the actual implementation, so we prefer not to group them together.

COMMENT # 2.5

*- l. 84: a thorough description of the direct insertion is required, as this is not as trivial as it seems: when assimilating HS, are you adding/removing mass, or do you just squeeze the snowpack layers? How do you handle the relayering?*

**Reply:**

After discussion among the authors, we have opted to remove the direct insertion. We are not convinced that this feature is in line with the MuSA philosophy. Furthermore, due to the necessary assumptions, and the problems also highlighted by the reviewer, we cannot ensure that this feature will be supported in the future, especially if new numerical models are eventually inserted into MuSA.

COMMENT # 2.6

*125-135: please improve the style, this could be written in a more concise/impacting way.*

**Reply:**

We have followed the reviewer's suggestion and made the following changes

**Changes:**

**2.1 Ensemble generation**

 The DA algorithms implemented in MuSA all require a prior ensemble  of simulations to represent uncertainty. The number of ensemble members  (or, equivalently, particles; 6), which we denote by $N_e$, should be specified by the user, as it drastically affects both the computational cost of the experiments and the performance of the data assimilation algorithms. To generate an ensemble of snowpack state trajectories, MuSA perturbs the  meteorological forcing to run an ensemble of FSM2 simulations. An arbitrary number of forcing variables can be perturbed. The perturbation of the forcing is performed by drawing spatially independent random perturbation parameters from a normal distribution  or log-normal  distribution. The prior standard deviation and mean of  these distributions should be specified by the user. MuSA supports normal or log-normal probability functions to generate  additive or multiplicative perturbations respectively, depending on the physical bounds of the variable to be perturbed.  Additive perturbations are typically used for air temperature. Multiplicative perturbations are recommended for precipitation to avoid negative values  as well as for shortwave radiation to avoid negative values and positive night-time values.. . .

COMMENT # 2.7

*157-163: same comment.*

**Reply:**

These lines have been changed as shown below.

**Changes:**

**2.2 Meteorological forcing**

...

 Each grid cell is solved independently  which includes the reading of the forcing  that occurs along the time dimension. Otherwise, each process would have to store considerably more data in memory leading to more costly I/O operations that would slow down the run time. Even so, just reading along the time dimension can come with a considerable computational cost if the time dimension is large. To alleviate this, the  time spent reading the forcing can be reduced by setting the chunk (a subset of the file to be read or written as a single I/O operation) of the netCDF forcing files along the time dimension. To speed up the subsequent relaunching of the simulations when smoothing and filtering, MuSA generates an intermediate binary file with the forcing information  needed to run a complete simulation  for each grid cell.

COMMENT # 2.8

*168-171: While assuming independent observation errors seems a reasonable assumption, attributing the same value for all observations of a given variable does not seem state of the art (take the example of the snow depth model error of e.g. Magnusson et al., 2014). Consider commenting on that since this is not much of a technical hurdle for further developments.*

**Reply:**

This is a thought provoking comment. Firstly, it is clear that the formulation here was somewhat imprecise, in that MuSA already offers the option to account for different observation error variances in the case that we have multiple observations of the same variable from different sensors. We have now clarified this. Secondly, although the example of (7) is laudable we would not consider this state of the art. In fact, although they try to account for heteroskedastic errors in an EnKF system (e.g.

their equation 18) their implementation is heuristic and violates the underlying theory. Instead the correct way of adding heteroskedasticity, and more generally adding sophistication to the observation error model, in DA is to modify the entire likelihood instead of just **R** directly (e.g. 8; 9). Since, to the best of our knowledge, this has yet to be pursued in snow DA it is beyond the scope of this work. It is certainly an important topic that is worthy of further pursuit. We have added a brief discussion of this in the relevant section as outlined below.

**Changes:**

**2.3 Observations and masked cells**

…

A temporally and spatially constant scalar corresponding to the assumed observation error variance must be provided for each  type of observation that is to be assimilated. This assumption implies a diagonal observation error covariance matrix, **R**, which is tantamount to assuming that observation errors are uncorrelated in both time and space. Note that this formulation allows the user to account for differences in observation error that arise in the case when a variable is observed by multiple sensors with varying accuracy. By modifying the likelihood, it would also possible to account for non-Gaussian observation errors (8; 9), but this is not yet supported in MuSA…

COMMENT # 2.9

*- l. 172-174: while there is no conceptual shortcoming in the case of smoothers, please give more implementation details as to how time-irregular observations are assimilated in the sequential filter algorithms: are they assimilated 'on the fly', or grouped by daily/hourly batches? For instance, I can imagine some IO bottlenecks or PF degeneracy problems in the situation where an observation would come every 15mins and would be assimilated on the fly.*

**Reply:**

Any time an observation (or multiple observations at the same time) occurs, the analysis is performed. It is true that if the temporal resolution of the observations is extremely high, I/O bottlenecks will be generated in the case of filters, but this is expected behaviour. Future developments with modified versions of FSM2 or other models may help to solve this problem. In any case in real scenarios this problem would only occur when assimilating data from weather stations or (less commonly) geostationary satellites and in this case the user will have to pre-process the products

as MuSA will not do it for them.

**COMMENT # 2.10**

*494-512: could be removed*

**Reply:**

We prefer to keep this section as this importance sampling theory is at the core of the particle methods. Were we to remove this then the same logic would lead us to remove most of Section 3. We would like to avoid this since we believe this section provides a valuable overview of the theory that is missing in many snow DA papers. Indeed, the reviewer also acknowledges that this section will be appreciated and is in the spirit of GMD. In summary, we have done our best to remove text elsewhere, but this paragraph is too important to cut.

**COMMENT # 2.11**

*577-579: Numerical instabilities: please expand as the reference is not open source. And what if several/all particles obtain likelihoods below the machine precision? How do you ensure that the likelihoods are still "sorted" in a proper way?*

**Reply:**

This reference (10) is actually open access (and the code is open source) since it is freely available via probml.ai which is also linked on the book's official MIT press webpage under "Read open access". We have now also included a link to the book in the bibtex reference to make it easier to discover the open access PDF. The problem that the reviewer is alluding to is exactly the problem that we are addressing via the so-called "log-sum-exp" trick described in Section 2.5.4 in (10). In terms of the code, the implementation of this "trick" is relatively straightforwards and it is included under modules/filters.py on the MuSA Github page in the pbs function. This seems to be the standard way of addressing this numerical problem, as an equivalent solution is also discussed on page 102 of (11) which is also open access via the book's Springer webpage. The reference to (11) has now been added to this sentence and we have tried to clarify the goal of this solution.

**Changes:**

**3.4 Particle methods**

...In practice, to ensure numerical stability, we first compute the natural logarithm of the weights by using the log-sum-exp trick to avoid potential overflow (10)and

COMMENT # 2.12

*4: consider enclosing each individual algorithm into a numbered algorithm block*

**Reply:**

Thanks for this nice suggestion, we have now added such algorithm blocks in Section 3 for the PF (Algorithm 1), PBS (Algorithm 2), and the Ensemble Kalman analysis with MDA (Algorithm 3). Note that the bullet points containing a mixture of text and equations within these blocks that describe these algorithms remain largely unchanged. For clarity and conciseness, we chose not to go with full pseudocode and retained most of the text, otherwise these blocks would be nearly double the size with nested for loops and if statements and the like. We have also added some more brief information about the techniques involved to ensure numerical stability, i.e. the log-sum-exp trick (10) for the particle schemes and the pseudo inversion (12) for the ensemble Kalman schemes. Curious readers should then be able to identify these techniques within the respective functions under the module modules/filters.py in the MuSA repository

COMMENT # 2.13

*3.5: consider condensing the bibliography on the EnKF*

**Reply:**

The bibliographic part on the use of ensemble Kalman methods in snow DA in Section 3.5 is about the same length as that for the particle methods (especially after adding new references suggested by the reviewer) in Section 3.4 which reflects their roughly equal use in the literature. While it is true that ensemble Kalman methods are perhaps less popular in snow DA than they were historically, i.e. before the adoption of particle methods, we do not see why this would warrant a shorter bibliography on these methods. As for the references from the wider field of DA, it is natural that there are slightly more references for the ensemble Kalman methods since here we introduce for 4 different algorithms (EnKF, EnKF-MDA, ES, ES-MDA) rather than just 2 (PF, PBS) for the particle methods. As such, we could not justify cutting the bibliography on the EnKF, given the already extensive bibliography in the remainder of the paper.

*6: In the first section (l. 820-840), the authors consider that the literature often lacks sufficient justification for their DA algorithmic choice, arguing that EnKF-based algorithms (both in the filtering and smoothing fashion), have been overlooked by the community for bad reasons (l. 827-828) ("due to the Gaussian linear assumption"). This statement is hazardous, and in the case of Largeron et al., (2020), just false. In Sec. 4.3 of Largeron et al., (2020), they list several additional advantages of PF vs. EnKF in the case of detailed snowpack models. Piazzi et al., (2018) as well as Cantet et al., (2019) also discuss the advantages of PF vs. EnKF in similar terms. These reasons may be bad or weak, but if the authors want to contradict them, they should be cited and contradicted extensively, otherwise this discussion is just not fair. Nevertheless, while I consider that the authors might be right, even with more literature substance I am not sure that this fits in the scope of this paper, and I would rather see this in a following, dedicated publication.*

**Reply:**

We have now weakened the statement concerning EnKF-based schemes in snow DA somewhat and tried to clarify the points that we are trying to make about actually comparing (empirically) the performance of DA algorithms. We were surprised to hear that our statements are hazardous or false. A case could be made that the situation is completely the opposite. In fact, no EnKF was run in any of the citations the reviewer provides where "the advantages of PF vs. EnKF" are discussed. This is precisely our point, not that one algorithm is superior to another but that one generally wouldn't know this without checking because it would depend on the problem at hand. While we do believe that many of the reasons raised in the references the reviewer provides can be misleading, in that they cut against our experience and (from what we can tell) that of the wider DA community, we agree that this is not the topic of this paper and have thus not pursued this further herein.

**Changes:**

**6. Discussion**
The results from the intercomparison of different data assimilation experiments exhibited a large variability in performance and computational cost across the schemes. This highlights the need for thorough testing, as the suitability of the assimilation algorithms will vary depending on the problem at hand. Despite the fact that  most of the data assimilation algorithms improved the simulations compared with the  reference simulations, their performance differed markedly. In fact,  these differences tend not to be explored in the literature, where often the choice of one algorithm over others is not sufficiently

 justified empirically. The lack of tools to compare the performance of different data assimilation algorithms has probably contributed to this problem, since it requires substantial coding effort to implement all available options in each data assimilation experiment. As an example, a case could be made that ensemble Kalman-based data assimilation approaches are often prematurely perceived as suboptimal for snow science applications due to the Gaussian linear assumption (13; 14). In spite of this, our first experiment showed that the iterative version of ensemble-based Kalman smoothers outperformed the other smoother algorithms, as also found by  (1) and (15). These findings are consistent with the broader DA literature, where basic  PFs (based on SIR) tend to suffer more from degeneracy due to the curse of dimensionality  in high dimensional problems (16). This degeneracy problem tends to be even worse with the PBS (based on SIS) due to the absence of resampling (17; 6; 18; 15).

In their review, (5) suggest promising remedies to this problem with more sophisticated particle methods that invoke  innovations such as using  better proposal distributions or iterations. The particle filter using redraws from a normal approximation to the posterior that we implemented in MuSA was loosely inspired by the use of proposal distributions and can overcome the degeneracy problems in the  classic bootstrap particle filter (see Table 1). The use of iterations was pursued by (3) who recast the ES-MDA in the framework of iterative particle methods, leading to improved performance on subsurface flow problems. This suggests that a hybridization of ensemble Kalman and particle methods (19; 15) may also be a promising avenue for future work. We reiterate that we are not advocating for one class of DA methods over another, the point we are making is rather that to do so prematurely would be unwise. Common wisdom embodied in the ´No Free Lunch´ theorems for optimization (20) warns us not to expect one particular algorithm to always perform best, but rather that this will depend on the problem at hand. The power of MuSA is  to provide a tool that simplifies the task of comparing different experimental set-ups by allowing the intercomparison of data assimilation algorithms as well as facilitating the implementation of new ones. In the future,  more data assimilation algorithms  could be implemented, including iterative versions of the PBS and particle filters as well MCMC methods which are the gold standard for Bayesian inference (21; 22), but have received relatively  less attention from the snow community (23) due to their often prohibitive computational cost.

COMMENT # 2.15

*I also find that that section is very badly structured, jumping back and forth between filters and smoothers. Snyder (2008) only treats the question of Particle filtering. While I agree that their statements can be transposed to particle smoothers, it is necessary to acknowledge that jump. More generally, discussing the advantages of smoothers vs. filters does not seem very interesting, as they are different solutions for different problems: reanalysis and operational forecast, respectively.*

**Reply:**

Thanks for pointing out the jump we were making with the (16) reference. We have now corrected this by including a discussion of the degeneracy problem for both the PF and the PBS, while also adding relevant references for the latter. We are not sure exactly which discussion of smoothers vs. filters that is uninteresting because we could not identify such a discussion in this section. When it comes to this discussion more generally, however, it is certainly not as simple as smoothers being for reanalysis and filters being for (initializing) operational forecasts. Instead, the reason for filtering over smoothing is almost always related to computational cost and time constraints. Crucially, filtering can be seen as a special case of the more general (not just batch) smoothing problem where only the latter allows information to propagate backwards in time (c.f. 18). As such, smoother-type algorithms could certainly be used with great effect for forecasting since they would undoubtedly provide superior initial conditions than filters, especially in land data assimilation problems with long memory (24).

**Changes:**

**6. Discussion**

…These findings are consistent with the broader DA literature, where basic  PFs (based on SIR) tend to suffer more from degeneracy due to the curse of dimensionality  in high dimensional problems (16). This degeneracy problem tends to be even worse with the PBS (based on SIS) due to the absence of resampling (17; 6; 18; 15).

COMMENT # 2.16

*848-849: "The assimilation of snow depth products has been shown to be a very robust approach to SWE estimation." This statement is not substantiated by any results from the paper.*

**Reply:**

It is true that we are validating the posterior snow depth (and not SWE directly) estimates, so the sentence has been modified accordingly.

**Changes:**

**6. Discussion**

...MuSA was also able to assimilate hyper-resolution snow depth maps in a distributed fashion. The assimilation of snow depth products has been shown to be a very robust approach to  snow depth estimation....

COMMENT # 2.17

*L849-851: "The posterior maps of the perturbation parameters showed intricately detailed and physically-consistent spatial patterns, especially considering the fact that there is not any cell 850 intercommunication in MuSA". Please use another term than "physically consistent": although that's the best we can do, it is not "physically consistent" to compensate an unrepresented process (wind drift) by other processes (precipitation and temperature biases).*

**Reply:**

We agree and have removed "physically consistent".

**Changes:**

**6. Discussion**

...The posterior maps of the perturbation parameters showed intricately detailed  spatial patterns, especially considering the fact that there is not any cell intercommunication in MuSA. The appearance of this  spatial pattern in the perturbation parameters indicates that they are compensating for a missing processes in the model, which in this case is most likely to be the wind-driven ablation and accumulation processes since most of the spatial variability in melt energy is provided by MicroMet shortwave radiation routine (e.g. 25).

COMMENT # 2.18

*854: "Thus, MuSA can be used to study the importance of missing snow processes in the FSM2 model". When looking closer at Fig. 6, the acute reader can realize that the assimilation is not only compensating for unrepresented wind redistribution, but also from strong precipitation and temperature biases coming from the downscaled ERA input: is it really possible to disentangle "missing snow processes" from input errors in such a setting? This is*

*highly debatable. And a reference to Günther et al., (2019) would be required here.*

**Reply:**

Equifinality is a well known problem in data assimilation exercises in particular and more generally in any numerical modelling. However, in this context, it should be considered as a feature, not a bug, since it tells us that we need more information from observations (i.e. a higher number of observations, more accurate observations, and/or different types of observations), more complex models that include these processes, and/or better forcing data to disentangle missing processes from input errors. The point here is not that we can directly disentangle these with MuSA. In fact we are not claiming this in the manuscript. Instead, the point we are making is that tools like MuSA are a step in the right direction in that the output spatio-temporally continuous SWE and snow depth fields can (even if they are 'right for the wrong reasons') be used to validate or calibrate more snow complex models that try to explicitly capture these processes.

COMMENT # 2.19

*Figures*

*Fig3: In l 397-400, the authors assume that the snow can be reset to zero during the summer (seasonal snow). Their theoretical developments in the corresponding section seem to rely on this assumption. However, in the top-right and top-bottom panels of Fig3., this is clearly not the case, with persistent snowpack through the 2019 summer. Can the authors comment on that?*

**Reply:**

Our choice of wording here was unfortunate as the seasonal snowpack assumption is not actually made in MuSA itself we merely included it in this section to simplify the presentation of the theory. This has now been clarified. Moreover, much of the persistence of the snowpack in Figure 3 for the ensemble Kalman methods was related to a numerical problem that can occur with the matrix inversion in the Kalman gain. This has now been remedied following the method outlined in Appendix A in (12) which is now mentioned when the ensemble Kalman methods are presented in the new manuscript around Algorithm 3.

**Changes:**

**3.3. Consistency**
...  Without loss of generality and to simplify the presentation of the theory, we assume that we are dealing with seasonal snow, so that the initial prior for

the internal  states, $p(\mathbf{v}_0)$, is known $\mathbf{v}_0 = \mathbf{v}_0$ . Thus $p(\mathbf{v}_0) = \delta(\mathbf{v}_0 - \mathbf{v}_0)$ since the annual integration period starts at the beginning of the water year where we assume that a snowpack has not yet formed so internal  state variables are either $0$ (snow depth, SWE) or undefined (e.g. snow surface temperature).

COMMENT # 2.20

*4: the "spaghetti" representation of ensemble members with imbalanced weights must account for the weight of the ensemble in the transparency parameter. Otherwise, this visually gives "weight" to particle with negligible weight and can lead to very misleading conclusions. For example, here the PBS could be completely degenerate, with only one scenario carrying all the weights (which is actually the case, we see in the 2019 summer that the mean drops to strictly zero (without any "tail" as with e.g. the ES-MDA) while some members have still non-negligible snow amounts. Alternatively, one can only display ensemble statistics (mean, quantiles) in such a situation.*

**Reply:**

Thanks for this helpful suggestion. It is correct that our previous spaghetti representation did not account for the weights of the particles which made them misleading. To remedy this, we now instead plot the posterior mean $\pm 1$ standard deviation for all the DA schemes in Figure 3&4.

COMMENT # 2.21

*Fig4: on the bottom-right panel, it is difficult to distinguish the timeseries at 100% zoom factor.*

**Reply:**

The visibility of this panel has been improved substantially by adjusting the aspect ratio of these subplots.

COMMENT # 2.22

*Fig.5: (spatial validation with independent observation) We are missing crucial pieces of information here: what are the precise dates of the assimilated maps and when is the independent validation date? It obviously makes a big difference in the expected performance, especially*

*with a smoother approach, whether the validation date is "closely surrounded" by assimilated observations or not.*

**Reply:**

Thanks for the suggestion, we have added the date of the observation to the caption

COMMENT # 2.23

*Fig 5: the scale is not consistent with the scale in Fig. 2B*

**Reply:**

This has been corrected so that the scale is consistent in both Figures.

COMMENT # 2.24

*Fig 6: Please use fair color scales for temperature and precipitation. In saturation/greyscale, the +1.5 K values should be indistinguishable from the -1.5K values which is not the case here. If you'd use such a fair color scale, you'd make the strong positive bias in the prior much more obvious. A significant prior underestimation of the precipitation would also be visible. Please also comment on both in the results section.*

**Reply:**

Thanks for the suggestion, we have modified the color scale of both panels accordingly.

COMMENT # 2.25

*Fig 7: please introduce month/year in the time axis labels. For the sake of interpretation, I would put the SWE timeseries panel below the SST and FSCA panels, and with the same horizontal extent.*

**Reply:**

We have now included the month and year on the time axis labels for the timeseries in Figure 7.

COMMENT # 2.26

*Code and Data policy*
*No statement on the drone observations, I think. Adding MODIS data to the zenodo archive is necessary to facilitate full reproducibility of assimilation experiments*

**Reply:**

The MODIS data have been added to the Zenodo archive.

COMMENT # 2.27

*A statement should be made regarding the availability of the Micromet code which was used to downscale ERA forcings.*

**Reply:**

We have added the following sentence:

**Changes:**

MicroMet code is available upon request to G.E. Liston. Note that this particular downscaling routine is not central to MuSA, and in theory a myriad of different approaches could be used to generate the forcing data.

COMMENT # 2.28

*Portability: state somewhere (as in the github repo) MuSA does not support Windows yet, but that it seems definitely possible.*

**Reply:**

MuSA can not be run directly in Windows, but it should be possible to generate a Windows version. However, this is not a priority, especially due to the fact that it can be run effortlessly (already tested) thanks to the Windows Subsystem for Linux (WSL) which is integrated in all modern Windows versions. This information has now been added to the *Code and data availability* section.

**Annotations in the attached PDF**

(we have added the approximate line numbers from the original manuscript)

COMMENT # 2.29

*L3: Not a native speaker, but this sounds a bit awkward*

**Reply:**

It is a relatively common expression, there are many indexed articles that use it (even in the title).

COMMENT # 2.30

*L4: I see your point, it makes sense, but the logic is missing here. I'd recommend rather introducing the need for opensource/community oriented tools to mutualize dev efforts*

**Reply:**

Thanks for the suggestion, we made the following changes.

**Changes:**

**Abstract**
... However, to date there is no standalone, open source  community-driven project dedicated to snow data assimilation which makes it difficult to compare existing algorithms, and fragments development efforts. Here we introduce a new data assimilation toolbox, the  Multiple Snow Data Assimilation System (MuSA), to help fill this gap.

COMMENT # 2.31

*L26: Try to find a better formulation, revolving around the spatial variability of the snowpack*

**Reply:**

Changed as follows:

**Changes:**

**1. Introduction**
... In addition, due to the  spatio-temporal variability of the snowpack (26), even dense monitoring networks may suffer from a lack of representativeness (27; 28)...

COMMENT # 2.32

*L27: this question has been revisited in Largeron et al., Frontiers (2020) and my latest paper Cluzet et al., TC, (2022), consider*

**Reply:**

As shown in the response above, we have added a reference to your recent study (28) (but not (14)) since it fits well here. As a side note, we are not sure if we should consider the problem solved with a leave-one-out approach even if it is a promising step in the right direction.

COMMENT # 2.33

*L32: not clear to which variables does this proposition applies, please clarify/reformulate*

**Reply:**

This proposition was deleted.

COMMENT # 2.34

*L35: What about Lievens et al., TC, (2022), using S1 radar backscatter to retrieve snow depth?*

**Reply:**

The radar backscatter approach to snow depth retrieval is certainly promising, but here we were specifically referring to the retrieval of SWE. For snow depth there are several promising options that are emerging in the literature that we cite in the manuscript.

COMMENT # 2.35

*L38: These references are appropriate, but not seminal consider citing Crocus/ SNOWPACK seminal papers on that*

**Reply:**

While we agree that those are seminal papers on complex multilayered snowpack modeling, here we are specifically referring to the task of explicitly (i.e. not semi-distributed) modeling of the spatiotemporal distribution of SWE in a snow hydrology context. For that, we believe, the references we provided are more appropriate given the intermediate complexity of these models (FSM, SnowModel) which allows them

to be more easily run at scale.

COMMENT # 2.36

*L38: Same here*

**Reply:**

In the absence of better suggestions, these seem to be appropriate reference given that they are some of the first studies using high-resolution atmospheric models (WRF in both cases) to help model the snowpack at large scales.

COMMENT # 2.37

*61: Unless I missed something no snowpack model uncertainty is accounted for in the present for pf MuSA, please remove.*

**Reply:**

It is correct that the current version of MuSA only explicitly accounts for forcing uncertainty and not uncertainties that are internal to the model. Of course, some of that forcing uncertainty helps to implicitly account for unresolved processes in the model such as snow redistribution by wind. Nonetheless, to keep matters simple, here we removed this as the reviewer suggests.

COMMENT # 2.38

*L69: This formulation is a bit awkward, consider rewriting*

**Reply:**

Reformulated as follows:

**Changes:**

**2. Overview of the data assimilation system**
…
FSM2 has  several options for the parameterizations of key processes related to the energy and mass balance of the snowpack.

COMMENT # 2.39

*L71: I found the 'instead' wording a bit heavy, I don't think that you need to detail what are the simplest parameterizations that you did not use ;)*

**Reply:**

Removed following the reviewer's suggestion.

**Changes:**

**2. Overview of the data assimilation system**

...

The most complex configuration is chosen by default in MuSA, leading to a more detailed simulation of the internal snowpack processes. Albedo is computed from the age of the snow, decreasing its value as snow ages and increasing it with fresh snowfalls. Thermal conductivity of the snowpack is computed as a function of the snow density. Snow density is computed considering overburden and thermal metamorphism. Turbulent energy fluxes are computed as a function of atmospheric stability. Melt water percolation in the snowpack is computed using the gravitational drainage. Although this is the default configuration, it is possible to choose any other FSM2 setup, which may result in slight performance differences both in terms of the computational cost and accuracy of the model (29).

COMMENT # 2.40

*L76: Don't you want to suggest the possibility to include multiphysics within the ensemble? That could be a nice place here.*

**Reply:**

Thanks for the suggestion. We agree that this opens up a fascinating opportunity to do model comparison, but we do not believe that this should be done within the same ensemble that is used for state and/or parameter estimation. Instead it could be done at the second level of inference in the Bayesian hierarchy. We nonetheless take the chance to highlight this possibility and point towards relevant literature.

**Changes:**

**2. Overview of the data assimilation system**

...

As envisaged in (30), the potential to run multiple model configurations leaves the door open for pursuing rigorous model comparison using the model evidence framework (31) in the cryospheric sciences.

COMMENT # 2.41

*L81: What is this constant file used for (briefly?) not sure it is worth mentioning if you don't say that*

**Reply:**

We have now included:

**Changes:**

**2. Overview of the data assimilation system**

...

and constants (constants.py) files that should be filled by the user to both configure the MuSA environment and define the priors respectively

COMMENT # 2.42

*L81: I think that from the user perspective, you'd make a better point here by briefly stating one or two very cumbersome implementation features (parallelization? IO?) that makes the life easier*

**Reply:**

We have added the following:

**Changes:**

**2. Overview of the data assimilation system**

...

Then, it solves most of the challenges of ensemble-based snow data assimilation frameworks for the user, as it internally handles the I/O and parallelization while keeping track of the ensembles that should flow in different ways depending of the chosen DA algorithm.

COMMENT # 2.43

*L88: OL usually stands for 'openloop', i.e. the ensemble without assimilation. Is this what you mean here? The standard name for deterministic run without assimilation/perturbation is 'control run', I think*

**Reply:**

Thanks for the suggestion. We have changed several sentences in the text and figures in order to be less ambiguous here, by using the term 'reference run' instead of 'open loop' for the deterministic runs that we use as a reference to gauge the performance of the DA experiments. Note that the terminology seems to be a bit ambiguous in the literature, several studies (e.g. 32; 33) use open-loop also to denote what the reviewer calls the control run while others follow the reviewer's definition. This mixed use of terminology is of course far from ideal. Our understanding was simply that open loop was any (possibly ensemble) run without DA. AS such, we prefer to leave this output file name unchanged for now. In future versions the output routines might be modified so we will reconsider this point at that stage, thanks for the suggestion.

COMMENT # 2.44

*L88: 'posterior ensemble' or 'posterior ensemble members', unless you refer to a group of ensembles.*

**Reply:**

Corrected.

COMMENT # 2.45

*Figure 1: Description of $N_o$ is missing*

**Reply:**

Fixed, thanks for spotting this. It should have read d.

COMMENT # 2.46

*Figure 1: DAW?*

**Reply:**

'Data assimilation window', we've now defined this acronym in the caption.

COMMENT # 2.47

*Figure 1: number of observation dates, I assume (several observations can be assimilated on the same date)*

**Reply:**

Corrected to "number of observation times" since dates implies strictly daily resolution.

COMMENT # 2.48

*L94: For the sake of completeness. I would add direct insertion to the list, mentioning that this is not a DA algorithm per se.*

**Reply:**

For clarity, we have completely removed direct insertion from the manuscript.

COMMENT # 2.49

*L96: I have the feeling that you are either telling too much, or too little here. Either you should expand on the technicalities in a dedicated subsection, or keep it simple and skip any jargon wording ('tempering', 'proposal', 'collapse', 'degenerate', 'over-inflating'). The first option would have the virtue of clarity and put more stance or your innovative contribution with the 'heuristic resampling'*

**Reply:**

We have now toned down the jargon here.

**Changes:**

**2. Overview of the data assimilation system**

. . .

Note that the Ensemble Kalman schemes involving multiple data assimilation are iterative schemes  (3; 5). For the PF, several standard resampling algorithms (see 34, for a review) are available in MuSA, namely: bootstraping, residual resampling, stratified resampling, and systematic resampling. In addition to these standard resampling techniques, we have also implemented a heuristic approach based on redrawing from a normal approximation to the posterior which is loosely inspired by  more advanced particle methods (18; 5). This redrawing from the approximate posterior generates new samples of perturbation parameters at each assimilation step. In ~~the case of a total collapse of the PF with this resampling strategy the posterior standard deviation of the perturbation parameters is 0 (all weight is assigned to one particle) which corresponds to a degenerate distribution. In this case, drawing new perturbation parameters from the degenerate posterior distribution would not generate new particles. To avoid this , in case of a complete collapse of the PF, MuSAover-inflating~~ overdispersing the ensemble.

COMMENT # 2.50

*L106: What scaling factor did you use then?*

**Reply:**

The default is 0.3, which we have now added to the text (see the response above).

COMMENT # 2.51

*L115: Could be removed, I think*

**Reply:**

Removed.

COMMENT # 2.52

*L120: You should probably say a word about OS portability in general here, this is the kind of things that might be OS-dependent*

**Reply:**

All the tmp file system operations are done using python libraries, so these should not be OS-dependent.

COMMENT # 2.53

*L124: How is direct insertion implemented? If you assimilate HS, are you "stretching" the layers? Are you changing the total mass or just adjusting density. This is essential.*

**Reply:**

Thanks for the question, but we have removed direct insertion from the manuscript.

COMMENT # 2.54

*L124: How is direct insertion implemented? If you assimilate HS, are you "stretching" the layers? Are you changing the total mass or just adjusting density. This is essential.*

**Reply:**

As previously stated, we have removed direct insertion from the manuscript.

COMMENT # 2.55

*L124: This seems useless/confusing*

**Reply:**

Removed.

COMMENT # 2.56

*This is not the first mention of "particle" so please introduce it earlier. Also the reference is not needed for that. Finally, I don't see a problem with using the wording of particles even when referring to EnKF ensemble members, but you'd better make that clear for the people coming from the EnKF world.*

**Reply:**

Although it is not the first mention (indeed it is the third if we exclude the abstract) of particle (outside of DA algorithm names), it is the first place that the "number of ensemble members" is introduced. Since this is synonymous with "number of particles" it arguably makes sense to establish this link here. There is in principle no issue using the words ensemble member and particle interchangeably, which is what we do. This is why we added a reference which advocates this equivalence. We have nonetheless corrected the reference which should have been (6) not (5).

**Changes:**

**2.1 Ensemble generation**

…The number of ensemble members  (or, equivalently, particles; 6), which we denote by $N_e$, should be specified by the user, as it drastically affects both the computational cost of the experiments and the performance of the data assimilation algorithms

COMMENT # 2.57

*L133: I disagree: perturbations error models are primarily related to our understanding of how the model errors actually are, which depend on the variable. for example NWP absolute temperature bias doesn't depend much on the forecast value, hence an additive noise model, while precipitation absolute error value is roughly proportional to the forecast value, hence the multiplicative model.Then as a boundary effect we need to ensure that the perturbed input remain physically sound, but this is not the first constraint*

**Reply:**

The first constraint is surely that the input remains physically sound, otherwise it would lead to nonsensical simulations that can in the worst case lead to the model crashing. Moreover, the entire point of Bayesian DA is that *we do not know* what the errors actually are for a given experiment, otherwise they could be trivially corrected, this is why we represent them using probability distributions that encode epistemic (i.e., related to our lack of knowledge) uncertainty about these errors. What we often do know is the expected behaviour, in terms of magnitudes and bounds, of the variable that is being perturbed. This is what guides us in constructing the prior. One is of course also free to use independent historical data in designing the prior, which is perhaps what the reviewer is referring to. However, this is not an imperative. It is also advisable to check that posterior uncertainty estimates are well calibrated, in the sense that they capture the actual errors, but this is necessarily a validation (also known as ensemble verification) step to avoid circularity. At the heart of this disagreement lies the interpretation of probability as a measure of epistemic uncertainty (degree of belief) rather than frequency. The Bayesian approach to inference that lies at the core of MuSA primarily follows the former epistemic interpretation. For more on this topic see the philosophical discussions in (35) and (36). We have now included citations to these papers, but chose not to delve into these issues in the manuscript since they are a matter of the philosophical interpretation of probability that we assume at the outset of this study.

**Changes:**

**3.1 Bayesian inference**

In line with most modern approaches to DA (37; 38; 5) the assimilation schemes used in MuSA are built on the foundation of Bayesian inference (e.g. 35; 31; 18; 36).

SMALL CAPS COMMENT # 2.58

*L157: It's not necessary clear for the reader that solving gird cells independently reduces the memory cost, please expand a bit more here (I guess it comes from the potential parallelization)*

**Reply:**

We have made the following changes to clarify this and related issues.

**Changes:**

**2.2 Meteorological forcing**

. . .  Each grid cell is solved independently  which includes the reading of the forcing  that occurs along the time dimension. Otherwise, each process would have to store considerably more data in memory leading to more costly I/O operations that would slow down the run time. Even so, just reading along the time dimension can come with a considerable computational cost if the time dimension is large. To alleviate this, the  time spent reading the forcing can be reduced by setting the chunk (a subset of the file to be read or written as a single I/O operation) of the netCDF forcing files along the time dimension. To speed up the subsequent relaunching of the simulations when smoothing and filtering, MuSA generates an intermediate binary file with the forcing information  needed to run a complete simulation for each grid cell.

SMALL CAPS COMMENT # 2.59

*L157; This is not clear.*

**Reply:**

See the answer above.

**COMMENT # 2.60**

*L160: What is a "chunk" ? People who know that word are probably well aware of the issue, while people who don't may have no clue of the meaning, and the problem. Please clarify*

**Reply:**

We have now defined chunk in the manuscript, see the response to Comment # 2.58.

**COMMENT # 2.61**

*L161: What does relaunching mean? After an assimilation step? (then ' model propagation' could be more appropriate) Or relaunching a whole simulation experiment from a starting point? Please clarify.*

**Reply:**

The actively used forcing for a particular grid cell is stored in these intermediate files, so the term relaunching is valid for all of these situations; i.e. either restarting from the start of the water year when smoothing or after a filtering step. We believe that relaunching in this context describes both situations adequately, but have clarified this in the manuscript as shown in the response to Comment # 2.58.

**COMMENT # 2.62**

*L161: I don' t get that, is this a unique file covering all grid cells and timesteps? What is the difference with a forcing file then?*

**Reply:**

Thanks for these detailed questions, we believe our answers and changes are helping to clarify the manuscript. As stated in the answers above (particularly the response to Comment # 2.58), there is one intermediate file for *each grid cell*. So not one for all grid cells. The difference is that the full forcing file will typically contain the entire spatiotemporal forcing for a given experiment, which we do not need for one water year and one particular grid cell. Keeping all that in memory or reading it several times for each process slows down the computations considerably. Instead, these individual temporary files are generated to simplify the ensemble simulation (which includes relaunches) of a particular grid cell. We believe that this has been explained

in enough detail now. Curious readers can consult the code or ask the developers why a particular design choice was made.

**COMMENT # 2.63**

*L162: Initialization usually hints at the preparation of the initial state rather than to the forcing, I don't see the link here*

**Reply:**

This sentence was removed.

**COMMENT # 2.64**

*L169: More importantly, are all observations of a same variable given the same error?*

**Reply:**

This is not the case although the reviewer's interpretation is understandable given the way this was initially formulated in the manuscript. Please see our answer to Comment # 2.8.

**COMMENT # 2.65**

*L171: repetition of l. 112-113*

**Reply:**

Correct, we have removed this sentence to avoid repetition.

**COMMENT # 2.66**

*L172: While this seems pretty easy with a batch smoother, I think that this would deserve simple explanations for the filtering cases*

**Reply:**

See the answer to Comment # 2.9.

**COMMENT # 2.67**

*L178: Is this really relevant if you assume a scalar error for a given variable nevertheless? (see my comment above)*

**Reply:**

Now that we have clarified that we assume the same scalar error for a particular observation product (there can be many products for a particular variable) we do believe that it is important to state that we do not recommend gap filling observed data. This is because the assumed error attached to each product is usually a result of independent validation studies (e.g. 39) for that particular product (of course, the actual error also varies in space and time). Crucially, doing gap-filling adds another layer of modeling onto the observation product which would need to be accounted for in the setting of the observation error variance for that particular gap-filled product.

COMMENT # 2.68

*L180: typo*

**Reply:**

Fixed.

COMMENT # 2.69

*L194: This term has been used earlier on, without introduction, consider restructuring this*

**Reply:**

The earlier use of likelihood was removed as part of the abolition of premature jargon as a response to Comment #2.49.

COMMENT # 2.70

*L194: This reasoning on evidence is a nice piece of theory, but I think that this is irrelevant here, as you actually acknowledge in the last sentence. Consider removing it. I'm really looking forward to reading this in a paper where you actually make use of this concept fruitfully, though!*

**Reply:**

The evidence pops up quite a bit in the discussion of the theory of importance sampling used in particle methods and now also in the new sentence we added as a response to Comment # 2.40. We would thus like to keep this section.

COMMENT # 2.71

*L215: I would rephrase into: in the case of the variables we assimilate within MuSA, or "in general". One could imagine assimilating Top Of Atmosphere visible radiances, or Radar backscatter coefficients by virtue of complex observation operators in the years to come*

**Reply:**

Point taken, we have now gone with "Often" instead of generally. Nonetheless, in the Bayesian inversion (or data generating model) view of the DA problem the distinction between the dynamical model and the observation operator is a bit artificial. Arguably both are part of the data generating model linking the hidden variables (or parameters) that we want to infer to the observed variable.

**Changes:**

**3.1 Bayesian inference**
… Often, as is the case in MuSA, the observation operator  picks out predicted observations, i.e. the state variables that correspond to observations, from the full state vector.

COMMENT # 2.72

*L237: This whole paragraph seems a bit odd here. For me the main purpose only seems to justify the use of MC methods versus variational methods. I would condense (skip the degeneracy) and move to the introduction.*

**Reply:**

Actually our intention was to explain that *in theory* the Bayesian inference at the heart of DA is easy and boils down to simple probability calculus. What is hard is doing it in practice beyond very simple toy models. This paragraph doesn't have anything to do with the particular case of degeneracy of particle filters per se, only the broader concept of the curse of dimensionality which is just to say that the volume of the state and/or parameter space with high posterior probability mass (the needle) is usually smaller than that of the prior (the haystack). This is a big reason why DA is a hard problem and why naive methods like the grid approximation are impractical since the discretization required to resolve "the needle" given that our likelihoods

(i.e. forward models) are too expensive to evaluate. To clarify what we mean by grid approximation, we have included an additional citation of (31) (which is available open access) where these naive and very costly (although potentially very accurate if you can afford them!) methods are discussed in Chapter 21.

**Changes:**

**3.1 Bayesian inference**
In theory, evaluating the posterior simply involves taking the product of two terms. Naïvely, this suggests that we can estimate the posterior through a simple grid approximation (31).

*L262: I would condense this to the maximum.*

**Reply:**

We have made an effort to condense this without losing the essence of the message we were trying to get across.

**Changes:**

**3.2 Prediction, filtering, and smoothing**
...  Since prediction is a step in both filtering and smoothing,  we explain how to implement it probabilistically.  Prediction from $k-1$ to $k$ can be formulated as follows:

$$\mathbf{x}_k = \mathcal{M}(\mathbf{x}_{k-1}) + \boldsymbol{\eta}_{k-1}, \tag{1}$$

where $\mathcal{M}(\cdot)$ is the dynamical model (FSM2 in MuSA) while $\boldsymbol{\eta} \sim N(\mathbf{0}, \mathbf{Q})$ is the additive model error  term which we assume to be independent in time and follow a zero-mean Gaussian distribution with covariance matrix $\mathbf{Q}$. These assumptions can be relaxed without loss of generality, but their convenience and broad justifiability mean that they are often employed  (38). Crucially, the above prediction step produces Markovian (memoryless) dynamics where the current state depends only on the previous state and  noise. The Markov property is crucial in making the filtering and smoothing problems tractable. It implies $p(\mathbf{x}_k|\mathbf{x}_{0:(k-1)}) = p(\mathbf{x}_k|\mathbf{x}_{k-1})$ which lets us factorize and simplify distributions such as the full prior as follows

$$p(\mathbf{x}_{0:k}) = p(\mathbf{x}_k|\mathbf{x}_{0:(k-1)})p(\mathbf{x}_{0:(k-1)}) = p(\mathbf{x}_k|\mathbf{x}_{k-1})p(\mathbf{x}_{0:(k-1)}), \tag{2}$$

where the transition density is Gaussian of the form $p(\mathbf{x}_k|\mathbf{x}_{k-1}) = N(\mathbf{x}_k|\mathbf{x}_{k-1}, \mathbf{Q})$. Applying this recursively we obtain

$$p(\mathbf{x}_{0:k}) = p(\mathbf{x}_k|\mathbf{x}_{k-1})p(\mathbf{x}_{k-1}|\mathbf{x}_{k-2})\dots p(\mathbf{x}_1|\mathbf{x}_0)p(\mathbf{x}_0) = p(\mathbf{x}_0)\prod_{j=1}^{k} p(\mathbf{x}_j|\mathbf{x}_{j-1}), \qquad (3)$$

Using this kind of factorization together with marginalization also helps us construct the  marginal predictive distribution $p(\mathbf{x}_k|\mathbf{y}_{1:\ell})$ where $\ell < k$ as follows

$$p(\mathbf{x}_k|\mathbf{y}_{1:\ell}) = \int p(\mathbf{x}_k, \mathbf{x}_{k-1}|\mathbf{y}_{1:\ell})\, d\mathbf{x}_{k-1} = \int p(\mathbf{x}_k|\mathbf{x}_{k-1})p(\mathbf{x}_{k-1}|\mathbf{y}_{1:\ell})\, d\mathbf{x}_{k-1}. \qquad (4)$$

This is the Chapman-Kolmogorov equation (18) which can be applied recursively to obtain the predictive distribution at the current time step using the transition density together with previous predictive distributions.

From prediction we move to filtering which  is the estimation of the current state given current and past observations: $p(\mathbf{x}_k|\mathbf{y}_{1:k})$. This is the problem solved by sequential DA where an archetypal example is the initialization of numerical weather predictions as new observations become available to delay the effects of chaos (40). To construct the filtering distribution we first re-introduce our Gaussian observation model in the dynamical context  and make the usual assumption that the current observations are conditionally independent of both the observation and state histories (18) resulting in the dynamic likelihood

$$p(\mathbf{y}_k|\mathbf{x}_{0:k}, \mathbf{y}_{1:(k-1)}) = p(\mathbf{y}_k|\mathbf{x}_k) = A_k \exp\left(-\frac{1}{2}[\mathbf{y}_k - \widehat{\mathbf{y}}_k]^\mathsf{T} \mathbf{R}_k^{-1} [\mathbf{y}_k - \widehat{\mathbf{y}}_k]\right), \qquad (5)$$

where $\widehat{\mathbf{y}}_k = \mathcal{H}(\mathbf{x}_k)$ are the predicted observations  and we have added a time index to the normalizing constant ($A_k$) and the observation error co-variance matrix ( $\mathbf{R}_k$) to emphasize that both the number and types of observations  may vary in time. By combining Markovian state dynamics with a conditionally independent observation model, we end up with a state-space or hidden Markov model (41) where the states at each timestep are hidden (or latent) because they are not observable due to measurement error. The filtering distribution  is obtained by combining the predictive distribution  (which serves as the prior) and the dynamic likelihood through Bayes' theorem

COMMENT # 2.74

*L326: kind of repetition of l. 317, consider rewriting*

**Reply:**

We removed part of this to minimize repetition, but since this is an important point we would like to keep these sentences.

**Changes:**

**3.2 Prediction, filtering, and smoothing**

... This  is  crucial in peak SWE reconstruction which is of great interest to snow hydrologists (42).

COMMENT # 2.75

*L333: I don't get the purpose of this paragraph here: I think that it would increase clarity if you state whether any of the implemented assimilation algorithms is prone to introduce this flavour of dynamical inconsistency.*

**Reply:**

The purpose of this paragraph is to explain what we mean by dynamical inconsistency and how it can arise. In the subsequent paragraph, we clearly state that both the PF and EnKF suffer from this flavor of dynamical inconsistency.

COMMENT # 2.76

*L350: To me this is equally as important as inconsistencies introduced via the assimilation itself so I would expand on both. For instance inconsitencies between perturbed meteorological parameters may induce weird model reaction.*

**Reply:**

We do not agree that we spend less time on this issue than inconsistencies from the assimilation step, with two paragraphs used on both. Moreover, unless the perturbations make the forcing completely unphysical, from our experience it is unlikely to cause the model to crash. While it can certainly make sense in some cases to consider adding correlation between the different forcing perturbations, or even better generate the forcing ensemble directly with atmospheric models, this is not an imperative. Instead, we see that this need stems more from the (often implicit) interpretation of probability as long run frequency rather than the more general measure of our epistemic uncertainty that we use here.

*L353: Please be more specific as this does not apply to the PF*

**Reply:**

We already made it clear that it is more of a concern for the ensemble Kalman schemes. See the answer below.

*L358: I don't see from where physical consistencies could arise from in the PF. Gaussian model errors may be suboptimal/violate assumptions in the case of bounded variables, but how can that introduce physical inconsistencies?*

**Reply:**

The issue we are discussing here is quite general and not a specific criticism of the PF. The point is just that if you use a Gaussian assumption in your forcing perturbations or for the model error (process noise) in the state this can lead to physical inconsistencies in *any* snow model just because of the fact that many snowpack variables are bounded from below (e.g. SWE, snow depth) and above (e.g. albedo, FSCA). Moreover, the usual assumption of Gaussian observation error, which is a very common choice in DA (including this study), is probably a poor choice of likelihood when dealing with bounded observations. By pointing this out we are not trying to single out any particular method, but just illuminate potential problems that arise if we do not think carefully about the probabilistic models we use. There are of course simple fixes to this that are widely used in practise, like manually enforcing the bounds, but this is technically then a violation of the underlying probabilistic model. Instead of just continuing with such solutions we should consider to pursue different probabilistic models (e.g. 8; 9), which Gaussian anamorphosis (and other tools), can help with.

*L363: This is absolutely essential. The problem with the way this section is written right now, is that this information is hardly accessible to the reader.*

**Reply:**

We agree that this is essential which is why we devote most of this page to discussing the matter, and return to it as well in Section 3.5.

**COMMENT # 2.80**

*L363: Is gaussian anamorphosis really applied in the case of the PF variants?*

**Reply:**

Yes, it is currently applied under the hood for the PF when we sample from the prior for bounded variables. Unlike the ensemble Kalman methods, however, we do not need to worry about these transformations in the assimilation step with the PF. As we make clear in the text, it is only the ensemble Kalman schemes that make an explicit Gaussian assumption which is why the potential for physical consistency is an even bigger concern for these schemes. With particle methods we are in theory much more flexible with our choice of priors and likelihoods.

**COMMENT # 2.81**

*L374: appraoch*

**Reply:**

Fixed.

**COMMENT # 2.82**

*L379: Also if relevant, consider work from myself and colleagues, where we include model uncertainty in the ensemble construction with a detailed snowpack model (Cluzet et al., 2020-2022)*

**Reply:**

Thanks for the suggestion, we have added these references here.

**COMMENT # 2.83**

*L391: ( SM2)*

**Reply:**

Fixed.

**COMMENT # 2.84**

*L392: This repetition is raising more ambiguity than it resolves them, please rephrase/delete*

**Reply:**

Removed.

**COMMENT # 2.85**

*L407: This is the kind of essential information (fundamentally different implementation between the filtering and the smoothing approaches), that the reader overwhelmed with theoretical developments will miss*

**Reply:**

Yes exactly, which is why we repeat this here for emphasis after having already mentioned it at the end of Section 3.2. We even repeat it briefly one more time at the end of the current Section (3.3).

**COMMENT # 2.86**

*L441: I'm not sure that such a level of Bibliographic detail is required here. If so, please consider my additions in the end.*

**Reply:**

We believe that the level of bibliographical detail is appropriate for the description of a toolbox that includes most widely used snow data assimilation techniques. We will thus add the suggested additions in the next comment.

**COMMENT # 2.87**

*L458: There is more to say on that topic: In Cluzet et al., (2022), we have investigated the propagation of information from true observations in a network of snow depth stations. Also consider the work of Cantet et al., 2019 and Odry et al., (2022) as well as Winstral et al. (2019)*

**Reply:**

We have added all but the last reference which we do not believe fits under this section on particle methods.

**Changes:**

**3.4 Particle methods**

…Recent efforts  (44; 43; 28; 45) have focused on the challenge of spatially propagating snowpack information from observed to unobserved locations when assimilating in-situ snow depth and SWE data using the PF.

C OMMENT # 2.88

*L466: Non-iteratve*

**Reply:**

Fixed.

C OMMENT # 2.89

*L475: Does this consideration only apply to smoothing methods? If not, consider restructuring a bit.*

**Reply:**

Good point, the last sentence here on satellite-based snow depth retrievals is quite general so we have modified it and moved it to the discussion.

**Changes:**

**6. Discussion**

…Furthermore, the assimilation of  high resolution snow depth maps may become a common practice in the future, even at wider scales, thanks to  satellite-based snow depth retrievals (46; 47; 48). .

C OMMENT # 2.90

*L491: Converege*

**Reply:**

Fixed.

**COMMENT # 2.91**

*I don't see the point of introducing the proposal density in this paper, since it is not used (i.e. trivially) in this paper as stated in l. 510-512 and l. 557-559*

**Reply:**

The point with introducing the proposal density is expressed on L520 in the former version of the manuscript, namely that unless the proposal and the target are close to one another then degeneracy is likely to occur. Since the prior is often not close to the posterior, it is actually a particularly poor choice of proposal even if it is often (as in the case of MuSA and most other snow DA we are aware of) the default and often unconscious choice. Defining the proposal clearly here helps us to weave it into the discussion later in the paper. Moreover, the special case where we recover direct MC discussed on L510-512 is about using the posterior (not the prior) as the proposal.

**COMMENT # 2.92**

*L530: what is N here?*

**Reply:**

For clarity, we have now replaced all $N$ with $N_e$ throughout the text to emphasize that this always denotes the number of ensemble members (particles) and to avoid confusion with $N(\cdot)$ which we use to denote a Gaussian distribution. This is now clearly stated the first time the concept is introduced in Section 2.1.

**Changes:**

**2.1 Ensemble generation**
...The number of ensemble members  (or, equivalently, particles; 6), which we denote by $N_e$, should be specified by the user, as it drastically affects both the computational cost of the experiments and the performance of the data assimilation algorithms.

**COMMENT # 2.93**

*L544: Consider skipping this statement.*

**Reply:**

We would like to keep this statement as it indicates that there are many flavors of particle smoothers not explored herein.

COMMENT # 2.94

*L555: Fair point indeed! (perhaps state more explicitly that this is an I/O bottleneck)*

**Reply:**

Done, thanks for the suggestion.

**Changes:**

**3.4 Particle methods**
The batch approach used in the PBS is appealing since it can be wrapped around the model allowing the dynamics to evolve freely for the whole data assimilation window from $t_0$ to $t_n$ without the need for I/O interruptions in the time integration, typically resulting in marked run time acceleration compared to sequential approaches.

COMMENT # 2.95

*L567: Very nice pragmatic addition, although since this book is not available open-source I'd really recommend to expand a bit.*

**Reply:**

Thanks for the recommendation, please see our response to Comment # 2.11

COMMENT # 2.96

*L575: Is there a real need to introduce this new terminology? I cannot see any conceptual difference with the coined term of particle filter "weights" defined as the normalized likelihoods?*

**Reply:**

We have now changed this particular mention to just weights since the in-line equation makes it clear what kind of weights we are referring to. Note that by following the nomenclature of (11) we are not advocating against the use of the general term weights, but rather want to be more precise where there could be ambiguity. Moreover, as should be clear from the discussion of proposals, these (auto-normalized) weights do not have to be normalized likelihoods that is just the typical (but special)

case of using the prior as the proposal.

COMMENT # 2.97

*L582: This is pretty obvious, I would skip, and keep this computational subtlety for the ensemble statistics in the smoother as in l. 595-596*

**Reply:**

This may not be obvious to all readers. Moreover, skipping this but keeping the corresponding discussion for the PBS could be misleading since there the implementation is different. In particular, for the PBS we use weights to calculate the posterior statistics while for the PF we do not need to use the weights for this since they are equal after resampling.

COMMENT # 2.98

*L598: I don' think that this much of a context is needed here to introduce the EnKF*

**Reply:**

One of our goals in extensively introducing the schemes used in MuSA is to help demystify them. To do that it helps to briefly discuss the origin of the schemes. Given that they seemingly have a longer history, this takes a few more sentences for the ensemble Kalman methods than with the particle methods.

COMMENT # 2.99

*L621: One full page of bibliography on EnKF/ES and variants applied to snow is just too much, please condense.*

**Reply:**

See the answer to Comment # 2.13.

COMMENT # 2.100

*L661: Again another instance of useful statement that would stand one much more in a condensed version of the paper.*

**Reply:**

It should be more visible now that we have condensed this paragraph, see answer below.

COMMENT # 2.101

*L662: I would skip this, or condense it to the maximum. At this point, the reader is over-whelmed with theoretical considerations that make very little to no difference in practice -as aknowledged-, and I think you want to avoid that.*

**Reply:**

We agree and have condensed this section accordingly, while retaining the relevant references.

**Changes:**

**3.5 Ensemble Kalman methods**
... There are actually several variants of the ensemble Kalman analysis step (see 38). In MuSA we use the so-called stochastic rather than deterministic (or square-root) implementation. This stochastic formulation  adds perturbations to the predicted observations to ensure adequate ensemble spread  and consistency with the underlying Bayesian theory (49; 50).

COMMENT # 2.102

*L667: Fair enough :)*

**Reply:**

We are not sure what the reviewer is pointing to here, but we are happy with his constructive comments and annotations.

COMMENT # 2.103

*The introduction of the iterative EnKF early on in the description like that might be a bit puzzling, you may want to recap on that very quickly before.*

**Reply:**

Good point, we have added the following before jumping into the presentation of the algorithms.

**Changes:**

**3.5 Ensemble Kalman methods**

...The iterative versions tend to perform better than their non-iterative counterparts for non-linear problems (4). Let $N_a$ denote the number of assimilation cycles (iterations) performed in a pseudo (rather than model) time.

COMMENT # 2.104

*L719: Please add the area.*

**Reply:**

Added.

**Changes:**

**4. Data and experimental setup**

...We developed two data assimilation experiments in the 55 ha Izas experimental catchment (51) in the Spanish Pyrenees (see Figure 2).

COMMENT # 2.105

*L726: (?Revuelto et al., 2021a)*

**Reply:**

Fixed.

COMMENT # 2.106

*L727: Consecutive*

**Reply:**

Done.

COMMENT # 2.107

*L730: This sentence seems a bit awkward, in particular the wording of "inflating the uncertainty of the forcing". (1) I don' t see where does that come from? Is there any retroaction loop between forcing perturbation spread and MuSA, that would lead to increase it? Please reformulate.(2)I would expand a bit more: you expect model errors to be primarily due to unrepresented processes, dominated by wind redistribution of snow. You expect MuSA to compensate this by reducing/increasing precipitation amounts/ablation on a point base level.*

**Reply:**

Agreed, modified as follows:

**Changes:**

**4.1 Single cell and distributed assimilation of drone-based snow depth retrievals**
Due to the high-resolution of the observations, we expect MuSA to be able to implicitly reproduce the wind redistribution patterns by  modifying the precipitation and temperature at grid cell scale to compensate for the deposition and removal of wind-blown snow.

COMMENT # 2.108

*Please add litterature/minimal justification for the perturbation model and their statistic*

**Reply:**

See the response to Comment # 2.57 above and Comment # 2.110 below.

COMMENT # 2.109

*This seems quite high (standard deviation of 0.2m) for a pure measurement error, does this include any representativeness error? I don't know whether these make sense @5m resolution, but a minimal discussion is required here as representativeness errors would very likely depend on observation value (Lopez-Moreno et al., 2011?)*

**Reply:**

An RMSE of 0.2m for drone observations of snow thickness is very reasonable as stated by the references in the text.

COMMENT # 2.110

*At this stage, it is essential to reference the section in which this implementation subtlety has been introduced.*

**Reply:**

Here we are once more back to the discussion of # 2.57 and the confusion between frequency and probability which is unfortunate at least when we adopt the Bayesian interpretation of probability theory. There is no correct or true prior out there in the external world, instead it encodes the epistemic uncertainty that we or some other agent (such as our model) has about a quantity. Making the mistake of attributing prior probability to the real world is nicely (if a bit polemically) summarized by E.T. Jaynes' "mind projection fallacy". Note that this does not mean that all priors are created equal, certainly the prior of a well informed expert (a real agent or a model) will be more valuable than a completely vague prior. We nonetheless agree that our introduction of priors lacked appropriate references and keywords that can help to explain our choices so we made the following changes:

**Changes:**

**3.1 Bayesian inference**

...

The next ingredient is the prior distribution over states, $p(\mathbf{x})$, which can be specified based on initial beliefs which may include physical bounds, expert opinion, 'objective' defaults (using e.g. maximum entropy), or knowledge from earlier analyses (35; 31; 52)...

**4.1 Single cell and distributed assimilation of drone-based snow depth retrievals**

...

As such, these are weakly informative priors (c.f. 52) in that they produce predictions that we expect a-priori (i.e., without looking at the data) to be in the right correct of magnitude while obeying physical bounds. ...

COMMENT # 2.111

*L736: mu=1?*

**Reply:**

This is the mean of the associated normal distribution, which produces a median for the lognormal distribution of $\exp(\mu) = \exp(0) = 1$. For the justification of this particular choice of value (or more precisely, the entire prior distribution), our response is the same as for the comment above.

COMMENT # 2.112

*L738: This seems quite high (standard deviation of 0.2m) for a pure measurement error, does this include any representativeness error? I don't know whether these make sense @5m resolution, but a minimal discussion is required here as representativeness errors would very likely depend on observation value (Lopez-Moreno et al., 2011?)*

**Reply:**

This is not high (although it is slightly conservative) given the values reported in (53) that we based it on. Therein, the highest RMSE at 1 m resolution are reported as being around $0.6$ m. If we assume that the errors are unbiased, independent, and identically Gaussian distributed with this error standard deviation $\sigma_1 = 0.6$ m, then using the central limit theorem this translates into an aggregated error at 5 m resolution (so containing 25 pixels at a resolution of 1 m) of $\sigma_5 = \sigma_1/\sqrt{25} = 0.12$ which is of the same order as the error standard deviation we assume of $\sigma_5 \simeq \sqrt{0.04} = 0.2$ m.

COMMENT # 2.113

*L740: At this stage, it is essential to reference the section in which this implementation subtlety has been introduced.*

**Reply:**

We've now added reference to the relevant section.

**Changes:**

**4.1 Single cell and distributed assimilation of drone-based snow depth retrievals**
... In the case of the single cell comparison, we followed two different resampling strategies, the bootstrap and redraw from a normal approximation of the posterior (see Section 2)...

COMMENT # 2.114

*L753: So the Izas catchment is covered by only one pixel?*

**Reply:**

Yes, the area of the Izas catchment (55 ha, see Comment # 2.104) is approximately the size of one MODIS pixel. We also make it clear (Line 753 in the original manuscript) that we pick the pixel that is closet to the centroid of the Izas catchment.

COMMENT # 2.115

*L763: As commented earlier, this is to my opinion not the best justification.*

**Reply:**

As should be clear by now from the preceding discussion, it seems like perhaps we fundamentally disagree on this point (i.e., the meaning of probability) on a philosophical level that is beyond the scope of this manuscript.

COMMENT # 2.116

*Table 1: Please introduce these acronyms here.*

**Reply:**

Done.

COMMENT # 2.117

*Table 1: A summary description of the different algorithms (filter, smoother and their parameter values/configuration would help here.*

**Reply:**

Done.

COMMENT # 2.118

*L780: This is not very clear here, the reader has to dig out the information from Sec. 3, please insert a recap (or a table, see comment above)*

**Reply:**

We have reformulated this slightly, but want to avoid describing the methods anew each time they are mentioned. The reader could use the search function to easily discover what redraw means in this context.

**Changes:**

**5.1 Single cell and distributed assimilation of drone-based snow depth retrievals**
…The particle filter with  redraw-based resampling allowed the assimilation process to recover from the initial collapse through particle rejuvenation, leading to a more realistic non-degenerate ensemble simulation.

COMMENT # 2.119

*L782: Please be more specific. In Fig. 3, the agreement seems pretty good in 2019-2020*

**Reply:**

Corrected.

**Changes:**

**5.1 Single cell and distributed assimilation of drone-based snow depth retrievals**
Here, the  EnKF produced unsatisfactory results when the observations fall very far from the  posterior ensemble in the first season, improving considerably in the second. Note, however, that we did not observe  the same issue with the EnKF-MDA..

COMMENT # 2.120

*L784: EnKF-MDA*

**Reply:**

Corrected, see above.

COMMENT # 2.121

*Figure 3: Add panel numbers*

**Reply:**

Added.

COMMENT # 2.122

*L787: 3, bottom left)*

**Reply:**

Added.

COMMENT # 2.123

*L791: The transition from point to spatial is a bit sharp, consider at least introducing a blank line, or a section break*

**Reply:**

This is already a new paragraph. Note that this will looks cleaner in the final typeset version of the manuscript were the manuscript to be accepted.

COMMENT # 2.124

*L797: Since the subtle computation of the ensemble statistics have been thoroughly explained in 3.4 l.582-586), I would remove the 'weighted' for clarity*

**Reply:**

Good point, we changed "weighted" to "posterior".

COMMENT # 2.125

*L797: I suggest also to skip these elements on the perturbation statistics, which should be clear at this stage*

**Reply:**

Done.

COMMENT # 2.126

*L798: Multiplier*

**Reply:**

Corrected.

COMMENT # 2.127

*L801: Precipitation multiplier. A bias is inherently based on a substraction*

**Reply:**

Corrected.

COMMENT # 2.128

*Figure 4: What does that mean?*

**Reply:**

Corrected.

COMMENT # 2.129

*Figure 4: Sequential Importance resampling?*

**Reply:**

Both PFs use SIR, the redraw is the heuristic approach to resampling that we tested here and seems to work quite well. The caption has been modified to clarify which scheme we are referring to.

COMMENT # 2.130

*L830:. hese*

**Reply:**

Corrected.

COMMENT # 2.131

*L831: This review is about filters, not smoothers, while you were taking about smoothers two sentences before: the logic is hard to follow.*

**Reply:**

These suggested remedies could be implemented in both filters and smoothers.

COMMENT # 2.132

*L842&L845: In addition x2*

**Reply:**

Corrected

COMMENT # 2.133

*Figure 7: this is LST, not SST*

**Reply:**

Corrected

COMMENT # 2.134

*Figure 7: IKS is never defined in the text (Iterative Kalman Smoother?)*

**Reply:**

Corrected to ES-MDA.

COMMENT # 2.135

*L859: Winstral et al., 2019, Cantet et al., 2019, Odry et al., (2022), Cluzet et al., (2022) are all necessary citations here.*

**Reply:**

Done.

**Changes:**

**6. Discussion**
. . . Despite the fact that there are some other examples of assimilating snow depth products, most of these have been carried out using in-situ observations (44; 28; 45), or were developed at coarser spatiotemporal resolutions (54).
.

COMMENT # 2.136

*L866: Please consider referring to the airborne snow observatory (ASO) as it is to my knowledge the only hyper-resolution remotely-sensed snow depth retrieval available.*

**Reply:**

Thanks for the suggestion, we agree that the ASO is a fantastic mission where it is flown. Still, we were alluding to satellite-based remote sensing of snow depth here and have now made that clear.

**Changes:**

**6. Discussion**

...Furthermore, the assimilation of  high resolution snow depth maps may become a common practice in the future, even at wider scales, thanks to  satellite-based snow depth retrievals (46; 47; 48)

COMMENT # 2.137

*L869: Do you mean LST? Otherwise I don' t understand this sentence.*

**Reply:**

Yes, fixed.

COMMENT # 2.138

*L870: This is not evidenced in this experiment, since LST and FSCA are available throughout he season, we cannot tell whether a positive impact would still be obtained with LST alone over the winter months. A synthetic experiment could easily be carried out to investigate this.*

**Reply:**

We have qualified our statement accordingly.

**Changes:**

**6. Discussion**

...The assimilation of  LST has the potential to provide additional information when FSCA saturates at 1,  for example during most of the  accumulation season and during the polar night

in the absence of sunlight.

COMMENT # 2.139

*L880: Insert blank line*

**Reply:**

It is already a new paragraph (hence the indent). The final typesetting style will be dependent on the style of the journal, not on us.

COMMENT # 2.140

*L892: Inclusion*

**Reply:**

Changed.

COMMENT # 2.141

*L896: Interesting, but I'm not sure I see what you mean here. Please expand/reformulate*

**Reply:**

Modified as follows:

**Changes:**

**6. Discussion**

. . . This suggests that a  more direct integration of FSM2  that avoids I/O operations and system calls could improve the overall  performance of MuSA. .

COMMENT # 2.142

*L902: References?*

**Reply:**

We have added some related references.

**Changes:**

**6. Discussion**

The implementation of more sophisticated models that include detailed radiative transfer schemes may provide MuSA the capability of ingesting new remotely sensed information such as  shortwave reflectances (55) or radar backscatter (48).

COMMENT # 2.143

*L903: Output temporal resolution. Changing internal timestep resolution is a bit more hazardous (both for numerical stability and accuracy), I don't think that you want to suggest that*

**Reply:**

Actually, this is what we meant. It is related to the fact that FSM2 runs with a fixed timestep. So for large scale implementations it would be worth experimenting with coarsening the timestep (e.g. 3 hours) and conversely for detailed site-level simulations.

COMMENT # 2.144

*Conclusions*
*L908&L909: This is very much of a detail, I would rather insist on the fact that MuSA includes smoothers and filters, covering both real-time and reanalysis applications.*

**Reply:**

Since the resampling strategy can impact the performance, we believe that it is worth to keep this sentence. We have nonetheless modified it slightly.

**Changes:**

**7. Conclusions**

MuSA is a new snow data assimilation system that encapsulates the FSM2 snowpack model. There are 6 different ensemble-based data assimilation algorithms implemented in MuSA, as outlined in detail in Section 3, with  several different resampling strategies in the case of particle filters ...

COMMENT # 2.145

*Code and data availability*
*L922: https://...*

**Reply:**

Added.

COMMENT # 2.146

*References*
*L986: Part of the title is missing*

**Reply:**

Fixed.

COMMENT # 2.147

*L1136: Missing DOI*

**Reply:**

Fixed.

REFERENCES

[1] K. Aalstad, S. Westermann, T. V. Schuler, J. Boike, and L. Bertino, "Ensemble-based assimilation of fractional snow-covered area satellite retrievals to estimate the snow distribution at Arctic sites," *The Cryosphere*, vol. 12, p. 247–270, 2018.

[2] A. A. Emerick and A. C. Reynolds, "Ensemble smoother with multiple data assimilation," *Computers & Geosciences*, vol. 55, p. 3–15, 2013.

[3] A. S. Stordal and A. H. Elsheikh, "Iterative ensemble smoothers in the annealed importance sampling framework," *Advances in Water Resources*, vol. 86, p. 231–239, 2015.

[4] G. Evensen, "Analysis of iterative ensemble smoothers for solving inverse problems," *Computational Geosciences*, vol. 22, p. 885–908, 2018.

[5] P. J. van Leeuwen, H. R. Künsch, L. Nerger, R. Potthast, and S. Reich, "Particle filters for high-dimensional geoscience applications: A review," *Quarterly Journal of the Royal Meteorological Society*, vol. 145, p. 2335–2365, 2019.

[6] P. J. van Leeuwen, "Particle Filtering in Geophysical Systems," *Monthly Weather Review*, 2009.

[7] J. Magnusson, D. Gustafsson, F. Hüsler, and T. Jonas, "Assimilation of point SWE data into a distributed snow cover model comparing two contrasting methods," *Water Resources Research*, vol. 50, p. 7816–7835, 2014.

[8] S. J. Fletcher and M. Zupanski, "A hybrid multivariate Normal and lognormal distributionfor data assimilation," *Atmospheric Science Letters*, 2006.

[9] A. Fowler and P. J. van Leeuwen, "Observation impact in data assimilation: the effect of non-Gaussian observation error," *Tellus A: Dynamic Meteorology and Oceanography*, 2013.

[10] K. P. Murphy, *Probabilistic Machine Learning: An Introduction*. MIT Press, 2022. probml.ai.

[11] N. Chopin and O. Papaspiliopoulos, *An Introduction to Sequential Monte Carlo*. Springer, 2020.

[12] A. A. Emerick and A. C. Reynolds, "History matching time-lapse seismic data using the ensemble Kalman filter with multiple data assimilations," *Computational Geosciences*, vol. 16, pp. 639–539, 2012.

[13] J. Helmert, A. Şensoy Şorman, R. Alvarado Montero, C. De Michele, P. De Rosnay, M. Dumont, *et al.*, "Review of Snow Data Assimilation Methods for Hydrological, Land Surface, Meteorological and Climate Models: Results from a COST HarmoSnow Survey," *Geosciences*, vol. 8, p. 489, 2018.

[14] C. Largeron, M. Dumont, S. Morin, A. Boone, M. Lafaysse, S. Metref, E. Cosme, T. Jonas, A. Wisntral, and S. Margulis, "Toward Snow Cover Estimation in Mountainous Areas Using Modern Data Assimilation Methods: A Review," *Frontiers in Earth Science*, vol. 8, p. 325, 2020.

[15] N. Pirk, K. Aalstad, S. Westermann, A. Vatne, A. van Hove, L. Tallaksen, M. Cassiani, and G. Katul, "Inferring surface energy fluxes using drone data assimilation in large eddy simulations," *Atmospheric Measurement Techniques Discussions*, 2022.

[16] C. Snyder, T. Bengtsson, P. Bickel, and J. Anderson, "Obstacles to High-Dimensional Particle Filtering," *Monthly Weather Review*, vol. 136, p. 4629–4640, 2008.

[17] P. J. van Leeuwen and G. Evensen, "Data Assimilation and Inverse Methods in Terms of a Probabilistic Formulation," *Monthly Weather Review*, vol. 124, p. 2898–2913, 1996.

[18] S. Särkkä, *Bayesian Filtering and Smoothing*. Cambridge University Press, 2013.

[19] N. Papadakis, E. Mémin, A. Cuzol, and N. Gengembre, "Data assimilation with the weighted ensemble Kalman filter," *Tellus A: Dynamic Meteorology and Oceanography*, vol. 62, no. 5, pp. 673–697, 2010.

[20] D. Wolpert and W. Macready, "No free lunch theorems for optimization," *IEEE Transactions on Evolutionary Computation*, 1997.

[21] R. M. Neal, *Probabilistic Inference Using Markov Chain Monte Carlo Methods*. Technical Report CRG-TR-93-1, Department of Computer Science, University of Toronto, 1993.

[22] A. Apte, M. Hairer, A. M. Stuart, and J. Voss, "Sampling the posterior: An approach to non-Gaussian data assimilation," *Physica D: Nonlinear Phenomena*, vol. 230, p. 50–64, 2007.

[23] S. A. Kolberg and L. Gottschalk, "Updating of snow depletion curve with remote sensing data," *Hydrological Processes*, vol. 20, p. 2363–2380, 2006.

[24] S. Dunne and D. Entekhabi, "An ensemble-based reanalysis approach to land data assimilation," *Water Resources Research*, vol. 41, p. W02013, 2005.

[25] M. Baba, S. Gascoin, C. Kinnard, A. Marchane, and L. Hanich, "Effect of Digital Elevation Model Resolution on the Simulation of the Snow Cover Evolution in the High Atlas," *55(7)*, pp. 5360–5378, 2019.

[26] J. I. López-Moreno, S. R. Fassnacht, S. Beguería, and J. B. P. Latron, "Variability of snow depth at the plot scale: Implications for mean depth estimation and sampling strategies," *The Cryosphere*, vol. 5, p. 617–629, 2011.

[27] N. P. Molotch and R. C. Bales, "SNOTEL representativeness in the Rio Grande headwaters on the basis of physiographics and remotely sensed snow cover persistence," *Hydrological Processes*, vol. 20, p. 723–739, 2006.

[28] B. Cluzet, M. Lafaysse, C. Deschamps-Berger, M. Vernay, and M. Dumont, "Propagating information from snow observations with CrocO ensemble data assimilation system: a 10-years case study over a snow depth observation network," *The Cryosphere*, 2022.

[29] D. Günther, T. Marke, R. Essery, and U. Strasser, "Uncertainties in Snowpack Simulations—Assessing the Impact of Model Structure, Parameter Choice, and Forcing Data Error on Point-Scale Energy Balance Snow Model Performance," *Water Resources Research*, vol. 55, p. 2779–2800, 2019.

[30] S. Westermann *et al.*, "The CryoGrid community model (version 1.0) – a multiphysics toolbox for climate-driven simulations in the terrestrial cryosphere," *Geoscientific Model Development Discussions*, 2022.

[31] D. J. C. MacKay, *Information Theory, Inference, and Learning Algorithms*. Cambridge University Press, 2003.

[32] M. Durand and S. A. Margulis, "Feasibility Test of Multifrequency Radiometric Data Assimilation to Estimate Snow Water Equivalent," *Journal of Hydrometeorology*, vol. 7, p. 443–457, 2006.

[33] E. J. Smyth, M. S. Raleigh, and E. E. Small, "Particle Filter Data Assimilation of Monthly Snow Depth Observations Improves Estimation of Snow Density and SWE," *Water Resources Research*, vol. 55, p. 1296–1311, 2019.

[34] T. Li, M. Bolic, and P. M. Djuric, "Resampling Methods for Particle Filtering: Classification, implementation, and strategies," *IEEE Signal Processing Magazine*, vol. 32, p. 70–86, 2015.

[35] D. Lindley, "The Philosophy of Statistics," *Journal of the Royal Statistical Society. Series D (The Statistician)*, 2000.

[36] G. Nearing, Y. Tian, H. V. Gupta, M. Clark, K. Harrison, and S. Weijs, "A philosophical basis for hydrological uncertainty," *Hydrological Sciences Journal*, 2016.

[37] C. K. Wikle and L. M. Berliner, "A Bayesian tutorial for data assimilation," *Physica D: Nonlinear Phenomena*, vol. 230, p. 1–16, 2007.

[38] A. Carrassi, M. Bocquet, L. Bertino, and G. Evensen, "Data assimilation in the geosciences: An overview of methods, issues, and perspectives," *WIREs Climate Change*, vol. 9, p. e535, 2018.

[39] K. Aalstad, S. Westermann, and L. Bertino, "Evaluating satellite retrieved fractional snow-covered area at a high-Arctic site using terrestrial photography," *Remote Sensing of Environment*, vol. 239, p. 111618, 2020.

[40] P. Bauer, A. Thorpe, and G. Brunet, "The quiet revolution of numerical weather prediction," *Nature*, vol. 525, p. 47–55, 2015.

[41] O. Cappé, E. Moulines, and T. Rydén, *Inference in Hidden Markov Models.* Springer, 2005.

[42] J. Dozier, E. H. Bair, and R. E. Davis, "Estimating the spatial distribution of snow water equivalent in the world's mountains," *Wiley Interdisciplinary Reviews: Water*, vol. 3, p. 461–474, 2016.

[43] B. Cluzet, M. Lafaysse, E. Cosme, C. Albergel, L.-F. Meunier, and M. Dumont, "CrocO_v1.0: a particle filter to assimilate snowpack observations in a spatialised framework," *Geoscientific Model Development*, vol. 14, p. 1595–1614, 2021.

[44] P. Cantet, M. A. Boucher, S. Lachance-Coutier, R. Turcotte, and V. V. Fortin, "Using a particle filter to estimate the spatial distribution of the snowpack water equivalent," *Journal of Hydrometeorology*, 2019.

[45] J. Odry, M.-A. Boucher, S. Lachance-Cloutier, R. Turcotte, and P.-Y. St-Louis, "Large-scale snow data assimilation using a spatialized particle filter: recovering the spatial structure of the particles," *The Cryosphere*, 2022.

[46] R. Marti, S. Gascoin, E. Berthier, M. De Pinel, T. Houet, and D. Laffly, "Mapping snow depth in open alpine terrain from stereo satellite imagery," *The Cryosphere*, vol. 10, p. 1361–1380, 2016.

[47] D. Treichler and A. Kääb, "Snow depth from ICESat laser altimetry — A test study in southern Norway," *Remote Sensing of Environment*, vol. 191, p. 389–401, 2017.

[48] H. Lievens, M. Demuzere, H. P. Marshall, R. H. Reichle, L. Brucker, I. Brangers, *et al.*, "Snow depth variability in the Northern Hemisphere mountains observed from space," *Nature Communications*, vol. 10, p. 1–12, 2019.

[49] G. Burgers, P. Jan van Leeuwen, and G. Evensen, "Analysis Scheme in the Ensemble Kalman Filter," *Monthly Weather Review*, vol. 126, p. 1719–1724, 1998.

[50] P. J. van Leeuwen, "A consistent interpretation of the stochastic version of the Ensemble Kalman Filter," *Quarterly Journal of the Royal Meteorological Society*, vol. 146, p. 2815–2825, 2020.

[51] J. Revuelto, C. Azorin-Molina, E. Alonso-González, A. Sanmiguel-Vallelado, F. Navarro-Serrano, I. Rico, and J. Ignacio López-Moreno, "Meteorological and snow distribution data in the Izas Experimental Catchment (Spanish Pyrenees) from 2011 to 2017," *Earth System Science Data*, vol. 9, p. 993–1005, 2017.

[52] K. M. Banner, K. M. Irvine, and T. J. Rodhouse, "The use of Bayesian priors in Ecology: The good, the bad and the not great," *Methods in Ecology and Evolution*, vol. 11, pp. 882–889, Aug. 2020.

[53] J. Revuelto, J. I. López-Moreno, and E. Alonso-González, "Light and Shadow in Mapping Alpine Snowpack With Unmanned Aerial Vehicles in the Absence of Ground Control Points," *Water Resources Research*, vol. 57, p. e2020WR028980, 2021.

[54] C. Deschamps-Berger, B. Cluzet, M. Dumont, M. Lafaysse, E. Berthier, P. Fanise, and S. Gascoin, "Improving the Spatial Distribution of Snow Cover Simulations by Assimilation of Satellite Stereoscopic Imagery," *Water Resources Research*, vol. 58, p. e2021WR030271, 2022.

[55] B. Cluzet, J. Revuelto, M. Lafaysse, F. Tuzet, E. Cosme, G. Picard, *et al.*, "Towards the assimilation of satellite reflectance into semi-distributed ensemble snowpack simulations," *Cold Regions Science and Technology*, vol. 170, p. 102918, 2020.

---

## Referee Report (RR1)

MuSA: The Multiple Snow Data Assimilation System (v1.0)
Esteban Alonso-González et al.

Review by Bertrand Cluzet (round 2)

General Statement

The authors have made a considerable effort in answering both reviewers' comments in a very neat, convincing, and rigorous manner. The scientific quality of the paper, which was excellent, is even improved.
My only concern at this stage is the fact that the readability of Sec. 3 is unchanged, in my humble opinion (see my minor comment #2.2 in the author's answers). Only marginal reformulations and adjustments have been made and the following comments, rejected (#2.10, #2.13, #2.70, #2.73, #2.79 #2.85, #2.91, the latter being the most prominent one). I think that the level of justification is sufficient for me to leave that up to editorial decision.

Below are some comments on discussion points which could seem subject to controversy but that the authors have perfectly addressed. Points 1 and 3 might require tiny technical changes.

Comments/Technical changes

1. #2.8: Agreed. The apparent disagreement comes from the fact that I was thinking about representativeness errors coming from a scale mismatch between point observations and model grid points but did not state this properly. I apologize for this lack of clarity. The answer to #2.108/#2.112 makes it clear that there is no scale mismatch between observations and the model, since observations are abundant (@1m) and can be aggregated into the model geometry (@5m). I suggest adding a short sentence about this aggregation in the beginning of Sec. 4.1
2. #2.14: Misunderstanding, my statement was that the way the authors described the message of Largeron et al., (2020) was misleading. I agree with the authors and am very much satisfied with the changes to the corresponding part of Sec. 6.
3. #2.22: Thanks for the addition to the caption. I understand that the chosen date corresponds to the maximal snow depth observation of Fig. 3 and 4. Please consider pointing that out for the curious readers.
4. #2.35: OK
5. #2.57: OK (not important)
6. #2.78: OK, yes of course in the case of state variable perturbations one can introduce physical inconsistencies.
7. #2.115: OK

---

## Author Response (AR2)

Dear Editor,

Thank you very much for your positive comments on our response. We are pleased to submit a revised version of the manuscript that includes addressing Bertrand Cluzet's technical corrections in his points 1 and 3.
We have also included a statement at the beginning of Section 3 as suggested, that warns readers that are well versed in DA methods to consider skipping this section. We believe that with these minor modifications all comments are now addressed.

Kind regards,
The authors